# Beyond Naïve Prompting: Strategies for Improved Zero-shot Context-aided Forecasting with LLMs

## Abstract

Forecasting in real-world settings requires models to integrate not only historical data but also relevant contextual information, often available in textual form. While recent work has shown that large language models (LLMs) can be effective context-aided forecasters via naïve direct prompting, their full potential remains underexplored. We address this gap with 4 strategies, providing new insights into the zero-shot capabilities of LLMs in this setting. ReDP improves interpretability by eliciting explicit reasoning traces, allowing us to assess the model's reasoning over the context independently from its forecast accuracy. CorDP leverages LLMs solely to refine existing forecasts with context, enhancing their applicability in real-world forecasting pipelines. IC-DP proposes embedding historical examples of context-aided forecasting tasks in the prompt, substantially improving accuracy even for the largest models. Finally, RouteDP optimizes resource efficiency by using LLMs to estimate task difficulty, and routing the most challenging tasks to larger models. Evaluated on different kinds of context-aided forecasting tasks from the CiK benchmark, our strategies demonstrate distinct benefits over naïve prompting across LLMs of different sizes and families. These results open the door to further simple yet effective improvements in LLM-based context-aided forecasting.

## 1 Introduction

Probabilistic time series forecasting is essential for optimal decision-making, involving predicting the evolution of various quantities over time, as well as estimating the likelihood of various scenarios (Hyndman & Athanasopoulos, 2021; Peterson, 2017). This problem has been extensively studied by both the statistical and machine learning communities (Hyndman et al., 2008; Box et al., 2015; Hyndman & Athanasopoulos, 2021), culminating in different methods such as classical methods (Hyndman et al., 2008; Gardner Jr., 1985), deep learning methods (Salinas et al., 2020; Drouin et al., 2022; Ashok et al., 2024), hybrid methods (Oreshkin et al., 2019), and more recently, foundation models (Rasul et al., 2023; Ansari et al., 2024; Woo et al., 2024). Research in forecasting has largely focused on building models that use numerical historical observations and engineered covariates, while in the real-world, accurate forecasts rely not only on them but also on contextual information about the problem or task in hand (Hyndman & Athanasopoulos, 2021). With the realistic assumption that such prior information can be expressed flexibly in natural language, a new, multimodal problem setting of *context-aided forecasting* has recently emerged in the literature (Jin et al., 2024; Liu et al., 2024a;c; Kong et al., 2025).

Several methods have been proposed for context-aided forecasting, and can be broadly classified into two types (Zhang et al., 2025): those that rely on training models on specific context-aided forecasting tasks (Jin et al., 2024; Zhang et al., 2023; Xu et al., 2024; Emami et al., 2024; Wang et al., 2024; Liu et al., 2024a; Zhou et al., 2025) and those that do not require training, and purely leverage the zero-shot capabilities of LLMs for context-aided forecasting (Merrill et al., 2024; Gruver et al., 2024; Requeima et al., 2024; Williams et al., 2025). Among those that use LLMs zero-shot, only simple strategies have been explored, such as direct prompting (Williams et al., 2025) and autoregressive LLM processes (Requeima et al., 2024) among others. These methods involve simply feeding historical numerical data and textual context into the LLM and generating forecasts timestep-by-timestep. The potential for sophisticated strategies to enhance forecast accuracy, efficiency, and interpretability of models remains largely unexplored.

In this work, we systematically investigate 4 strategies that address and improve different aspects of zero-shot forecasting with LLMs (illustrated in Figure 1):

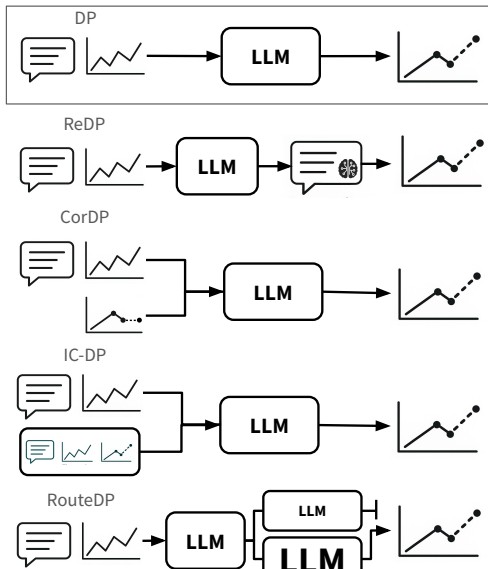

- **ReDP: Direct Prompting with Reasoning over Context** (Section 4) improves interpretability by prompting models to output explicit context reasoning traces and comparing them with gold standard reasoning traces, providing an additional dimension of evaluation. This helps uncover a key failure mode of models: inability to apply their reasoning on their forecasts.

- **CorDP: Direct Prompting for Forecast Correction** (Section 5): utilizes LLMs to solely modify existing probabilistic forecasts with context, instead of forecasting from scratch. This improves models by up to 50%, and allows for practical adoption of LLMs in existing forecasting workflows.

- **IC-DP: In-Context Direct Prompting** (Section 6) explores prompting LLMs with historical examples of context-aided forecasting tasks, showing that they can substantially improve accuracy even for the largest models.

- **RouteDP: Direct Prompting with Model Routing** (Section 7) enables accurate forecasting under resource constraints by using a small model for easy tasks and delegating more difficult ones to a larger model, guided by a router. We observe substantial improvements in forecast accuracy at a fraction of the cost.

Figure 1: Direct Prompt (DP) (Williams et al., 2025) prompts the LLM with the context and historical data of a context-aided forecasting task. Our work explores nuanced strategies with distinct benefits over naïve prompting: Direct Prompting with Reasoning over Context (ReDP) which improves interpretability, Direct Prompting for Forecast Correction (CorDP) which bootstraps LLMs on prior forecasts and modifies them with context, In-Context Direct Prompting (IC-DP) that boosts performance through exemplars and Direct Prompting with Model Routing (RouteDP) which enables accurate forecasting under resource constraints.

In what follows, we evaluate these strategies on diverse zero-shot context-aided forecasting tasks from the Context-Is-Key (CiK) benchmark (Williams et al., 2025), and demonstrate distinct benefits that each strategy offers over direct prompting. We show that such prompting strategies can prove extremely effective in studying various capabilities of models and obtaining significantly better context-aided forecasts from LLMs across different sizes and families (Yang et al., 2024; Grattafiori et al., 2024).

## 2 Related Work

### 2.1 Large Models for Forecasting

Historically, classical methods such as ETS, ARIMA, and ensembles of such models have been at the cornerstone of time series forecasting (Hyndman et al., 2008; Box et al., 2015). Following the explosion of deep learning in modalities such as vision and language, such methods were explored for forecasting tasks, starting with RNN-based and LSTM-based models (Hewamalage et al., 2021; Salinas et al., 2020), followed by transformer methods (Lim et al., 2021; Wu et al., 2021; Zhou et al., 2021; Drouin et al., 2022; Wen et al., 2023; Nie et al.; Ashok et al., 2024). Recently, following the success of pretrained large models in language (Brown et al., 2020), the time series community has also proposed foundation models for forecasting tasks (Rasul et al., 2023; Goswami et al., 2024; Woo et al., 2024; Ansari et al., 2024) pretrained on a large amount of time series data, and shown that they can output strong forecasts on unseen datasets zero-shot, outperforming models trained on those datasets. Research in large models for forecasting continues to grow, with efforts on better understanding their capabilities (Potosnak et al., 2024) and limitations (Liang et al., 2024). A separate stream of research has explored using large language models (LLMs) for forecasting tasks. Gruver et al. (2024) propose LLMTime, where an LLM is prompted to autoregressively generate a digit for each

timestep in the prediction horizon of a forecasting task. They demonstrate surprising performance using LLMs for forecasting compared to time series models trained on specific datasets. Requeima et al. (2024) improve this method to propose LLM Processes, where they show that the exact formatting of the prompt as well as the scaling factor used with the time series both matter. (Liu et al., 2024b) propose LSTPrompt, which uses chain-of-thought methods to improve the performance of LLMs in quantitative forecasting tasks. As LLMs continue to improve regularly, several works have continued exploring the value of LLMs and limitations in forecasting (Tang et al., 2025). Our work is related to these as we also aim to improve the forecasting performance of LLMs, however we operate in a setting where quantitative forecasting capabilities are insufficient, and tasks require models to understand the textual context to succeed in. Thereby, this setting of context-aided forecasting comes with different challenges and requires different capabilities from models (Williams et al., 2025), which we study with our methods.

## 2.2 Context-aided Forecasting Methods

One key capability that LLMs offer is the ability to condition on complementary side-information in text (Jin et al., 2024; Liu et al., 2024c; Wang et al., 2024; Xu et al., 2024; Liu et al., 2024a). Jin et al. (2024) propose Time-LLM, a multimodal model trained to use dataset-level metadata in addition to historical data for forecasting. The authors propose to use an LLM to encode the metadata and a transformer architecture to process time series, and train them jointly. Xu et al. (2024); Liu et al. (2024a) propose similar multimodal architectures that can be trained for forecasting tasks on a specific dataset, where additional text information is available per time series window. Liu et al. (2024c) expand the setting to a multi-dataset setup and propose objective functions that allow better training on multiple time series based on their textual metadata, while preventing "domain confusion". As opposed to training a time series model from scratch, Wang et al. (2025) propose an architecture that uses a pretrained time series foundation model in conjunction with an LLM in a similar setup, proposing to only train adapters between them. Wang et al. (2024) adapt a purely-LLM-based approach to this, fine-tuning LLMs such as Qwen-7B on dataset-specific context-aided forecasting tasks, demonstrating the value of pure-LLM approaches. However, all these above methods involve training the model, which specializes the model to perform well solely on the time series and contexts that it is trained on (Zhang et al., 2025). Gruver et al. (2024); Requeima et al. (2024); Williams et al. (2025); Merrill et al. (2024) explore a different, zero-shot forecasting setup where the goal is to perform well on a diverse range of contexts and time-series. The focus in this setting is on how well models can use unambiguous, relevant context to succeed in forecasting scenarios, instead of on specializing models to specific scenarios (Wang et al., 2025). Both Gruver et al. (2024); Requeima et al. (2024) demonstrate with preliminary results the ability of LLMs to successfully condition on textual information. Merrill et al. (2024) evaluate a series of LLMs on context-aided forecasting tasks generated by GPT-4, and show that there is a huge gap between the performance of humans and that of LLMs on these tasks. To study the context-aided forecasting abilities of LLMs systematically, Williams et al. (2025) propose a real-world evaluation benchmark of 71 zero-shot context-aided forecasting tasks across 8 different domains, each of which requires models to necessarily use the textual context to succeed in. The authors evaluate a range of LLMs zero-shot, with the LLMP method of Requeima et al. (2024), and a faster and simpler prompting method they propose called Direct Prompt (DP), and demonstrate promising results with different LLMs. Our work builds on these results, going beyond early work on naïve direct prompting (Williams et al., 2025) and exploring variants which, as we demonstrate, can offer complementary advantages and reveal interesting insights into model capabilities, while significantly improving their performance.

# 3 Background

## 3.1 Problem Setting

The goal of context-aided forecasting is to produce statistical forecasts by incorporating relevant side information (i.e. context) (Williams et al., 2025; Wang et al., 2024; Kong et al., 2025; Zhang et al., 2025). We focus on the case where the context information is available in textual form, which is well-studied in the literature. Formally, let $\mathbf{X}_H = [X_1, \ldots, X_t]$ denote a sequence of random variables representing historical observations at discrete time steps, with each $X_\tau \in \mathcal{X} \subseteq \mathbb{R}$. The future observations are denoted by

$\mathbf{X}_F = [X_{t+1}, \ldots, X_T]$. The textual *context*, $\mathbf{C}$, provides additional information pertinent to forecasting $\mathbf{X}_F$, supplementing the information contained in $\mathbf{X}_H$. The forecasting task is thus to estimate the conditional distribution $P(\mathbf{X}_F \mid \mathbf{X}_H, \mathbf{C})$.

## 3.2  Direct Prompt

Williams et al. (2025) introduce a method for context-aided forecasting that instructs an LLM to generate forecasts as a structured output for all of the required timestamps, given the history and context information of a context-aided forecasting task. With this method, termed Direct Prompt (DP), it has been shown that several instruction-tuned LLMs can improve their performance with context on multiple benchmark datasets (Williams et al., 2025; Kupferschmidt et al., 2024), indicating that they are capable of interpreting textual context to improve their forecasts, providing them a sizeable advantage over quantitative methods such as statistical models and time-series foundation models that cannot use context (Kong et al., 2025; Zhang et al., 2025). They further show that such naive prompting can elicit accurate numerical predictive distributions from LLMs without the computational overhead of elaborate autoregressive procedures as in prior work that prompt the LLM digit-by-digit (Gruver et al., 2024; Requeima et al., 2024). Our work pushes the limit of direct prompting methods with 4 different strategies that offer advantages over direct prompting in different dimensions, and allow obtaining considerably better performance with minimal overhead.

## 3.3  Experimental Protocol

Following the evaluation setup for zero-shot context-aided forecasting methods used in prior work (Zhang et al., 2025), we use the Context-Is-Key (CiK) benchmark (Williams et al., 2025) to evaluate zero-shot forecasting methods. The CiK benchmark is a collection of 71 manually designed context-aided forecasting tasks from 2644 time series spanning 7 real-world domains namely Climatology (Sengupta et al., 2018); Economics (U.S. Bureau of Labor Statistics, 2024), Energy (Godahewa et al., 2021), Mechanics (Gamella et al., 2024), Public Safety (Ville de Montréal, 2020), Transportation (Chen et al., 2001), and Retail (Godahewa et al., 2021) across diverse sampling frequencies, with observations ranging from every 10 minutes to monthly intervals. The tasks in CiK encompass diverse types of contextual information and require models to use various reasoning capabilities to succeed, making it a comprehensive test of context-aided forecasting abilities. Crucially, CiK is the only benchmark where accurate forecasts cannot be achieved without incorporating the context, making it uniquely suitable for evaluating zero-shot context-aided forecasting capabilities (Zhang et al., 2025). This distinguishes it from other benchmarks (Merrill et al., 2024; Liu et al., 2024a; Wang et al., 2024; 2025) where context may not be always essential for high-quality forecasts (Zhang et al., 2025). Additionally, the tasks in CiK are designed to mitigate memorization effects, making it suitable for evaluating LLMs.

We use the Region-of-interest CRPS (RCRPS) metric to evaluate context-aided forecasting performance (Williams et al., 2025), which prioritizes context-sensitive windows called the region of interest (RoI) and accounts for any hard constraints that could have been mentioned in the context (e.g., can never be negative). We use one realization of each of the 71 tasks in CiK for our experiments, and report the average RCRPS as the performance metric. We experiment with instruction-tuned models from the Qwen and Llama families of models, namely, Qwen-2.5-0.5B, 1.5B, 3B, 7B, 14B, 32B, 72B (Yang et al., 2024) and Llama-3.2-1B, 3B, Llama-3-8B, Llama-3.3-70B, Llama-3.1-405B (Grattafiori et al., 2024). We use these models for consistency with prior work (Williams et al., 2025). To obtain probabilistic forecasts, we draw 25 samples from the output distribution of all models. Additional details on the metric are in appendix A, and additional details on the implementation of each model are provided in App. G.

In the following sections, we introduce and evaluate our four proposed strategies, demonstrating their respective improvements over direct prompting (DP) on the CiK benchmark and discussing their implications for context-aided forecasting.

# 4 ReDP: Direct Prompting with Reasoning over Context

## 4.1 The Need for Interpretability

Context-aided forecasting requires models to perform two sequential tasks: first, correctly reasoning about how context should influence the forecast, and second, translating this reasoning into accurate quantitative forecasts. Current evaluation approaches focus exclusively on final forecasting accuracy, treating the model as a black box, providing limited interpretability into the model. This limitation becomes particularly problematic when models fail: we cannot determine whether the failure stems from poor reasoning about the context or from an inability to apply its reasoning to produce accurate forecasts. Such ambiguity hinders our ability to diagnose the limitations of models. Recent work has further emphasized the importance of interpretability in understanding model behavior (Rudin, 2019; Doshi-Velez & Kim, 2017; Lipton, 2018). By explicitly evaluating reasoning quality alongside forecasting performance, we can disentangle these two capabilities and potentially develop targeted improvements to models.

## 4.2 ReDP as a Diagnostic Tool for Reasoning Quality Evaluation

To allow evaluating the model's reasoning process, we propose a modification to Direct Prompt, instructing the LLM to also produce an explicit reasoning trace before the forecast, where the LLM is asked to explain in detail how it would use the context in its forecast (see appendix C.1 for the prompt). This approach builds on the chain-of-thought prompting literature (Wei et al., 2022), where reasoning traces of LLMs have been used for evaluation (Lightman et al., 2023) and diagnosis (Fu et al., 2023) of the model's reasoning. We call this method ReDP (Direct Prompting with Reasoning over Context) akin to the ReAct method (Yao et al., 2023).

Next, we design a protocol for evaluating reasoning correctness. We first curate gold standard reasoning traces for tasks where context affects only a specific region of interest (RoI), ensuring there is a single, well-defined reasoning path. We generate these gold standards using GPT-4.1 (Achiam et al., 2023), asking it to provide a detailed account of how a task's context should influence the forecast, then manually verifying and correcting them if necessary. To evaluate a target model's reasoning quality, we use GPT-4.1 as an LLM judge (Fu et al., 2023; Lightman et al., 2023; Gu et al., 2024), comparing the model's reasoning trace against the gold standard. The judge determines correctness by checking if key points from the gold standard appear in the model's trace. Prompts used for generation and verification of reasoning traces are in Appendix C.2. The gold standard reasoning traces are provided in Appendix C.4. Examples of correct and incorrect reasoning traces as judged by the LLM are provided in Appendix C.5.

Finally, to understand whether correct reasoning translates to improved forecasting performance, we measure the relative improvement in CRPS within the RoI when context is provided versus when it is not, using a 50% improvement threshold to identify significant gains. Additional details on the protocol are provided in Appendix C.2.

## 4.3 Reasoning Quality Analysis

Our analysis reveals several key insights about the reasoning capabilities of models. First, we find that with ReDP, models sometimes fail to follow the instruction in the prompt and do not produce a reasoning trace before the forecast, despite several retries. Interestingly, all tested models were prone to this failure, with different models failing in different tasks. We only include models that produce a reasoning trace in at least 75% of the tasks to ensure statistical significance, which leaves out the smallest models (<3B) from the analysis. We also find that on average, models achieve the same performance with ReDP and DP (results in appendix Appendix C.3).

The results of the reasoning quality analysis are in Table 1. First, we find that the percentage of tasks in which the model's reasoning is correct improves with the model size, across both families, and as does the percentage of tasks with a meaningful improvement with context, as seen in the second and third columns of the table. Next, we find that the smaller models (3B-10B) can reason correctly in only a portion of the tasks (30%-70%), and further, successfully apply it to improve with context in only a fraction of the tasks

| Model | Correct Reasoning | Improvement with Context | Correct Reasoning and Improvement with Context | Correct Reasoning but no Improvement with Context | Wrong Reasoning but Improvement with Context | Wrong Reasoning and no Improvement with Context |
|---|---|---|---|---|---|---|
| Llama-3.2-3B-Inst | 38.9% | 33.3% | 16.7% | 22.2% | 16.7% | 44.4% |
| Llama-3.1-8B-Inst | 70.0% | 30.0% | 30.0% | 40.0% | 0.0% | 30.0% |
| Llama-3.3-70B-Inst | 100.0% | 78.9% | 78.9% | 21.1% | 0.0% | 0.0% |
| Llama-3.1-405B-Inst | 90.0% | 80.0% | 70.0% | 20.0% | 10.0% | 0.0% |
| Qwen-2.5-3B-Inst | 46.7% | 0.0% | 0.0% | 46.7% | 0.0% | 53.3% |
| Qwen-2.5-7B-Inst | 84.2% | 42.1% | 36.8% | 47.4% | 5.3% | 10.5% |
| Qwen-2.5-14B-Inst | 95.0% | 80.0% | 75.0% | 20.0% | 5.0% | 0.0% |
| Qwen-2.5-32B-Inst | 94.7% | 68.4% | 68.4% | 26.3% | 0.0% | 5.3% |
| Qwen-2.5-72B-Inst | 94.7% | 78.9% | 73.7% | 21.1% | 5.3% | 0.0% |

Table 1: Results of the reasoning quality analysis using ReDP. The first two columns show the percentage of tasks where models produce correct reasoning traces and achieve meaningful improvements with context (at least 50% RCRPS reduction in the Region-of-Interest), respectively. The remaining columns show the joint distribution: correct reasoning with/without improvement, and incorrect reasoning with/without improvement. We find that smaller models (<10B) often reason correctly but fail to apply their reasoning, while larger models (>70B) achieve both correct reasoning and successful application.

(Qwen2.5-3B-Inst even failing to apply even in a single task). Next, mid-sized models (Qwen 14B, 32B) and large models (>32B) can reason correctly in almost all the tasks, and apply their reasoning correctly in about 70% of the tasks. These findings are reflected in Table 1 in the fourth and fifth columns, where the latter indicating the percentage of tasks with "Correct Reasoning but no Improvement with Context" is consistently at 20-30% for the mid-sized and large models, while it goes up to 40-50% for the smaller models.

Finally, as seen in the last two columns, in the absence of correct reasoning, tasks rarely or never see a success in improvement with context, indicating that the reasoning trace is a faithful reflection of the model's forecasting process with context. Examples from the analyses showcasing their evaluated reasoning correctness and improvement with context are provided in Appendix C.7. These findings suggest that enhancing models' ability to effectively leverage their reasoning traces is a promising and tractable direction for future research. For smaller models (<10B), improving the quality of reasoning itself also appears to be a crucial factor for better performance on context-aided forecasting tasks. In this context, approaches such as the RL-based post-training (DeepSeek-AI et al., 2025) may offer a strong foundation for further exploration.

## 5 CorDP - Direct Prompting for Forecast Correction

### 5.1 Limitations of LLM-based Context-Aided Forecasting

Real-world forecasting applications require both high accuracy in quantitative predictions and the ability to incorporate contextual information when available. However, integrating contextual reasoning with specialized quantitative models presents a major implementation challenge. Quantitative forecasting systems typically employ specialized, highly-tuned models that achieve optimal performance for specific use cases (Petropoulos et al., 2022; Januschowski et al., 2024). These models are carefully selected and optimized for their particular domain, making them difficult to replace without significant performance degradation. Current approaches to context-aided forecasting attempt to replace these specialized quantitative models entirely with LLMs (Williams et al., 2025; Requeima et al., 2024) which can lead to suboptimal performance given that such models were never intended to perform forecasting. Rather than replacing quantitative models entirely, we propose an augmentation approach that preserves their specialized forecasting capabilities while adding contextual reasoning. This approach leverages the strengths of both paradigms: the quantitative accuracy of domain-specific models and the contextual reasoning capabilities of LLMs.

### 5.2 Forecast Correction: A Practical Approach to Context-Aided Forecasting

We propose Direct Prompting for Forecast Correction (CorDP), with which an LLM is repurposed to be a *forecast corrector*, instead of being repurposed as a *forecaster* as in prior work (Williams et al., 2025;

| Model | Direct Prompt (DP) | Median Corrector (Median-CorDP) | | | SampleWise Corrector (SampleWise-CorDP) | | |
|---|---|---|---|---|---|---|---|
| | | LAG-LLAMA | CHRONOS LARGE | ARIMA | LAG-LLAMA | CHRONOS LARGE | ARIMA |
| Llama3.2-1B-Inst | **0.396 ± 0.027** | **0.394 ± 0.004** | 0.515 ± 0.007 | 0.612 ± 0.018 | 0.541 ± 0.009 | 0.634 ± 0.005 | 0.672 ± 0.015 |
| Llama3.2-3B-Inst | 0.687 ± 0.025 | **0.344 ± 0.011** | 0.455 ± 0.009 | 0.573 ± 0.022 | 0.509 ± 0.026 | 0.423 ± 0.007 | 0.663 ± 0.031 |
| Llama3-8B-Inst | 0.543 ± 0.026 | **0.315 ± 0.004** | 0.453 ± 0.005 | 0.571 ± 0.004 | 0.426 ± 0.009 | 0.410 ± 0.004 | 0.636 ± 0.010 |
| Llama3.3-70B-Inst | 0.230 ± 0.006 | 0.281 ± 0.002 | 0.251 ± 0.004 | 0.352 ± 0.006 | 0.223 ± 0.004 | **0.215 ± 0.004** | 0.311 ± 0.007 |
| Llama3.1-405B-Inst | **0.173 ± 0.003** | 0.278 ± 0.009 | 0.226 ± 0.004 | 0.257 ± 0.008 | 0.199 ± 0.006 | 0.194 ± 0.004 | 0.229 ± 0.008 |
| Qwen2.5-0.5B-Inst | 0.592 ± 0.027 | 0.633 ± 0.002 | 0.801 ± 0.003 | 0.761 ± 0.054 | **0.494 ± 0.008** | 0.644 ± 0.076 | 0.655 ± 0.055 |
| Qwen2.5-1.5B-Inst | 0.616 ± 0.018 | **0.426 ± 0.013** | 0.537 ± 0.003 | 0.682 ± 0.006 | 0.522 ± 0.018 | 0.474 ± 0.005 | 0.719 ± 0.013 |
| Qwen2.5-3B-Inst | 0.424 ± 0.017 | 0.490 ± 0.005 | 0.491 ± 0.004 | 0.597 ± 0.009 | **0.398 ± 0.028** | 0.451 ± 0.005 | 0.512 ± 0.032 |
| Qwen2.5-7B-Inst | 0.401 ± 0.006 | 0.419 ± 0.004 | 0.641 ± 0.008 | 0.633 ± 0.008 | **0.382 ± 0.007** | 0.402 ± 0.020 | 0.540 ± 0.011 |
| Qwen2.5-14B-Inst | **0.247 ± 0.006** | 0.315 ± 0.003 | 0.334 ± 0.006 | 0.423 ± 0.004 | 0.364 ± 0.006 | 0.410 ± 0.006 | 0.471 ± 0.009 |
| Qwen2.5-32B-Inst | 0.397 ± 0.008 | **0.248 ± 0.004** | 0.272 ± 0.005 | 0.329 ± 0.008 | 0.310 ± 0.005 | 0.338 ± 0.007 | 0.414 ± 0.009 |
| Qwen-2.5-72B-Inst | **0.202 ± 0.009** | 0.319 ± 0.008 | 0.358 ± 0.010 | 0.428 ± 0.009 | 0.255 ± 0.010 | 0.322 ± 0.010 | 0.386 ± 0.010 |
| GPT-4o | 0.317 ± 0.009 | 0.253 ± 0.004 | 0.240 ± 0.004 | 0.354 ± 0.007 | **0.184 ± 0.004** | 0.196 ± 0.004 | 0.251 ± 0.008 |
| GPT-4o-mini | 0.389 ± 0.010 | 0.364 ± 0.006 | 0.340 ± 0.004 | 0.516 ± 0.005 | 0.302 ± 0.008 | **0.296 ± 0.005** | 0.415 ± 0.011 |
| Base Quantitative Forecaster | - | 0.382 ± 0.011 | 0.492 ± 0.004 | 0.636 ± 0.014 | 0.382 ± 0.011 | 0.492 ± 0.004 | 0.636 ± 0.014 |

Table 2: Aggregate results on CiK, accompanied by standard errors. The best performing method for each model is in **bold**. Results on various groups of tasks are in Appendix D.2.

Requeima et al., 2024; Gruver et al., 2024). With CorDP, an LLM bootstraps off probabilistic forecasts from a quantitative model, and is instructed to "correct" the quantitative forecast based on the context, only modifying it where the LLM believes the context provides relevant insights. We propose two variations of this approach:

- SampleWise-CorDP: The LLM corrects each sample of the probabilistic forecasts.

- Median-CorDP: The LLM corrects the median of the forecast multiple times.

SampleWise-CorDP preserves the original forecast distribution, while Median-CorDP uses the summary of the distribution and produces a new context-aided forecast distribution. This mimics how human forecasters regularly incorporate context information to improve forecasts post-hoc, called judgmental correction in the forecasting literature (Hyndman & Athanasopoulos, 2021); CorDP instead proposes to do so with LLMs, zero-shot. With CorDP, the quantitative forecasting accuracy of the base model is preserved, and the LLM bears minimal computational load since it only makes corrections to existing forecasts. This design also enables easy integration with existing forecasting pipelines, as it only requires adding a correction step without modifying the core quantitative forecasting infrastructure. The CorDP prompt is provided in Appendix D.1.

### 5.3 Performance of CorDP

Table 2 presents results aggregated across all tasks from CiK. CorDP methods achieve the best performance across 10/14 LLMs and equal performance with 1 LLM, benefiting models across different scales and families, with improvements of up to 50%. SampleWise-CorDP performs the best in 6/14 LLMs where CorDP is the best, while Median-CorDP outperforms it with the other 4/14 LLMs. The performance of CorDP methods differ widely depending on the quantitative forecaster used, with all winning methods except one using Lag-Llama's forecasts. This is likely because Lag-Llama is the best performing quantitative forecaster (as seen in the last row of the table), highlighting the importance of selecting an appropriate base forecaster for CorDP. Many models perform strictly better than the base forecaster, when used with CorDP. However, some models sometimes seem to deteriorate the base quantitative forecast, particularly the smallest models (Llama3.2-1B-Inst and Qwen2.5-1.5B-Inst). These models still improve over Direct Prompting, suggesting that conditioning on base forecasts may be useful nevertheless. CorDP methods also do not surpass Direct Prompting with the largest model (Llama3.1-405B-Inst). This suggests that these models may be inherently better forecasters with direct prompting, or that they are unable to bootstrap off forecasts like the other models do.

Beyond aggregate performance, looking at the performance of CorDP in different kinds of tasks (results in Appendix D.2), we find that SampleWise-CorDP has an advantage on tasks with a partial RoI (region of interest, the context-sensitive region within the prediction window), achieving the best performance in most

models, both within and outside the RoI. Median-CorDP however has a clear advantage on tasks where the shape of the entire forecast is influenced by the context, achieving the best performance in half the models, and trailing closely with DP in the other. Median-CorDP overwhelmingly outperforms DP and bags the best performance in tasks with constraints, sometimes achieving perfect performance with large models. This shows that when choosing between CorDP methods, the kind of tasks that will be encountered is an important factor to consider. Example forecasts with the two CorDP methods are provided in Appendix D.4.

These results establish CorDP as a practical solution for context-aided forecasting that allow leveraging existing numerical forecasting tools, augmenting them with the advanced reasoning abilities of LLMs , though success depends on careful LLM selection. Future work could explore fine-tuning LLMs with CorDP to better match the forecast distributions of specific quantitative forecasters. Analyzing the distributional properties of LLM-generated versus quantitative forecasts would also be useful for improving CorDP performance and understanding when the method is most effective.

## 6 IC-DP - In-Context Direct Prompting

### 6.1 Leveraging Historical Examples

Real-world forecasting applications typically deal with domain-specific contexts that repeat over time, such as seasonal heat waves in electricity consumption forecasting. Previous work has confirmed this pattern across various domains, where similar contextual events recur with varying impact on forecasts (Wang et al., 2024; 2025). In an ideal case, a model could be trained to understand and forecast with the domain-specific contexts. However, this approach requires significant overhead in model selection, training, and maintenance. In contrast, LLMs can leverage in-context learning to improve performance by learning from examples provided in the prompt (Brown et al., 2020), offering a zero-shot alternative to domain-specific training. We explore this capability for context-aided forecasting by demonstrating to models how past contexts affected forecasts through in-context examples.

### 6.2 IC-DP: In-Context Forecasting

To test the in-context forecasting capabilities of LLMs, we modify Direct Prompt to include example context-aided forecasting tasks in the prompt, providing their respective histories, contexts, and ground truths (see Appendix E.1 for the prompt). We call this "In-Context Direct Prompt" (IC-DP). IC-DP allows understanding the extent of the zero-shot performance of LLMs, in context-aided forecasting settings where past examples with similar contexts are available. We evaluate IC-DP using a single past instance of the same context-aided forecasting task as the example, where the time series and textual context differ but the context's influence on the prediction window is the same (see Appendix E.3 for examples).

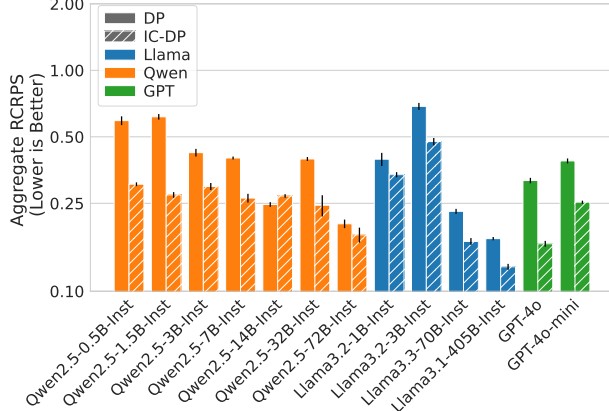

Figure 2: Aggregate results of models with DP and IC-DP respectively, accompanied by standard errors. IC-DP significantly improves the performance of 12/13 tested models.

### 6.3 Performance Gains

Aggregate results are shown in Figure 2, while results on various kinds of tasks are in Table 3. IC-DP improves the performance of 12/13 tested models, indicating that models can leverage their context-aided forecasting capabilities more effectively with just a single in-context example. IC-DP provides substantial improvements over DP across models of various sizes, with small models improving by 14-55.7%, particularly the smallest Qwen models, while mid-size and large models show 20-40% improvements. This also shows that to reach a certain level of performance, one can use much smaller models with IC-DP, compared to with DP. IC-DP provides outsized improvements in tasks where the entire forecast is shaped by the context (i.e. tasks

| Model | RoI RCRPS | | non-RoI RCRPS | | RCRPS of tasks with full RoI | | Constraints RCRPS | |
|---|---|---|---|---|---|---|---|---|
| | DP | IC-DP | DP | IC-DP | DP | IC-DP | DP | IC-DP |
| Llama3.2-1B-Inst | $0.336 \pm 0.026$ | $\mathbf{0.218 \pm 0.006}$ | $0.248 \pm 0.026$ | $\mathbf{0.187 \pm 0.006}$ | $0.467 \pm 0.041$ | $\mathbf{0.428 \pm 0.015}$ | $0.275 \pm 0.092$ | $\mathbf{0.007 \pm 0.031}$ |
| Llama3.2-3B-Inst | $0.281 \pm 0.013$ | $\mathbf{0.147 \pm 0.005}$ | $0.162 \pm 0.013$ | $0.209 \pm 0.005$ | $1.004 \pm 0.040$ | $\mathbf{0.679 \pm 0.031}$ | $1.030 \pm 0.090$ | $\mathbf{0.163 \pm 0.068}$ |
| Llama3.3-70B-Inst | $\mathbf{0.105 \pm 0.003}$ | $0.134 \pm 0.003$ | $0.182 \pm 0.003$ | $\mathbf{0.122 \pm 0.003}$ | $0.289 \pm 0.011$ | $\mathbf{0.194 \pm 0.010}$ | $\mathbf{0.000 \pm 0.024}$ | $0.025 \pm 0.020$ |
| Llama3.1-405B-Inst | $0.126 \pm 0.004$ | $\mathbf{0.094 \pm 0.004}$ | $0.150 \pm 0.004$ | $\mathbf{0.115 \pm 0.004}$ | $0.196 \pm 0.005$ | $\mathbf{0.146 \pm 0.006}$ | $0.004 \pm 0.009$ | $\mathbf{0.000 \pm 0.012}$ |
| Qwen2.5-0.5B-Inst | $0.339 \pm 0.010$ | $\mathbf{0.288 \pm 0.004}$ | $\mathbf{0.129 \pm 0.010}$ | $0.209 \pm 0.004$ | $0.836 \pm 0.046$ | $\mathbf{0.343 \pm 0.010}$ | $0.243 \pm 0.103$ | $\mathbf{0.005 \pm 0.020}$ |
| Qwen2.5-1.5B-Inst | $0.317 \pm 0.020$ | $\mathbf{0.224 \pm 0.009}$ | $0.224 \pm 0.020$ | $\mathbf{0.163 \pm 0.009}$ | $0.851 \pm 0.026$ | $\mathbf{0.327 \pm 0.011}$ | $0.706 \pm 0.147$ | $\mathbf{0.023 \pm 0.023}$ |
| Qwen2.5-3B-Inst | $0.269 \pm 0.015$ | $\mathbf{0.265 \pm 0.009}$ | $0.186 \pm 0.015$ | $\mathbf{0.180 \pm 0.009}$ | $0.558 \pm 0.027$ | $\mathbf{0.349 \pm 0.017}$ | $0.234 \pm 0.056$ | $\mathbf{0.031 \pm 0.039}$ |
| Qwen2.5-7B-Inst | $0.285 \pm 0.006$ | $\mathbf{0.164 \pm 0.007}$ | $\mathbf{0.164 \pm 0.006}$ | $0.187 \pm 0.007$ | $0.521 \pm 0.009$ | $\mathbf{0.325 \pm 0.020}$ | $0.470 \pm 0.078$ | $\mathbf{0.063 \pm 0.045}$ |
| Qwen2.5-14B-Inst | $0.162 \pm 0.005$ | $\mathbf{0.099 \pm 0.003}$ | $\mathbf{0.146 \pm 0.005}$ | $0.148 \pm 0.003$ | $\mathbf{0.310 \pm 0.010}$ | $0.369 \pm 0.008$ | $\mathbf{0.039 \pm 0.015}$ | $0.455 \pm 0.009$ |
| Qwen2.5-32B-Inst | $\mathbf{0.116 \pm 0.001}$ | $0.129 \pm 0.003$ | $0.140 \pm 0.001$ | $\mathbf{0.133 \pm 0.003}$ | $0.580 \pm 0.013$ | $\mathbf{0.323 \pm 0.045}$ | $0.479 \pm 0.019$ | $\mathbf{0.186 \pm 0.103}$ |
| Qwen2.5-72B-Inst | $\mathbf{0.115 \pm 0.004}$ | $0.125 \pm 0.003$ | $0.138 \pm 0.004$ | $\mathbf{0.113 \pm 0.003}$ | $0.253 \pm 0.015$ | $\mathbf{0.221 \pm 0.023}$ | $\mathbf{0.032 \pm 0.028}$ | $0.068 \pm 0.052$ |
| GPT-4o | $\mathbf{0.123 \pm 0.004}$ | $0.125 \pm 0.004$ | $\mathbf{0.106 \pm 0.004}$ | $0.120 \pm 0.004$ | $0.455 \pm 0.014$ | $\mathbf{0.192 \pm 0.007}$ | $0.455 \pm 0.029$ | $\mathbf{0.004 \pm 0.014}$ |
| GPT-4o-mini | $0.263 \pm 0.005$ | $\mathbf{0.207 \pm 0.003}$ | $\mathbf{0.150 \pm 0.005}$ | $0.167 \pm 0.003$ | $0.513 \pm 0.017$ | $\mathbf{0.297 \pm 0.006}$ | $0.001 \pm 0.032$ | $\mathbf{0.000 \pm 0.010}$ |

Table 3: Results of models with IC-DP in various groups of tasks in CiK. The best-performing method with each model in every group is in **bold**.

with full-ROI), with significant improvements in ROI RCRPS and constraints RCRPS across many models, and minor improvements in non-ROI RCRPS. IC-DP surprisingly also improves the performance of the best model, Llama-405B-Inst, by 25% on average across all types of tasks, indicating that in-context examples can benefit even the largest models, unlike CorDP which primarily benefits smaller models. IC-DP also differs from CorDP in the kinds of tasks it provides most improvements on, indicating that CorDP and IC-DP can be complementary strategies, used according to the kind of task in the application. Qwen-2.5-14B remains an outlier as observed with CorDP, degrading 9% on average with IC-DP, with only minor improvements in non-ROI RCRPS. This is further evidence for its strong zero-shot forecasting capabilities with naïve direct prompting, where modifications may be detrimental. Forecasts of models with IC-DP with the respective in-context examples used are provided in Appendix E.3.

The general success of IC-DP validates that in-context examples can significantly enhance LLM forecasting capabilities. Our current evaluation uses a single example; future work could explore varying the number and similarity of examples to understand optimal in-context learning conditions. To handle contexts that are completely different from previously seen contexts, synthetic examples could help applications cover a broader range of real-world scenarios. One limitation is the increased input token count, which raises inference costs, especially for larger models. However, this trade-off could often be worthwhile given the substantial performance improvements across most models and task types.

# 7 RouteDP: Direct Prompt with Model Routing

## 7.1 Balancing Performance and Efficiency

Work in context-aided forecasting has shown that larger LLMs generally tend to perform better at context-aided forecasting tasks on average (Williams et al., 2025; Zhang et al., 2025; Kupferschmidt et al., 2024), which we also observe with the Direct Prompt method (see Appendix B). However, in many real-world applications where resources are limited, utilizing the largest LLMs such as Llama-405B-Inst (Grattafiori et al., 2024) for every task would be prohibitively expensive and often unnecessary, as smaller models may suffice for simpler tasks. Model routing strategies (Ong et al., 2024; Madras et al., 2018) aim to allocate tasks adaptively, sending only the most challenging cases to larger, more capable models, while routing easier tasks to smaller, more efficient ones. This is also useful in practice where given a set of tasks and a fixed compute budget, it is important to know when to spend more compute and when to use a much smaller model, while performing sufficiently well on average. To do so, we ask the question of whether LLMs can first assess the difficulty of a set of tasks, and then based on the ranking and the available compute, route tasks to a pre-determined large model, while using a small LLM for the easier tasks.

| Router Model | Percentage of tasks sent to large model | | | | | |
|---|---|---|---|---|---|---|
| | 0% | 20% | 40% | 60% | 80% | 100% |
| Qwen2.5-0.5B-Inst | 0.592 ± 0.027 | 0.316 ± 0.027 | 0.222 ± 0.005 | 0.206 ± 0.005 | 0.199 ± 0.004 | 0.173 ± 0.003 |
| Qwen2.5-1.5B-Inst | 0.592 ± 0.027 | 0.504 ± 0.009 | 0.449 ± 0.007 | 0.404 ± 0.004 | 0.407 ± 0.004 | 0.173 ± 0.003 |
| Qwen2.5-3B-Inst | 0.592 ± 0.027 | 0.507 ± 0.026 | 0.490 ± 0.026 | 0.393 ± 0.003 | 0.282 ± 0.003 | 0.173 ± 0.003 |
| Qwen2.5-7B-Inst | 0.592 ± 0.027 | 0.510 ± 0.010 | 0.437 ± 0.007 | 0.412 ± 0.004 | 0.181 ± 0.004 | 0.173 ± 0.003 |
| Qwen2.5-14B-Inst | 0.592 ± 0.027 | 0.581 ± 0.027 | 0.439 ± 0.027 | 0.324 ± 0.027 | 0.187 ± 0.004 | 0.173 ± 0.003 |
| Qwen2.5-32B-Inst | 0.592 ± 0.027 | 0.383 ± 0.010 | 0.368 ± 0.008 | 0.230 ± 0.006 | 0.196 ± 0.004 | 0.173 ± 0.003 |
| Qwen2.5-72B-Inst | 0.592 ± 0.027 | 0.509 ± 0.010 | 0.395 ± 0.009 | 0.287 ± 0.009 | 0.243 ± 0.009 | 0.173 ± 0.003 |

Table 4: Average RCRPS with Qwen2.5-0.5B-Inst as the main model, as a function of the percentage of tasks routed to the large model (Llama-405B-Inst), using different models as the router. Each row corresponds to a different router model, and each column to a routing budget. The results show that Qwen2.5-0.5B-Inst is the most effective router for itself, achieving the largest performance gains at low routing percentages, while other routers are less effective. The means are accompanied by standard errors. Results with other main models are in Appendix F.2.

## 7.2 Task Difficulty Ranking and Two-Model Routing

We explore a two-model routing setup: a small model (e.g., Qwen 0.5B) serves as the main model for most tasks, while the largest model (Llama-3.1-405B-Inst) is used as the large model for the most difficult tasks. A separate router model is prompted zero-shot to assign a "difficulty" score between 0 (easiest) and 1 (hardest) to each task, based on the task's context and history (prompt in App. Appendix F.1). For a given compute budget and $N$ context-aided forecasting tasks, the $k$ most difficult tasks as judged by the router are routed to the large model, while the remaining $N - k$ tasks are handled by the main model. We vary $k$ from 0 (all tasks to the small model) to 71 (all tasks to the large model), and measure the aggregate performance as a function of $k$. The goal of the router model is to identify the best task to send to the large model as more compute is available to be used. We call this approach RouteDP. As baselines, we compare to random routing (assigning $k$ random tasks to the large model, at each $k$) and ideal routing (assigning tasks in the order that most improves average RCRPS). We test the Qwen family of LLMs as the Router Model, and the same family of models as the Main Model. We stick to Llama-3.1-405B-Inst as our large model, as it has been observed to be disproportionally better at context-aided forecasting tasks compared to the small models. Additional details on the protocol are provided in Appendix F.

## 7.3 Results with RouteDP

We find that the RouteDP approach can meaningfully exploit differences between tasks to predict their difficulty, achieving significantly better performance than random routing. For example, with Qwen2.5-0.5B-Inst as the router, with it also being the main model, captures 66% of the total area between the random and ideal router as shown in Figure 3 (see App. Table 15 for area captured by all router models with all main models). Table 4 shows the performance of Qwen2.5-0.5B-Inst (the main model), as an increasing percentage of tasks are routed to the large model, with different models as the router model (Table 14 contains detailed results with all models). RouteDP serves as a simple yet effective approach to improve performance: routing less than 20% of tasks to the large model yields a sharp drop in average RCRPS, with a 46.6% improvement already, capturing a sizeable portion of the potential area of improvement between random

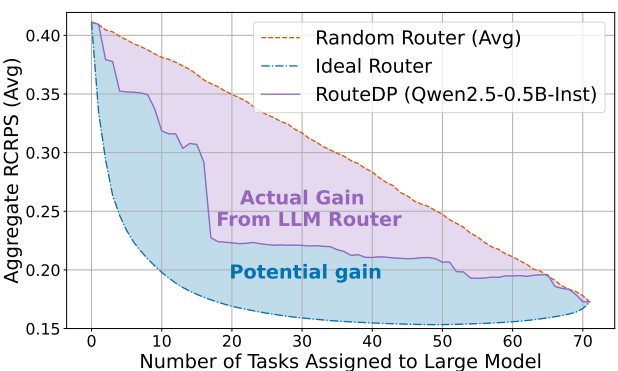

Figure 3: The plot shows the average RCRPS achieved using Qwen2.5-0.5B-Inst as the main model as an increasing percentage of tasks are routed to the large model (Llama-405B-Inst), using Qwen2.5-0.5B-Inst as the router model, compared to random and ideal routers. The router captures a significant 66% of the possible area between random and ideal routing. Area captured by other models are shown in Appendix F.3.

and ideal routing. Since most of the performance gain in RCRPS is achieved by routing only 20-40% of tasks, practitioners can achieve significant improvements in accuracy at a fraction of the cost, which is highly relevant for real-world deployment. We also observe diminishing returns as more tasks are routed to the large model. This holds across several routers and main models.

Next, we observe that the smallest main models benefit the most from routing (See Table 15). This is expected as the possible improvement with routing decreases with model size: for Qwen2.5-0.5B-Inst, there is a 71% reduction possible (using Llama-3.1-405B-Inst's performance as the ceiling), while for Qwen2.5-14B-Inst, the possible reduction is only 30%. This is reflected in the performance improvement obtained by routing - for e.g. by routing 20% of the tasks, Qwen2.5-0.5B-Inst as the main model achieves a 46.6% improvement while Qwen2.5-14B-Inst only achieves 16.6% with their respective best routers.

Finally, each model acts as its own best router: for example, as seen in Table 4, when Qwen2.5-0.5B-Inst is the main model, using Qwen2.5-0.5B-Inst as the router yields the best results across different compute budgets, a trend that holds for many other small models as well. Further, among the tested router models, several small models, particularly Qwen2.5-0.5B-Inst, stand out as disproportionately effective routers, sometimes even outperforming the main model itself when used as the router.

### 7.4 Implications of Routing Approaches

Our results reveal that by leveraging LLMs' ability to assess task difficulty, it is possible to achieve significantly better forecasting performance as our compute budget increases. The benefits of routing are not limited to theoretical efficiency: in practice, even small models such as Qwen2.5-0.5B-Inst as the router can capture the majority of the possible performance gains, which offers immediate practical benefits for real-world deployment. Future work should investigate how to better predict task difficulty, aiming to approach the ideal routing curve. One promising direction is to allow the router to explicitly select among a list of LLMs, potentially training the router for this purpose. Further, incorporating domain knowledge to provide more objective measures of difficulty could result in better performance. For example, tasks involving causal reasoning in the context or complex patterns in the history may be inherently more difficult. Finally, incorporating human-labeled data on task difficulty could also result in improved routing strategies.

## 8 Unifying the Proposed Strategies

While the proposed strategies each offer different benefits over naïve direct prompting (DP), a natural question is whether they are complementary to each other and can be integrated in a framework for practitioners seeking to deploy systems in the real-world. This can depend on several factors, such as available compute budget, inference time constraints, the need for interpretability, availability of historical context-aided forecasting tasks etc. We first find that the cost and inference time differ based on the method used (reported in Appendix H): IC-DP and CorDP both increase the number of input tokens consumed by the model due to longer prompts, and ReDP increases the number of output tokens generated as it involves eliciting reasoning traces. At the same time, this increase in inference time can allow for significantly improved forecasts, and hence the trade-off with inference time may be worthwhile. Beyond computational considerations, when better interpretability is required than what naïve direct prompting offers, the ReDP method shows that certain models can indeed output meaningful reasoning traces and allows for evaluating the intermediate reasoning of models, while CorDP shows that certain models can successfully condition on base forecasts and only modify them where necessary. When examples of prior context-aided forecasting tasks are available, the IC-DP method shows that conditioning on such examples further improves performance. Finally, to obtain better performance under compute constraints and to allow using more compute only for complex tasks, the RouteDP method allows one to delegate complex tasks to larger models. In theory, the nature of the proposed methods indeed allows practitioners to combine them and use them complementarily. We find that the methods can be complementary in practice as well, with simple experiments that demonstrate the complementarity of the methods (results in Appendix D.3 and Appendix F.4). This flexibility in combining strategies provides practitioners with a comprehensive toolkit for addressing diverse real-world deployment scenarios, making LLM-based context-aided forecasting more effective and interpretable across different operational scenarios.

# 9 Discussion and Future Work

This work builds on recent research on prompting large language models (LLMs) for zero-shot, context-aided time series forecasting. We introduce four complementary strategies to perform zero-shot context-aided forecasting performance of LLMs, namely Direct Prompting with Reasoning over Context (ReDP), Direct Prompting for Forecast Correction (CorDP), In-Context Direct Prompting (IC-DP), and Direct Prompting with Model Routing (RouteDP) which offer different benefits over naïve direct prompting (DP), improving interpretability, performance and resource efficiency.

Our work opens up several directions of research, highlighting the room for potential improvements in context-aided forecasting methods without any training. One active field of research to bring the power of LLMs to time series forecasting is to use the LLM either as an orchestrator Ye et al. (2025b) or an agent Garza & Rosillo (2025) that can decide which forecasting technique is appropriate based on the available information or adjust the forecast produced by said technique. Therefore, it would be worthwhile to investigate whether these frameworks would see similar results as RouteDP and CorDP. Additionally, since DP could be computationally expensive compared to more traditional time series techniques due to it being LLM-based, determining whether orchestrators or agents know when it is worth its cost would make it much more appealing. Next, while we limit our scope to studying these strategies with the DP method, it is also interesting to study them in the context of other methods such as LLMP (Requeima et al., 2024) and LLMTime (Gruver et al., 2024). Next, exploring the usefulness of these strategies in more unconstrained setups where for e.g. all context may not be relevant, or where the context is long, are also interesting directions, however first requires the development of datasets where the contexts have the respective properties to test for. Our scope is also limited to improving the *zero-shot* performance of LLMs; exploring these strategies in the other paradigm of training-based methods could be useful. In particular, moving to training-based methods can also broaden the scope of these strategies e.g. training the router in RouteDP or training a model to use any base forecaster in CorDP. Finally, while these strategies improve the zero-shot performance of LLMs, the high cost of LLMs compared to canonical forecasting methods still limit the applicability of LLMs in deployment; these strategies and studies must in-turn be used to develop more efficient models from the ground-up, keeping the requirements of the respective forecasting application in mind.

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

# Appendix

## Table of Contents

# A   Additional Details on the Context-Is-Key (CiK) benchmark

## A.1   The RCRPS metric

We use the Region-of-Interest CRPS (RCRPS) metric to evaluate context-aided forecasting performance (Williams et al., 2025), which modifies the CRPS metric (Gneiting & Raftery, 2007) to prioritize context-sensitive windows and accounts for constraint satisfaction. Given an inferred forecast distribution $\widetilde{\mathbf{X}}_F$ and a ground truth $\mathbf{x}_F$, the RCRPS metric is defined as:

$$\mathrm{RCRPS}(\widetilde{\mathbf{X}}_F, \mathbf{X}_F) \;=\; \alpha \cdot \left[ \frac{1}{2|\mathcal{I}|} \sum_{i \in \mathcal{I}} \mathrm{CRPS}(\widetilde{X}_i, x_i) + \frac{1}{2|\neg\mathcal{I}|} \sum_{i \in \neg\mathcal{I}} \mathrm{CRPS}(\widetilde{X}_i, x_i) \;+\; \beta \cdot \mathrm{CRPS}(v_{\mathbf{C}}(\widetilde{\mathbf{X}}_F), 0) \right],$$

where the terms respectively account for the CRPS inside the RoI, the CRPS outside of the RoI, and the constraint violation penalty. The $\alpha$ term is a task-dependent normalization factor to make the RCRPS scale-independent, while $\beta$ is a scaling factor that controls the impact of constraint violation on the score; we use $\beta = 10$ in our experiments as used in Williams et al. (2025).

# B   Additional Results with the Direct Prompt (DP) Method

## B.1   Aggregate Results of models

Results of various models with Direct Prompt (DP), with and without context are given in Table 5.

| Model | Without Context | With Context |
|---|---|---|
| Qwen2.5-0.5B-Inst | **0.404 ± 0.028** | 0.592 ± 0.027 |
| Qwen2.5-1.5B-Inst | 0.631 ± 0.039 | **0.616 ± 0.018** |
| Qwen2.5-3B-Inst | 0.513 ± 0.039 | **0.424 ± 0.017** |
| Qwen2.5-7B-Inst | 0.610 ± 0.011 | **0.401 ± 0.006** |
| Qwen2.5-14B-Inst | 0.551 ± 0.007 | **0.247 ± 0.006** |
| Qwen2.5-32B-Inst | 0.607 ± 0.008 | **0.397 ± 0.008** |
| Qwen2.5-72B-Inst | 0.549 ± 0.009 | **0.202 ± 0.009** |
| Llama3.2-1B-Inst | 0.481 ± 0.028 | **0.396 ± 0.027** |
| Llama3.2-3B-Inst | 0.950 ± 0.041 | **0.687 ± 0.025** |
| Llama3-8B-Inst | 0.758 ± 0.009 | **0.543 ± 0.026** |
| Llama3.3-70B-Inst | 0.700 ± 0.009 | **0.230 ± 0.006** |
| Llama3.1-405B-Inst | 0.686 ± 0.011 | **0.173 ± 0.003** |
| GPT-4o | 0.665 ± 0.004 | **0.317 ± 0.009** |
| GPT-4o-mini | 0.648 ± 0.012 | **0.389 ± 0.010** |
| Chronos-Large | 0.492 ± 0.004 | - |
| Lag-Llama | 0.382 ± 0.011 | - |
| Arima | 0.636 ± 0.014 | - |

Table 5: Aggregate Results (RCRPS) of models on the CiK benchmark.

## B.2   Results of models on various kinds of tasks

Results of various models with Direct Prompt (DP), with and without context, partitioned by different kinds of tasks are given in Table 6.

| | ROI | | non-ROI | | Full ROI | | Constraints | |
|---|---|---|---|---|---|---|---|---|
| Model | Without Context | With Context | Without Context | With Context | Without Context | With Context | Without Context | With Context |
| Llama3.2-1B-Inst | 0.357 ± 0.018 | **0.336 ± 0.026** | **0.236 ± 0.018** | 0.248 ± 0.026 | 0.607 ± 0.045 | **0.467 ± 0.041** | 0.604 ± 0.064 | **0.275 ± 0.092** |
| Llama3.2-3B-Inst | 0.832 ± 0.118 | **0.281 ± 0.013** | 0.769 ± 0.030 | **0.162 ± 0.013** | 1.022 ± 0.048 | **1.004 ± 0.040** | **0.613 ± 0.075** | 1.030 ± 0.090 |
| Llama3-8B-Inst | 0.336 ± 0.017 | **0.255 ± 0.008** | 0.239 ± 0.017 | **0.163 ± 0.008** | 1.078 ± 0.009 | **0.771 ± 0.043** | 0.460 ± 0.199 | **0.169 ± 0.172** |
| Qwen2.5-1.5B-Inst | 0.327 ± 0.009 | **0.317 ± 0.020** | **0.142 ± 0.009** | 0.224 ± 0.020 | 0.900 ± 0.065 | **0.851 ± 0.026** | **0.379 ± 0.242** | 0.706 ± 0.147 |
| Qwen2.5-7B-Inst | 0.520 ± 0.008 | **0.285 ± 0.006** | **0.157 ± 0.008** | 0.164 ± 0.006 | 0.794 ± 0.018 | **0.521 ± 0.009** | 0.476 ± 0.041 | **0.470 ± 0.078** |
| Qwen2.5-14B-Inst | 0.376 ± 0.008 | **0.162 ± 0.005** | 0.155 ± 0.008 | **0.146 ± 0.005** | 0.745 ± 0.010 | **0.310 ± 0.010** | 0.473 ± 0.019 | **0.039 ± 0.015** |
| Qwen2.5-32B-Inst | 0.537 ± 0.003 | **0.116 ± 0.001** | 0.152 ± 0.003 | **0.140 ± 0.001** | 0.786 ± 0.014 | **0.580 ± 0.013** | 0.503 ± 0.031 | **0.479 ± 0.019** |
| Llama3.3-70B-Inst | 0.531 ± 0.010 | **0.105 ± 0.003** | **0.147 ± 0.010** | 0.182 ± 0.003 | 0.945 ± 0.014 | **0.289 ± 0.011** | 0.475 ± 0.031 | **0.000 ± 0.024** |
| Llama3.1-405B-Inst | 0.537 ± 0.002 | **0.126 ± 0.004** | **0.147 ± 0.002** | 0.150 ± 0.004 | 0.920 ± 0.019 | **0.196 ± 0.005** | 0.478 ± 0.038 | **0.004 ± 0.009** |
| Qwen2.5-3B-Inst | 0.280 ± 0.006 | **0.269 ± 0.015** | **0.155 ± 0.006** | 0.186 ± 0.015 | 0.713 ± 0.065 | **0.558 ± 0.027** | **0.087 ± 0.147** | 0.234 ± 0.056 |
| Qwen2.5-72B-Inst | 0.530 ± 0.001 | **0.115 ± 0.004** | 0.141 ± 0.001 | **0.138 ± 0.004** | 0.695 ± 0.015 | **0.253 ± 0.015** | 0.513 ± 0.034 | **0.032 ± 0.028** |
| Qwen2.5-0.5B-Inst | **0.249 ± 0.005** | 0.339 ± 0.010 | 0.149 ± 0.005 | **0.129 ± 0.010** | **0.544 ± 0.046** | 0.836 ± 0.046 | 0.557 ± 0.104 | **0.243 ± 0.103** |
| GPT-4o | 0.524 ± 0.003 | **0.123 ± 0.004** | 0.148 ± 0.003 | **0.106 ± 0.004** | 0.888 ± 0.006 | **0.455 ± 0.014** | 0.477 ± 0.009 | **0.455 ± 0.029** |
| GPT-4o-mini | 0.495 ± 0.008 | **0.263 ± 0.005** | **0.102 ± 0.008** | 0.150 ± 0.005 | 0.885 ± 0.019 | **0.513 ± 0.017** | 0.461 ± 0.042 | **0.001 ± 0.032** |
| Chronos-Large | 0.536 ± 0.003 | | 0.115 ± 0.003 | | 0.605 ± 0.006 | | 0.487 ± 0.010 | |
| Lag-Llama | 0.224 ± 0.005 | | 0.202 ± 0.005 | | 0.497 ± 0.018 | | 0.204 ± 0.037 | |
| Arima | 0.272 ± 0.004 | | 0.159 ± 0.004 | | 0.921 ± 0.023 | | 0.843 ± 0.050 | |

Table 6: Aggregate Results (RCRPS) of models on various groups of tasks from the CiK benchmark.

# C   Additional Details on ReDP

## C.1   ReDP Prompt

We use the following prompt for the ReDP method, where **{history}** is replaced by the respective numerical history for the task instance in the format (timestamp, value), **{context}** is replaced by the respective textual context for the task instance, and ((**pred_time**)) is replaced with the prediction timesteps.

```
I have a time series forecasting task for you.

Here is some context about the task. Make sure to factor in any background knowledge,
satisfy any constraints, and respect any scenarios.
<context>
{context}
</context>

Here is a historical time series in (timestamp, value) format:
<history>
{history}
</history>

You are tasked with predicting the value at the following timestamps: {pred_time}.

First, within <reason> and </reason> tags, walk-through step-by-step how you would incorporate
    each piece of the context to improve your forecast. If you think any of the context is
    irrelevant, please indicate.

Next, return your forecast in (timestamp, value) format in between <forecast> and </forecast>
    tags.
Do not include any other information (e.g., comments) in the forecast.
```

One could use constrained decoding tools such as lm-format-enforcer (Gat & contributors, 2024) and XGrammar (Dong et al.) to constrain the output format, however we found that using using constrained decoding with free-form text (between the <reasoning> and </reasoning>) was very slow, taking several hours for a single instance and at times not completing. Therefore, we do not use any constrained decoding and instead retry 15 times if a model fails to output in the specified format.

## C.2 Reasoning Quality Analysis - Protocol Details

**Tasks considered for the analysis.**  For the analysis, we consider the 20 context-aided forecasting tasks from the CiK benchmark (Williams et al., 2025) that have a region-of-interest (RoI) indicated. We consider only these tasks from the benchmark, due to the following reasons.

(i) **There exists only a single ground truth reasoning for these tasks.** For e.g. in the ElectricityIncreaseInPredictionTask, the only correct reasoning from the context would be to multiply the usual consumption for each hour affected by the heat wave, by the amount specified. In the other tasks where the full prediction window is the region of interest and the context does not specify a targeted region in the prediction window, several possible deductions can be made from the context that can help produce better forecasts (Williams et al., 2025). Thereby, obtaining a single gold-standard reasoning trace that covers all these deductions is more difficult than in the former case. Evaluating a model's reasoning trace with such a gold-standard is also difficult, as the model's reasoning may partially be correct, which complicates evaluation.

(ii) **Measuring meaningful improvement with context is straight-forward in the case where there is a region of interest**. This is because originally, these tasks were created such that obtaining high accuracy in the region of interest is impossible without the context (as the data in these regions are modified appropriately, according to the context). Thereby, any improvement in this region of interest with context would mean that the model applied its reasoning correctly to obtain better performance: we conclude with an empirical analyses any improvement of 50% in this region as an improvement with context. On the contrary, in tasks where the entire prediction region is the region of interest which are more difficult tasks than the former (Williams et al., 2025), while models can improve their forecast with context, we found that the amount of percentage improvement required to conclude that the model has Due to these complications, we leave out these tasks from the analysis.

**Obtaining gold-standard reasoning traces.**  For the reasoning quality analysis, we use the following prompt to obtain the gold-standard reasoning for the tasks considered in the analysis,

```
You are a forecasting expert. Given the following information:

CONTEXT:
{context}

Please provide a concise reasoning trace (one sentence) that explains how someone could
    logically produce a forecast based on the context.

Format your response as:
<reason>
[Your detailed reasoning here]
</reason>
```

We use GPT-4.1 to generate the gold-standard reasoning for all tasks. We manually verify the gold-standard reasoning traces and modify them if required.

**Reasoning quality evaluation.**  Then, to compare a model's reasoning trace produced with ReDP Appendix C.1 (**{model_reasoning}** in the below prompt), with the gold-standard reasoning trace (**{ground_truth_reasoning}** in the below prompt), we use the following prompt, again with GPT4.1.

```
Compare these two reasoning traces for a forecasting task:

Model Reasoning:
{model_reasoning}

Ground Truth Reasoning:
{ground_truth_reasoning}

Question: Is the model reasoning aligned with the key points mentioned in ground truth
    reasoning approach?

Answer with exactly one word: YES or NO

<answer>YES/NO</answer>
```

We found that for the tasks considered in the analysis, the reasoning traces produced by a model were similar or many in cases exactly the same. Thereby, we only consider the reasoning trace corresponding to the first sample for the quality evaluation, for simplicity and to save costs (as the comparison needs to be done only once for a task instance). We will however release all the reasoning traces corresponding to all samples of a task, produced by a model. Extensions of the analysis methodology could look into comparing multiple reasoning traces using advancements in LLM-as-a-judge methodologies (Fu et al., 2023; Lightman et al., 2023; Gu et al., 2024).

### C.3  Performance of ReDP models

We plot in figures Figure 4 and Figure 5 the performance of ReDP models in the tasks considered for the analysis, compared to their respective performances using naive DP. We plot violin plots for each model, as different model fails in different tasks, precluding us from producing aggregate results on all tasks. Overall, we find that in many tasks, ReDP can slightly improve performance, but not significantly. In some tasks, ReDP even degrades performance. As the focus of the work is on evaluating the reasoning quality of the models through the ReDP method, we do not focus on the performance improvements or deteriorations that ReDP brings over DP.

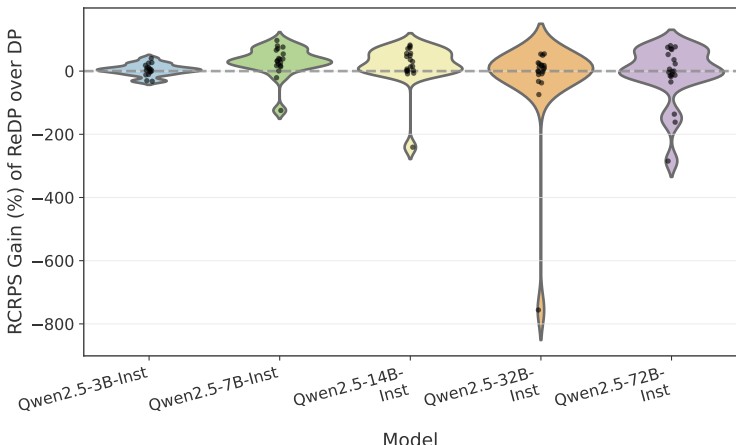

Figure 4: Distribution of improvement in RCRPS with ReDP over DP with Qwen models on the tasks considered for the analysis.

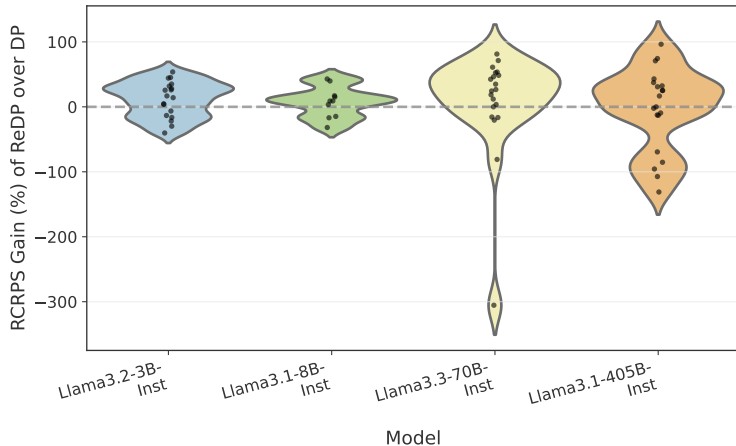

Figure 5: Distribution of improvement in RCRPS with ReDP over DP with Llama models on the tasks considered for the analysis.

### C.4 Gold-standard reasoning traces

#### C.4.1 Verification

Once the ground truth reasoning traces are generated, for each task, we verify the following with a human evaluation panel consisting of the authors of this paper:

- If the ground truth reasoning trace captured the key change in forecast that the context brings about.

- If the dates and duration of the context-induced change is accurate in the ground truth reasoning trace.

- The ground truth reasoning trace is crisp, and only contains the direct impact of the context

E.g. For the ATMBuildingClosedTask with context that says

"Background: This is the number of cash withdrawals from an automated teller machine (ATM) in an arbitrary location in England. Constraints: None. Scenario: Consider that the building which contains the ATM is closed from 1996-11-24 00:00:00, for 10 days.",

We verified that the reasoning trace indicates that

- There will be a period in the forecast where the number of withdrawals would be zero

- The said period would be from 1996-11-24 00:00:00 for 10 days

- The reasoning trace does not mention anything else other than the two points above

As per the above verification method, we found that the ground truth reasoning traces met the above criteria for all tasks except 2 namely the SensorMaintenanceInPredictionTask (where criterion 1 was not met) and the DecreaseInTrafficInPredictionTask (where criterion 2 was not met). Hence, we adapted them to meet the criteria:

For **SensorMaintenanceInPredictionTask**:

**Context**: Background: This series represents the occupancy rate (%) captured by a highway sensor. Constraints: None. Scenario: Consider that the meter will be offline for maintenance between 2024-01-18 08:00:00 and 2024-01-18 14:00:00, which results in zero readings.

**Original reasoning trace**: A logical forecast would interpolate or exclude the zero readings during the offline maintenance period and use historical occupancy patterns from similar time windows on previous days to estimate the expected occupancy rate.

> This does not meet criterion 1 as it is the wrong interpretation of the context, as the mentioned dates are in the prediction horizon and not in the historical data.

**Modified reasoning trace:** The context mentions that the meter will be offline for maintenance between 2024-01-18 08:00:00 and 2024-01-18 14:00:00. During this period, one should forecast a value of 0%, as the meter will not be capturing any data.

For **DecreaseInTrafficInPredictionTask**:

**Context**: Background: This is hourly traffic data. Constraints: None. Scenario: Suppose that there is an accident on the road and there is 20.0% of the usual traffic from 2024-01-18 06:00:00 for 5 hours.

**Original reasoning trace**: To forecast traffic during the accident, multiply the usual hourly traffic by 0.2 for each hour from 2024-01-18 06:00:00 to 2024-01-18 10:00:00, since the accident reduces traffic to 20% of normal levels for 5 hours.

> This does not meet criterion 2 as the end time is incorrectly interpreted as 10:00:00 while it is supposed to be 11:00:00.

**Modified reasoning trace**: To forecast traffic during the accident, multiply the usual hourly traffic by 0.2 for each hour from 2024-01-18 06:00:00 to 2024-01-18 11:00:00, since the accident reduces traffic to 20% of normal levels for 5 hours.

### C.4.2 Per-task gold-standard reasoning traces

We release in Table 7 the gold-standard reasoning traces for the tasks considered for the analysis.

Table 7: Gold Standard Reasoning for each task considered for the reasoning quality analysis.

| Task Name | Context | Ground Truth Reasoning |
|---|---|---|
| ATMBuildingClosedTask | Background: This is the number of cash withdrawals from an automated teller machine (ATM) in an arbitrary location in England. Constraints: None. Scenario: Consider that the building which contains the ATM is closed from 1996-11-24 00:00:00, for 10 days. | Since the building housing the ATM is closed for 10 days starting 1996-11-24, a logical forecast would set the number of cash withdrawals to zero for that period, as no one can access the ATM during the closure. |
| ATMUnderPeriodicMaintenanceTaskWithConclusion | Background: This is the number of cash withdrawals from an automated teller machine (ATM) in an arbitrary location in England. The ATM was under maintenance for 7 days, periodically every 15 days, starting from 1996-08-12 00:00:00, resulting in no withdrawals recorded. Assume that the ATM will not be in maintenance in the future. Constraints: None. Scenario: None. | A logical forecast can be produced by analyzing the historical withdrawal patterns while excluding data from the maintenance periods, then extrapolating the underlying trend and seasonality to predict future withdrawals, given that maintenance interruptions will no longer occur. |

| Task Name | Context | Ground Truth Reasoning |
|---|---|---|
| ATMUnderPeriodicMaintenanceTaskWithConclusionLessExplicit | Background: This is the number of cash withdrawals from an automated teller machine (ATM) in an arbitrary location in England. The ATM was under maintenance for various periods, resulting in no withdrawals recorded. Assume that the ATM will not be in maintenance in the future. Constraints: None. Scenario: None. | A logical forecast can be produced by identifying and removing periods of maintenance (with zero withdrawals) from the historical data to estimate the typical withdrawal rate during operational periods, then projecting this rate forward under the assumption that the ATM will remain operational. |
| ATMUnderPeriodicMaintenanceTaskWithoutConclusion | Background: This is the number of cash withdrawals from an automated teller machine (ATM) in an arbitrary location in England. The ATM was under maintenance for 7 days, periodically every 15 days, starting from 1996-08-12 00:00:00. Assume that the ATM will not be in maintenance in the future. Constraints: None. Scenario: None. | One could logically forecast future cash withdrawals by identifying and removing the recurring 7-day drops in activity caused by scheduled maintenance every 15 days, then modeling the underlying demand trend using the adjusted data to predict future withdrawals now that maintenance will no longer occur. |
| CashDepletedinATMScenarioTask | Background: This is the number of cash withdrawals from an automated teller machine (ATM) in an arbitrary location in England. Constraints: None. Scenario: Consider that cash is depleted in the ATM from 1996-11-24 00:00:00, for 10 days, resulting in no withdrawals during that period. | A forecaster could logically produce a forecast by identifying the 10-day period of zero withdrawals as an anomaly due to cash depletion, then modeling expected withdrawal counts for other periods based on historical data while treating the anomaly as missing or censored data rather than as indicative of typical demand. |
| DecreaseInTrafficInPredictionTask | Background: This is hourly traffic data. Constraints: None. Scenario: Suppose that there is an accident on the road and there is 20.0% of the usual traffic from 2024-01-18 06:00:00 for 5 hours. | To forecast traffic during the accident, multiply the usual hourly traffic by 0.2 for each hour from 2024-01-18 06:00:00 to 2024-01-18 11:00:00, since the accident reduces traffic to 20% of normal levels for 5 hours. |
| ElectricityIncreaseInPredictionTask | Background: This is the electricity consumption recorded in Kilowatt (kW) in city A. Constraints: None. Scenario: Suppose that there is a heat wave in city A from 2013-05-28 12:00:00 for 2 hours in city A, leading to excessive use of air conditioning, and 4 times the usual electricity being consumed. | To forecast electricity consumption during the heat wave, multiply the usual consumption for each hour between 2013-05-28 12:00:00 and 2013-05-28 14:00:00 by 4, since the scenario specifies consumption is quadrupled due to excessive air conditioning use. |

| Task Name | Context | Ground Truth Reasoning |
|---|---|---|
| ElectricityIncreaseInPredictionWithDistractorText | Background: This is the electricity consumption recorded in Kilowatt (kW) in city A. Constraints: None. Scenario: Suppose that there is a heat wave in city A from 2013-05-28 12:00:00 for 2 hours, leading to excessive use of air conditioning, and 4 times the usual electricity being consumed. A brief technical issue in the electricity grid caused a major dip of 75% in electricity consumption 2 weeks ago. This issue is not expected to happen again this week. | To forecast electricity consumption for the upcoming week, one should adjust the baseline usage by excluding the anomalous dip from two weeks ago, and account for a fourfold increase during the 2-hour heat wave period due to increased air conditioning demand. |
| ElectricityIncreaseInPredictionWithDistractorWithDates | Background: This is the electricity consumption recorded in Kilowatt (kW) in city A. Constraints: None. Scenario: There was a festival in neighbouring cities B and C that resulted in 10 times the usual electricity being consumed there from 2013-05-28 12:00:00 for 2 hours. But this did not affect electricity consumption in city A. Suppose that there is a heat wave in city A from 2013-05-28 12:00:00 for 2 hours, leading to excessive use of air conditioning, and 4 times the usual electricity being consumed. | To forecast electricity consumption in city A during the heat wave, one would identify the typical consumption for the relevant 2-hour period and multiply it by 4, since the scenario specifies that usage increases fourfold due to excessive air conditioning. |
| ElectricityIncreaseInPredictionWithSplitContext | Background: This is the electricity consumption recorded in Kilowatt (kW) in city A. Constraints: None. Scenario: Suppose that there is a heat wave in city A from 2013-05-28 12:00:00 for 2 hours, which would typically lead to excessive use of air conditioning, and 10 times the usual electricity being consumed. But in this case, residents sought to conserve energy and used lesser air conditioning, resulting in excessive usage of only 4 times the usual electricity. | A forecaster could estimate electricity consumption during the heat wave by identifying the usual consumption for the affected hours and multiplying it by 4, reflecting the adjusted behavior of residents who used less air conditioning than typical during such events. |
| ExplicitTrafficForecastTaskwithHolidaysInPredictionWindow | Background: This series contains the road occupancy rates on a freeway in the San Francisco Bay area. Note that 2024-07-04 is a holiday due to Independence Day. Note that traffic on this freeway typically reduces on holidays. Constraints: None. Scenario: None. | Given that July 4th is a holiday (Independence Day) and historical patterns show reduced freeway traffic on holidays, one could logically forecast a lower road occupancy rate for 2024-07-04 compared to typical weekdays by referencing past holiday data and general traffic trends. |

| Task Name | Context | Ground Truth Reasoning |
|---|---|---|
| ExplicitWithDatesAndDaysTrafficForecastTaskwithHolidaysInPredictionWindow | Background: This series contains the road occupancy rates on a freeway in the San Francisco Bay area. The days for which the forecast is required are Thursday 2024-07-04, Friday 2024-07-05, Saturday 2024-07-06. Note that 2024-07-04 is a holiday due to Independence Day. Note that traffic on this freeway typically reduces on holidays. Constraints: None. Scenario: None. | A logical forecast can be produced by analyzing historical occupancy rates for the same weekdays and dates, especially focusing on past Independence Days and adjacent days, to account for typical holiday traffic reductions and altered travel patterns. |
| ExplicitWithDaysTrafficForecastTaskwithHolidaysInPredictionWindow | Background: This series contains the road occupancy rates on a freeway in the San Francisco Bay area. The days for which the forecast is required are Thursday, Friday, Saturday. Note that 2024-07-04 is a holiday due to Independence Day. Note that traffic on this freeway typically reduces on holidays. Constraints: None. Scenario: None. | To forecast occupancy rates for Thursday, Friday, and Saturday, one could analyze historical occupancy patterns for the same days of the week, adjust for the expected reduction on Thursday due to the Independence Day holiday, and consider potential spillover effects on Friday and Saturday, as holiday travel and reduced commuter activity may influence these days as well. |
| ImplicitTrafficForecastTaskwithHolidaysInPredictionWindow | Background: This series contains the road occupancy rates on a freeway in the San Francisco Bay area. Note that traffic on this freeway typically reduces on holidays. Constraints: None. Scenario: None. | Given that road occupancy rates decrease on holidays, a logical forecast can be produced by identifying upcoming holidays in the calendar and adjusting expected occupancy rates downward for those dates relative to typical non-holiday levels. |
| IncreasedWithdrawalScenario | Background: This is the number of cash withdrawals from an automated teller machine (ATM) in an arbitrary location in England. Constraints: None. Scenario: Suppose that there is a carnival from 1996-11-22 00:00:00, for 11 days leading to more people in the area, and 4 times the number of usual withdrawals during that period. | To forecast cash withdrawals during the carnival, multiply the usual daily withdrawal count by 4 for each of the 11 carnival days, while keeping the forecast for other days unchanged, as the scenario specifies a fourfold increase only during the event. |

| Task Name | Context | Ground Truth Reasoning |
|---|---|---|
| LongNewsElectricityIncreaseInPredictionTask | Background: This is the electricity consumption recorded in Kilowatt (kW) in city A. Constraints: None. Scenario: A sudden and intense heatwave struck the city, causing a dramatic surge in electricity consumption as residents sought refuge from the scorching temperatures. The extreme weather event, which began on 2013-05-28 12:00:00 and lasted for approximately 2 hours, saw temperatures soar to unprecedented levels. In response, citizens across the metropolitan area turned to their air conditioning units en masse, leading to a significant strain on the local power grid. According to the city's electricity provider, power consumption during the peak of the heatwave reached approximately 4 times the typical usage for this time of year. "We've never seen anything quite like this," said Jane Smith, spokesperson for PowerCity Utilities. "The sudden spike in demand pushed our systems to their limits." As the city recovers from this unprecedented power surge, experts are already discussing long-term solutions to manage similar situations in the future. These may include upgrades to the power grid, incentives for energy-efficient appliances, and the development of more robust emergency response protocols. For now, citizens are encouraged to stay hydrated, check on vulnerable neighbors, and use air conditioning responsibly as the community works together to beat the heat. | By analyzing the correlation between extreme heat events and surges in electricity consumption—specifically, noting that consumption during the heatwave was approximately four times the typical level for this period—one can forecast that future similar heatwaves are likely to cause comparable spikes in demand, unless mitigating measures are implemented. |

| Task Name | Context | Ground Truth Reasoning |
|---|---|---|
| MediumNewsElectricityIncreaseInPredictionTask | Background: This is the electricity consumption recorded in Kilowatt (kW) in city A. Constraints: None. Scenario: A sudden and intense heatwave struck the city, causing a dramatic surge in electricity consumption as residents sought refuge from the scorching temperatures. The extreme weather event, which began on 2013-05-28 12:00:00 and lasted for approximately 2 hours, saw temperatures soar to unprecedented levels. In response, citizens across the metropolitan area turned to their air conditioning units en masse, leading to a significant strain on the local power grid. According to the city's electricity provider, power consumption during the peak of the heatwave reached approximately 4 times the typical usage for this time of year. For now, citizens are encouraged to stay hydrated, check on vulnerable neighbors, and use air conditioning responsibly as the community works together to beat the heat. | By analyzing historical electricity consumption data for this time of year and multiplying the typical usage by four (as reported during the heatwave), one can estimate the likely electricity demand during similar future extreme heat events. |
| SensorMaintenanceInPredictionTask | Background: This series represents the occupancy rate (%) captured by a highway sensor. Constraints: None. Scenario: Consider that the meter will be offline for maintenance between 2024-01-18 08:00:00 and 2024-01-18 14:00:00, which results in zero readings. | The context mentions that the meter will be offline for maintenance between 2024-01-18 08:00:00 and 2024-01-18 14:00:00. During this period, one should forecast a value of 0%, as the meter will not be capturing any data. |
| SensorPeriodicMaintenanceTask | Background: This series represents the occupancy rate (%) captured by a highway sensor. The sensor was offline for maintenance every day between 08:00 and 14:00, which resulted in zero readings. Assume that the sensor will not be in maintenance in the future. Constraints: None. Scenario: None. | To produce a forecast, one could first remove the zero readings caused by maintenance periods to avoid bias, and then model the underlying occupancy rate pattern using the cleaned data, assuming future data will not be affected by maintenance downtime. |
| ShortNewsElectricityIncreaseInPredictionTask | Background: This is the electricity consumption recorded in Kilowatt (kW) in city A. Constraints: None. Scenario: A heatwave struck the city, which began on 2013-05-28 12:00:00 and lasted for approximately 2 hours, saw temperatures soar to unprecedented levels. According to the city's electricity provider, power consumption during the peak of the heatwave reached approximately 4 times the typical usage for this time of year. | By identifying the typical electricity consumption for late May at noon, multiplying this value by four to account for the reported heatwave surge, and applying this adjustment to the relevant 2-hour window, one can forecast the likely consumption during the heatwave event. |

### C.5 Examples of Correct and Incorrect Reasoning Traces

### C.5.1 Task: *ElectricityIncreaseInPredictionTask*

Background: This series contains the amount of sunlight (in Watts per squared meter) arriving on a horizontal surface, for a location in Florida, United States. Over the previous 90 days, the maximum sunlight happened on average at 12:25:33.
Constraints: None
Scenario: None

Figure 6: Context

To forecast electricity consumption during the heat wave, multiply the usual consumption for each hour between 2013-05-28 12:00:00 and 2013-05-28 14:00:00 by 4, since the scenario specifies consumption is quadrupled due to excessive air conditioning use.

Figure 7: Gold Standard Reasoning Trace

To improve the forecast, we need to incorporate the given context. The scenario mentions a heat wave in city A from 2013-05-28 12:00:00 for 2 hours, leading to excessive use of air conditioning and 4 times the usual electricity being consumed.
First, we need to identify the usual electricity consumption pattern. From the historical data, we can observe the daily patterns and weekly patterns. However, we don't have any information about the seasonality or yearly patterns.
The heat wave is the most critical context for this forecasting task. We should consider that the electricity consumption will be 4 times the usual consumption during the heat wave (from 2013-05-28 12:00:00 to 2013-05-28 14:00:00).
For the timestamps before the heat wave (2013-05-28 04:00:00 to 2013-05-28 11:00:00), we can use the historical data to identify the daily pattern and forecast the values.
For the timestamps during the heat wave (2013-05-28 12:00:00 to 2013-05-28 14:00:00), we will multiply the forecasted values by 4.
For the timestamps after the heat wave (2013-05-28 15:00:00 onwards), we will assume that the electricity consumption returns to normal and use the historical data to forecast the values.
By incorporating the context, we can improve the accuracy of the forecast.

Figure 8: A Correct Reasoning Trace (Model: Llama3.3-70B-Inst)

First, I would incorporate the historical time series data into my model. I would split the data into training and testing sets, with the training set consisting of data from 2013-05-22 04:00:00 to 2013-05-27 03:00:00, and the testing set consisting of data from 2013-05-27 04:00:00 to 2013-05-27 23:00:00 and 2013-05-28 12:00:00 (since it's the start of the heatwave).

Next, I would incorporate the scenario that there is a heat wave in city A from 2013-05-28 12:00:00 for 2 hours, leading to excessive use of air conditioning, and 4 times the usual electricity being consumed. To incorporate this, I would analyze the historical data and identify patterns in electricity consumption that are likely to be affected by heatwaves, such as increased consumption during the peak sun hours (usually between 11am and 3pm). I would then adjust the model to account for these increased consumption patterns during the heatwave.

I would also consider the relevance of each piece of the context. In this case, I consider the historical data, the heatwave scenario, and the timestamps for which I need to make predictions as relevant.

Here's how I would implement the model:

I would use a time series forecasting model that is capable of handling seasonal and trend components, such as an ARIMA or a seasonal decomposition model. I would then use the historical data to train the model and make predictions for the future timestamps.

However, since the heatwave scenario changes the behavior of the system, I would need to adjust the predictions accordingly. I would do this by identifying the periods affected by the heatwave and adjusting the predicted values to reflect the increased consumption due to the heatwave.

Finally, I would evaluate the performance of the model on the testing data and refine the model as needed.

In terms of which of the context is irrelevant, I would say that the specific background information about city A is not directly relevant to the task at hand, although it could be useful for additional analysis.

Figure 9: A Correct Reasoning Trace (Model: Llama3.1-8B-Inst)

Given the provided historical data and the context of a heat wave causing a significant increase in electricity consumption, I will make adjustments to the forecast based on the sudden surge in demand during the heat wave period. Here's the step-by-step reasoning:

1. **Identify the Heat Wave Period**: From the history, identify the period around 2013-05-28 12:00:00 when electricity consumption spiked to 555.84 kW. This likely represents the start of the heat wave impact.

2. **Analyze the Trend Before and After the Heat Wave**: Analyze the trend before and after the identified heat wave period to understand the normal variation and any potential anomalies or patterns during the heat wave.

- **Before Heat Wave** (e.g., 2013-05-22 to 2013-05-27): Observe that values tend to stabilize around 400-600 kW. - **After Heat Wave** (e.g., 2013-05-28 onwards): Values increase significantly due to higher air conditioning usage, potentially reaching up to 700 kW.

3. **Forecast Adjustments**: - For the immediate post-heat wave period (2013-05-28), expect a continuation of the high values seen during the heat wave but with some normalization as it cools down. - After the initial spike, the forecast should follow the pre-heat wave trend. - During the subsequent days (2013-05-29 and beyond), the electricity consumption should return to a more stable level, possibly within the normal range observed before the heat wave.

Based on this analysis, the adjusted forecasts can be made accordingly.

Figure 10: An Incorrect Reasoning Trace (Model: Qwen2.5-3B-Inst)

To improve the forecast, I will incorporate the context as follows:
1. **Background:** The data represents electricity consumption in Kilowatt (kW) in city A. This information is useful to understand the general trend and pattern of electricity consumption.
2. **Scenario:** There is a heat wave in city A from 2013-05-28 12:00:00 for 2 hours, leading to excessive use of air conditioning, and 4 times the usual electricity being consumed. This information is crucial to adjust the forecast for the specified time period. I will make sure to increase the forecasted values for 2013-05-28 12:00:00 and 2013-05-28 13:00:00 by a factor of 4.
To generate the forecast, I will use a simple moving average approach, considering the patterns and trends observed in the historical data.

Figure 11: An Incorrect Reasoning Trace (Model: Llama3.1-405B-Inst)

### C.5.2  Task: *SensorMaintenanceInPredictionTask*

Background: This series represents the occupancy rate (%) captured by a highway sensor.
Constraints: None.
Scenario: Consider that the meter will be offline for maintenance between 2024-01-18 08:00:00 and 2024-01-18 14:00:00, which results in zero readings.

Figure 12: Context

The context mentions that the meter will be offline for maintenance between 2024-01-18 08:00:00 and 2024-01-18 14:00:00. During this period, one should forecast a value of 0

Figure 13: Gold Standard Reasoning Trace

The provided historical data shows a daily pattern where occupancy rates increase significantly during morning hours, peak around midday, and then decrease gradually throughout the rest of the day. The meter will be offline for maintenance from 2024-01-18 08:00:00 to 2024-01-18 14:00:00, resulting in zero readings during this period. To account for this, we should predict normal values until 08:00 and set all predictions to zero during the maintenance window. After 14:00, we should resume forecasting based on the historical patterns.

Figure 14: A Correct Reasoning Trace (Model: Qwen2.5-14B-Inst)

To incorporate the context provided into the forecast:
1. **Background**: The series represents an occupancy rate captured by a highway sensor. This indicates we should expect patterns related to traffic flows, which are likely influenced by daily routines such as rush hours, commuting, and other regular activities.
2. **Meter Offline Scenario**: The meter will be offline for maintenance between 2024-01-18 08:00:00 and 2024-01-18 14:00:00, resulting in zero readings during this period. For the timestamps within this range, we must predict zero occupancy rates directly based on this constraint.
Given the data history, it shows significant variability, particularly around early morning and late evening times, suggesting peak and off-peak traffic periods. We can assume similar patterns will continue unless there's a specific reason to believe otherwise.
For the timestamps before and after the maintenance period, we can infer trends from the previous days. Observing the last few days (January 15 to January 17), we see that the occupancy rate typically peaks in the morning around 7 AM, drops significantly midday, and then has another small peak in the late afternoon/early evening. We will use these patterns to guide our predictions.

Figure 15: A Correct Reasoning Trace (Model: Qwen2.5-32B-Inst)

Given the provided historical data, the first step is to preprocess the data to handle the scenario where the meter is offline from 2024-01-18 08:00:00 to 2024-01-18 14:00:00. Since the readings are zero during this period, we need to identify patterns or trends before and after this period to forecast values accurately.
### Step-by-Step Reasoning:
1. **Preprocess Data**: Identify the missing values due to the meter being offline. - From the history, we see that the readings from 2024-01-18 08:00:00 to 2024-01-18 14:00:00 are all zeros.
2. **Identify Patterns Before and After**: Analyze the trend before and after the offline period to make informed predictions. - Examine the trends from the day before (2024-01-17) and the next day (2024-01-19) to understand any seasonality or cyclical patterns.
3. **Forecast Values**: Use the identified patterns and possibly apply smoothing techniques like moving averages or seasonal decomposition to predict the missing values.
4. **Return Forecasts**: Return the predicted values for the requested timestamps.

Figure 16: An Incorrect Reasoning Trace (Model: Qwen2.5-3B-Inst)

1. **Understanding the Context**: The provided time series data represents an occupancy rate captured by a highway sensor. The sensor is expected to be offline from 2024-01-18 08:00:00 to 2024-01-18 14:00:00, which means the data for these timestamps will be missing. This gap needs to be accounted for in the forecasting model.
2. **Analyzing the Historical Data**: The historical data shows a range of occupancy rates, indicating a daily cycle. The data starts at a lower occupancy rate in the early hours and gradually increases to a peak in the late morning and early afternoon, then decreases as the day progresses.
3. **Model Selection**: Given the daily cycle pattern, an ARIMA model could be suitable. However, considering the specific missing data, a more comprehensive model like a Seasonal ARIMA (SARIMA) might be necessary to account for both the daily seasonality and the missing data.
4. **Handling Missing Data**: For the period during the maintenance (2024-01-18 08:00:00 to 2024-01-18 14:00:00), we will impute the values with a method that considers the surrounding data points to make the forecast as accurate as possible. We can use the average or median of the surrounding data points to fill in the gaps.
5. **Forecasting**: We will extend the model with the historical data and imputed values for the maintenance period. The forecast will then be continued beyond the maintenance period based on the identified pattern.

Figure 17: An Incorrect Reasoning Trace (Model: Qwen2.5-7B-Inst)

### C.5.3 Task: *CashDepletedinATMScenarioTask*

Background: This is the number of cash withdrawals from an automated teller machine (ATM) in an arbitrary location in England.
Constraints: None.
Scenario: Consider that cash is depleted in the ATM from 1996-11-24 00:00:00, for 10 days, resulting in no withdrawals during that period.

Figure 18: Context

A forecaster could logically produce a forecast by identifying the 10-day period of zero withdrawals as an anomaly due to cash depletion, then modeling expected withdrawal counts for other periods based on historical data while treating the anomaly as missing or censored data rather than as indicative of typical demand.

Figure 19: Gold Standard Reasoning Trace

To address this forecasting task, I will first assess the historical data pattern and then incorporate the provided context, which includes a scenario where the ATM was out of cash for ten days starting from 1996-11-24 00:00:00. The historical data shows variability, indicating a mix of daily and potentially weekly patterns. The scenario of the ATM being out of cash for ten days implies that for these days, there would be no withdrawals recorded. Following this event, it is reasonable to expect an increase in withdrawal numbers as the machine is replenished and users resume their normal withdrawal behaviors.

Given the lack of specific seasonal or trend data beyond day-to-day fluctuations, the forecast will be based on averaging recent withdrawal counts while considering the impact of the ATM's refill post-cash depletion. Since the exact replenishment amount and the timing of users resuming withdrawals are unknown, I'll use a conservative approach by estimating withdrawals to return to a typical level observed in the past few weeks, taking into account the likely higher activity after the refill.

For the period when the ATM was empty (1996-11-24 to 1996-12-03), I will predict zero withdrawals. After this period, withdrawals will be predicted based on a moving average of the recent non-zero values, adjusted for the expected increase due to replenishment.

Figure 20: A Correct Reasoning Trace (Model: Qwen2.5-32B-Inst)

Given the provided historical data, it appears there was a significant disruption in the time series on November 13, 1996, as the withdrawal count dropped significantly and remained low for the next several days. It is likely that this event had a strong influence on the subsequent values. To incorporate this information effectively:

1. **Identify the Abrupt Change**: The value drops sharply from 1996-11-13 to 1996-11-14 (from approximately 21.97 to 0.23). This drop suggests a sudden decrease in transactions which could be influenced by various factors such as an event or system downtime.

2. **Examine Patterns Before and After**: The immediate aftermath shows a sustained low level of transactions followed by a gradual increase. The drop and recovery might imply that once the disruption is resolved, activity will likely return to its previous levels but may not reach pre-disruption levels immediately due to residual effects.

3. **Forecasting Strategy**: - For the days directly affected (11/13-11/14), we can forecast based on the abrupt drop. Assuming the trend resumes after the initial shock, we can predict higher values compared to the pre-shock trend. - For the subsequent days, the forecast should reflect a gradual increase towards normalcy.

Figure 21: An Incorrect Reasoning Trace (Model: Qwen2.5-3B-Inst)

## C.6 Reasoning Quality Evaluation with Human Judges

The proposed method uses an LLM judge (GPT-4.1) to compare the ground truth reasoning traces with a model's reasoning trace. While such an LLM-as-a-judge approach is scalable and common in the literature (Gu et al., 2024), some works have shown that they can be systematically biased to certain factors such as the wording of the text, the length of the text, the LLM used as the judge etc. (Ye et al., 2025a).

To verify this approach, we ran a human evaluation, using humans as the judge instead of the LLM. The evaluators see the same information as the LLM judge, namely the ground truth reasoning trace, and a model's reasoning trace, to answer the same question of whether the model's reasoning is aligned with the key points mentioned in the ground truth reasoning (yes / no). Note that the evaluators do not see the results of the LLM-as-a-judge, and the human evaluation was done independently.

We run this for all the tasks and all the models considered in the experiment, and measure the agreement of the evaluators and the LLM-judge on the reasoning correctness as well as the meaningful improvement of the forecast with context.

The percentage agreement between the human and LLM judges, per model is given in Figure 22. On average, we find that a high agreement of 89.9% on the reasoning correctness, i.e. 89.9% of the time, the model and the humans agree on the correctness of reasoning of the model. This shows that the LLM-as-a-judge approach (specifically with the GPT-4.1 model) may suffice as a scalable alternative compared to human evaluation for future models.

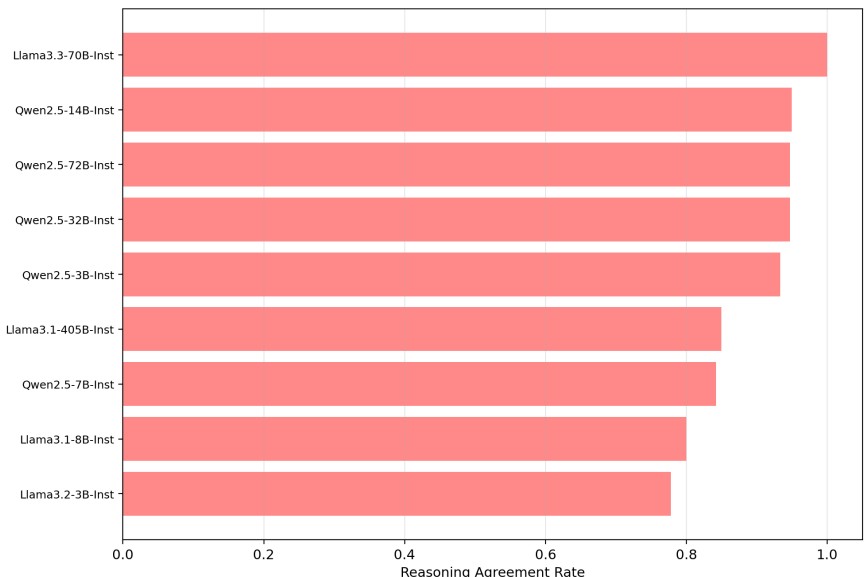

Figure 22: Reasoning Quality Agreement between human and LLM judges

## C.7  End-to-end Analyses Examples

### C.7.1  Task: *ATMBuildingClosedTask*

Background: This is the number of cash withdrawals from an automated teller machine (ATM) in an arbitrary location in England.
Constraints: None.
Scenario: Consider that the building which contains the ATM is closed from 1996-11-24 00:00:00, for 10 days.

Figure 23: Context

Since the building housing the ATM is closed for 10 days starting 1996-11-24, a logical forecast would set the number of cash withdrawals to zero for that period, as no one can access the ATM during the closure.

Figure 24: Gold Standard Reasoning Trace

First, we need to understand the historical data and identify any patterns or seasonal trends in the time series. In this case, we see that the number of cash withdrawals from the ATM is quite seasonal, with the highest values in the summer months and lowest values in the winter months.

Next, we need to incorporate the background information about the building being closed from 1996-11-24 00:00:00 for 10 days into our model. This could potentially impact the number of cash withdrawals during that period, so we need to account for this in our forecasting model.

We also need to consider the given historical data and ensure that our model is not overfitting to the training data. We can do this by using a technique such as cross-validation to evaluate the performance of our model on unseen data.

Another important consideration is the fact that the data is from England and we should consider the local seasonal patterns, if any.

Finally, we need to decide on a suitable forecasting model that can handle seasonal and non-seasonal trends, as well as the potential impact of the building closure on the data.

(a) Reasoning trace

(b) Forecast without context (RCRPS=0.434)

(c) Context-aided Forecast (RCRPS=0.565)

Figure 25: Model: Llama3.2-3B-Inst. The reasoning trace is evaluated as **Incorrect Reasoning**. The forecast changes by -30.1% and is hence evaluated as **No Improvement with Context**.

To improve the forecast, we need to consider the context provided. The building which contains the ATM is closed from 1996-11-24 00:00:00 for 10 days. This means that there will be no cash withdrawals during this period.

First, we need to identify the start and end dates of the closure. The start date is 1996-11-24 00:00:00, and the end date is 1996-12-04 00:00:00 (10 days after the start date).

Next, we need to adjust our forecast to reflect the closure. For the dates between 1996-11-24 00:00:00 and 1996-12-04 00:00:00, we should predict a value of 0, since there will be no cash withdrawals during this period.

For the dates after 1996-12-04 00:00:00, we can use a time series forecasting model, such as ARIMA or Prophet, to predict the values. We can train the model using the historical data provided.

The historical data shows a daily frequency, with some fluctuations in the values. We can use this data to identify patterns and trends, and then use these patterns to make predictions for the future dates.

Overall, the context provided is relevant to the forecast, as it gives us information about a specific event (the closure of the building) that will affect the time series.

(a) Reasoning trace

(b) Forecast without context (RCRPS=0.608)

(c) Context-aided Forecast (RCRPS=0.013)

Figure 26: Model: Llama-3.3-70B-Inst. The reasoning trace is evaluated as **Correct Reasoning**. The forecast changes by 97.9% and is hence evaluated as **Improvement with Context**.

The context provides us with a scenario where the building containing the ATM will be closed from 1996-11-24 00:00:00 for 10 days. During this period, we can reasonably expect the ATM to be inaccessible and therefore no cash withdrawals will occur. This means that for the timestamps '1996-11-24 00:00:00' to '1996-11-27 00:00:00', we should predict zero values as the ATM will not be operational. For the remaining timestamps, we can use the historical data to forecast the expected withdrawal counts, taking into account the closure period. However, after the closure ends on 1996-11-28 00:00:00, the number of withdrawals should return to a level similar to the pre-closure trend, but it might take some time for usage to fully recover to normal levels.

(a) Reasoning trace

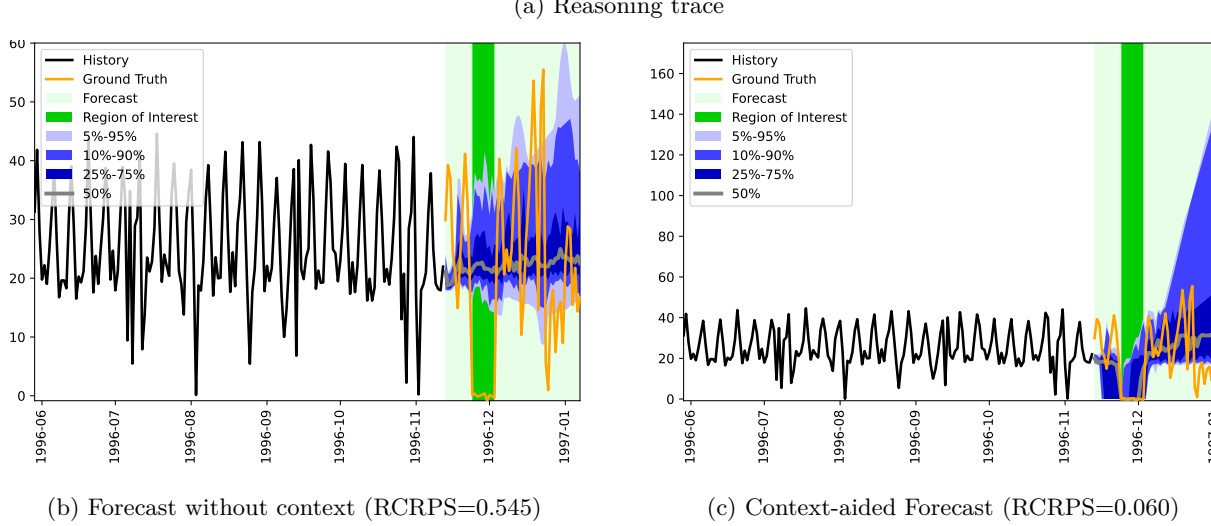

(b) Forecast without context (RCRPS=0.545)

(c) Context-aided Forecast (RCRPS=0.060)

Figure 27: Model: Qwen2.5-14B-Inst. The reasoning trace is evaluated as **Correct Reasoning**. The forecast changes by 88.9% and is hence evaluated as **Improvement with Context**.

To improve my forecast, I will incorporate each piece of the context as follows:
1. **Background**: The data represents the number of cash withdrawals from an automated teller machine (ATM) in an arbitrary location in England. This context is relevant as it helps me understand the type of data I'm dealing with and potential seasonal or periodic patterns.
2. **Scenario**: The building containing the ATM is closed from 1996-11-24 00:00:00 for 10 days. This context is crucial as it directly affects the data during the forecast period. I will ensure that my forecast reflects the closure by predicting zero or significantly reduced cash withdrawals during this period.
Next, I will analyze the historical data to identify any patterns, trends, or seasonality that can inform my forecast.

(a) Reasoning trace

(b) Forecast without context (RCRPS=0.609)

(c) Context-aided Forecast (RCRPS=0.025)

Figure 28: Model: Llama3.1-405B-Inst. The reasoning trace is evaluated as **Correct Reasoning**. The forecast changes by 95.9% and is hence evaluated as **Improvement with Context**.

### C.7.2 Task: *CashDepletedinATMScenarioTask*

Background: This is the number of cash withdrawals from an automated teller machine (ATM) in an arbitrary location in England.
Constraints: None.
Scenario: Consider that cash is depleted in the ATM from 1996-11-24 00:00:00, for 10 days, resulting in no withdrawals during that period.

Figure 29: Context

A forecaster could logically produce a forecast by identifying the 10-day period of zero withdrawals as an anomaly due to cash depletion, then modeling expected withdrawal counts for other periods based on historical data while treating the anomaly as missing or censored data rather than as indicative of typical demand.

Figure 30: Gold Standard Reasoning Trace

Given the provided historical data, it appears there was a significant disruption in the time series on November 13, 1996, as the withdrawal count dropped significantly and remained low for the next several days. It is likely that this event had a strong influence on the subsequent values. To incorporate this information effectively:
1. **Identify the Abrupt Change**: The value drops sharply from 1996-11-13 to 1996-11-14 (from approximately 21.97 to 0.23). This drop suggests a sudden decrease in transactions which could be influenced by various factors such as an event or system downtime.
2. **Examine Patterns Before and After**: The immediate aftermath shows a sustained low level of transactions followed by a gradual increase. The drop and recovery might imply that once the disruption is resolved, activity will likely return to its previous levels but may not reach pre-disruption levels immediately due to residual effects.
3. **Forecasting Strategy**: - For the days directly affected (11/13-11/14), we can forecast based on the abrupt drop. Assuming the trend resumes after the initial shock, we can predict higher values compared to the pre-shock trend. - For the subsequent days, the forecast should reflect a gradual increase towards normalcy.

(a) Reasoning trace

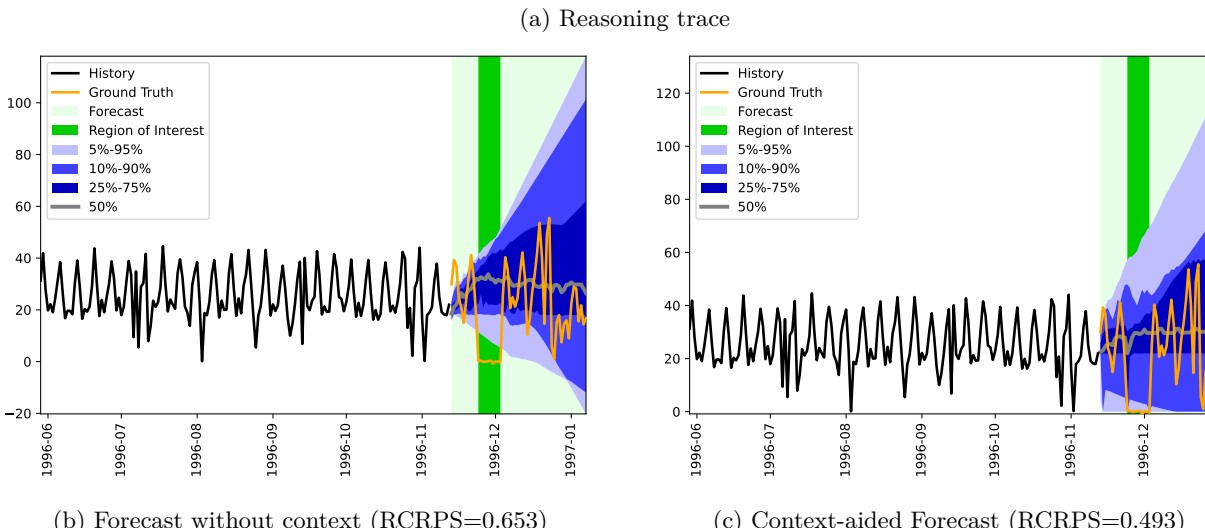

(b) Forecast without context (RCRPS=0.653)

(c) Context-aided Forecast (RCRPS=0.493)

Figure 31: Model: Qwen2.5-3B-Inst. The reasoning trace is evaluated as **Incorrect Reasoning**. The forecast changes by 24.6% and is hence evaluated as **No Improvement with Context**.

To address this forecasting task, I will first assess the historical data pattern and then incorporate the provided context, which includes a scenario where the ATM was out of cash for ten days starting from 1996-11-24 00:00:00. The historical data shows variability, indicating a mix of daily and potentially weekly patterns. The scenario of the ATM being out of cash for ten days implies that for these days, there would be no withdrawals recorded. Following this event, it is reasonable to expect an increase in withdrawal numbers as the machine is replenished and users resume their normal withdrawal behaviors.

Given the lack of specific seasonal or trend data beyond day-to-day fluctuations, the forecast will be based on averaging recent withdrawal counts while considering the impact of the ATM's refill post-cash depletion. Since the exact replenishment amount and the timing of users resuming withdrawals are unknown, I'll use a conservative approach by estimating withdrawals to return to a typical level observed in the past few weeks, taking into account the likely higher activity after the refill.

For the period when the ATM was empty (1996-11-24 to 1996-12-03), I will predict zero withdrawals. After this period, withdrawals will be predicted based on a moving average of the recent non-zero values, adjusted for the expected increase due to replenishment.

(a) Reasoning trace

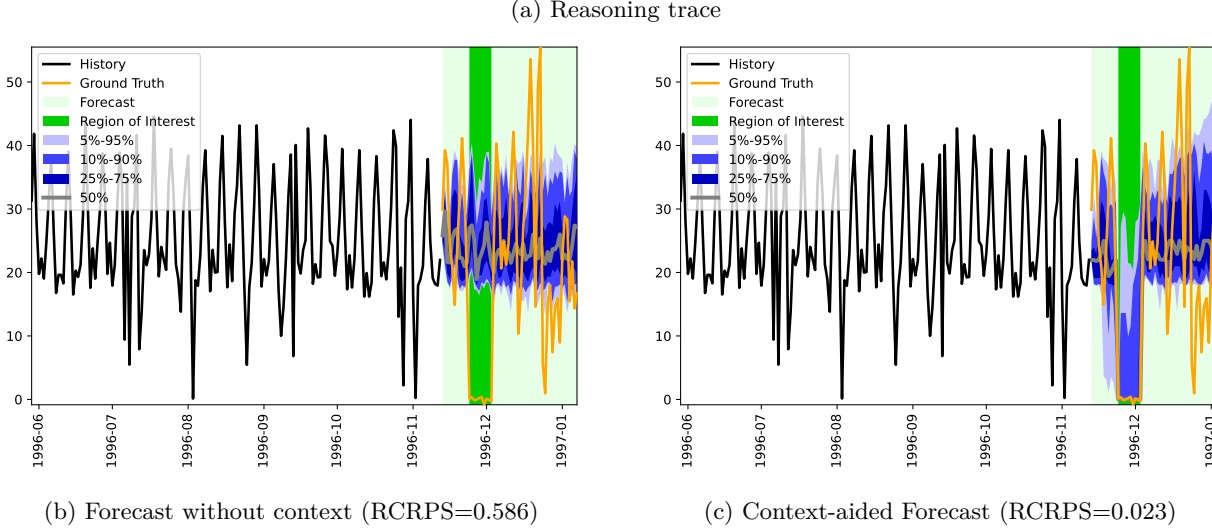

(b) Forecast without context (RCRPS=0.586)

(c) Context-aided Forecast (RCRPS=0.023)

Figure 32: Model: Qwen2.5-32B-Inst. The reasoning trace is evaluated as **Correct Reasoning**. The forecast changes by 96.0% and is hence evaluated as **Improvement with Context**.

1. **Context Inclusion**: We need to consider the context of the task, which includes the historical data showing a significant drop in withdrawals from 1996-11-24 to 1996-11-33 due to ATM cash depletion. This drop needs to be factored into our forecasting model.

2. **Model Choice**: Given the need for robust time-series forecasting and the presence of a data gap, we will use a combination of exponential smoothing and ARIMA (AutoRegressive Integrated Moving Average) models. These methods are well-suited for handling missing data and incorporate both the seasonal and trend components of the time series.

3. **Handling the Gap**: For the period from 1996-11-24 to 1996-11-33, we will assume that the withdrawals were zero or, if necessary, we will backfill the missing values using a conservative estimate or interpolation techniques. This is critical for avoiding bias in our forecasting model.

4. **Seasonality and Trend Analysis**: We will analyze the time series for seasonality and trends. The historical data shows variability, with some periodic patterns, which we will capture in our model.

5. **Model Training**: We will train our model on the entire historical dataset, ensuring to handle the gap appropriately. This will help the model learn from the historic consistency and variability.

6. **Forecasting**: Finally, we will use the trained model to predict future values, ensuring to respect the natural patterns and handle the gap by predicting zeros for the known period.

(a) Reasoning trace

(b) Forecast without context (RCRPS=0.572)

(c) Context-aided Forecast (RCRPS=0.193)

Figure 33: Model: Qwen-2.5-7B-Inst. The reasoning trace is evaluated as **Correct Reasoning**. The forecast changes by 66.2% and is hence evaluated as **Improvement with Context**.

### C.7.3   Task: *ElectricityIncreaseInPredictionTask*

Background: This series contains the amount of sunlight (in Watts per squared meter) arriving on a horizontal surface, for a location in Florida, United States. Over the previous 90 days, the maximum sunlight happened on average at 12:25:33.
Constraints: None
Scenario: None

Figure 34: Context

To forecast electricity consumption during the heat wave, multiply the usual consumption for each hour between 2013-05-28 12:00:00 and 2013-05-28 14:00:00 by 4, since the scenario specifies consumption is quadrupled due to excessive air conditioning use.

Figure 35: Gold Standard Reasoning Trace

To incorporate the given context into the time series forecasting task, we can follow these steps:
1. Preprocess the historical time series data: Clean and preprocess the historical data by converting all timestamps to a uniform format, handling missing values, and normalizing the data.
2. Identify the trend in the historical data: Use techniques such as linear regression, polynomial regression, or ARIMA to identify the trend in the historical data.
3. Account for seasonal patterns: Given that the data is hourly, it is likely that there are seasonal patterns in the data. We can use techniques such as seasonal decomposition or SARIMA models to account for these patterns.
4. Consider anomalies and outliers: Given that there was a heat wave in city A from 2013-05-28 12:00:00 for 2 hours, we can consider this as an anomaly in the data. We can use techniques such as ARIMA or Exponential Smoothing models to account for this anomaly.
5. Incorporate the heat wave anomaly into the model: We can use the heat wave anomaly as a basis to predict the consumption for the next two hours after the heat wave. This will require a model that can capture the anomalies and make predictions based on them.
6. Use the model to forecast future values: Once we have trained the model with the historical data and the heat wave anomaly, we can use it to forecast future values for the next 24 hours.

(a) Reasoning trace

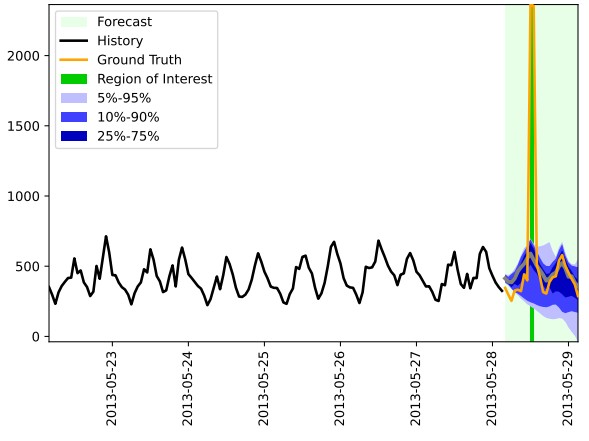

(b) Forecast without context (RCRPS=0.068)

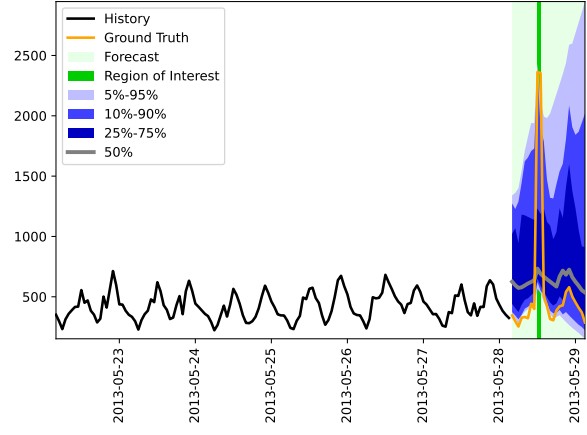

(c) Context-aided Forecast (RCRPS=0.038)

Figure 36: Model: Llama3.2-3B-Inst. The reasoning trace is evaluated as **Incorrect Reasoning**. The forecast changes by 43.9% and is hence evaluated as **No Improvement with Context**.

1. **Understand the Historical Data**: Analyze the historical data to identify patterns, trends, and seasonality. The data shows hourly electricity consumption over a period, which can be used to understand typical consumption patterns.
2. **Identify the Heat Wave Period**: The heat wave is specified to occur from 2013-05-28 12:00:00 for 2 hours. During this period, the electricity consumption is expected to be 4 times the usual amount. This is a critical piece of information that needs to be incorporated into the forecast.
3. **Adjust for the Heat Wave**: For the timestamps 2013-05-28 12:00:00 and 2013-05-28 13:00:00, the forecasted values should be adjusted to 4 times the expected consumption based on historical data. For the other timestamps, the forecast can be made based on historical patterns, considering the usual daily and weekly trends.
4. **Use Historical Data for Forecasting**: Use the historical data to predict the values for the remaining timestamps. For example, if the consumption on a similar day (e.g., 5 days prior) was around 400 kW at 04:00:00, the forecast for 2013-05-28 04:00:00 can be based on this value.
5. **Incorporate Seasonality**: Ensure that the forecast takes into account the typical increase in consumption during the afternoon and evening hours, which is evident from the historical data.
6. **Relevance of Context**: The context about the heat wave is highly relevant and must be factored into the forecast. The historical data is also crucial for making accurate predictions.

(a) Reasoning trace

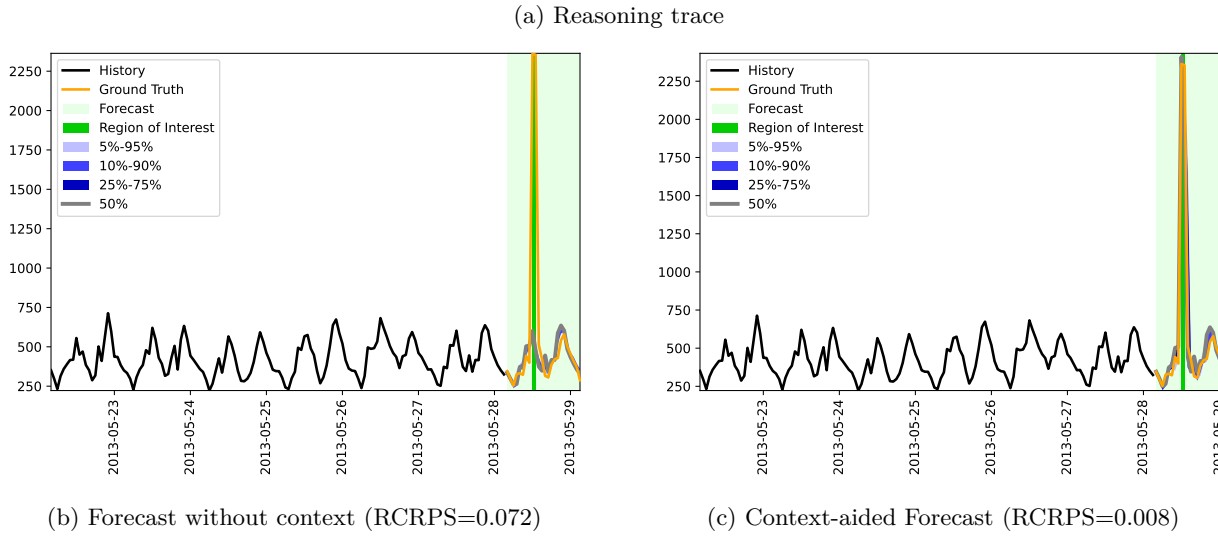

(b) Forecast without context (RCRPS=0.072)  (c) Context-aided Forecast (RCRPS=0.008)

Figure 37: Model: Qwen-2.5-72B-Inst. The reasoning trace is evaluated as **Correct Reasoning**. The forecast changes by 89.3% and is hence evaluated as **Improvement with Context**.

To improve the forecast, we need to incorporate the given context. The scenario mentions a heat wave in city A from 2013-05-28 12:00:00 for 2 hours, leading to excessive use of air conditioning and 4 times the usual electricity being consumed.

First, we need to identify the usual electricity consumption pattern. From the historical data, we can observe the daily patterns and weekly patterns. However, we don't have any information about the seasonality or yearly patterns.

The heat wave is the most critical context for this forecasting task. We should consider that the electricity consumption will be 4 times the usual consumption during the heat wave (from 2013-05-28 12:00:00 to 2013-05-28 14:00:00).

For the timestamps before the heat wave (2013-05-28 04:00:00 to 2013-05-28 11:00:00), we can use the historical data to identify the daily pattern and forecast the values.

For the timestamps during the heat wave (2013-05-28 12:00:00 to 2013-05-28 14:00:00), we will multiply the forecasted values by 4.

For the timestamps after the heat wave (2013-05-28 15:00:00 onwards), we will assume that the electricity consumption returns to normal and use the historical data to forecast the values.

By incorporating the context, we can improve the accuracy of the forecast.

(a) Reasoning trace

(b) Forecast without context (RCRPS=0.069)

(c) Context-aided Forecast (RCRPS=0.006)

Figure 38: Model: Llama-3.3-70B-Inst. The reasoning trace is evaluated as **Correct Reasoning**. The forecast changes by 91.9% and is hence evaluated as **Improvement with Context**.

### C.7.4 Task: *ExplicitWithDatesAndDaysTrafficForecastTaskwithHolidaysInPredictionWindow*

Background: This series contains the road occupancy rates on a freeway in the San Francisco Bay area. The days for which the forecast is required are Thursday 2024-07-04, Friday 2024-07-05, Saturday 2024-07-06. Note that 2024-07-04 is a holiday due to Independence Day. Note that traffic on this freeway typically reduces on holidays.
Constraints: None.
Scenario: None.

Figure 39: Context

A logical forecast can be produced by analyzing historical occupancy rates for the same weekdays and dates, especially focusing on past Independence Days and adjacent days, to account for typical holiday traffic reductions and altered travel patterns.

Figure 40: Gold Standard Reasoning Trace

To create an effective forecast, let's break down the context and steps needed:
1. **Understand the Context**: The given data shows daily occupancy rates for a freeway in the San Francisco Bay area over a week. We need to focus on Thursday, Friday, and Saturday forecasts for July 4th, 2024, and July 5th, 2024, noting that these dates fall on a holiday (Independence Day).
2. **Holiday Impact**: Holidays generally see a reduction in traffic volumes compared to regular days. Observing past data on similar holidays can help estimate the occupancy rate on July 4th, 2024.
3. **Pattern Analysis**: Identify if there are any seasonal patterns or trends that we can leverage. For example, weekend traffic might be slightly higher than weekdays but less than the peak on workdays.
4. **Use Historical Data**: Use the provided historical data to train our model and make predictions.
5. **Consider Known Holidays**: Since Independence Day falls on a Thursday, we know the occupancy rate will likely be lower than usual for that day.
6. **Forecast the Remaining Days**: After understanding the Independence Day pattern, we can extrapolate the occupancy rates for the remaining days in the forecast.

(a) Reasoning trace

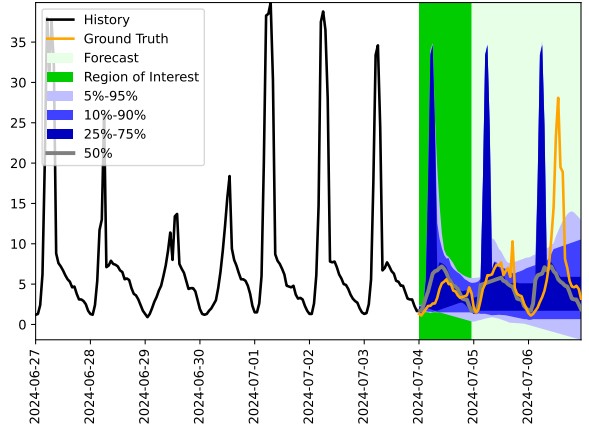

(b) Forecast without context (RCRPS=0.135)

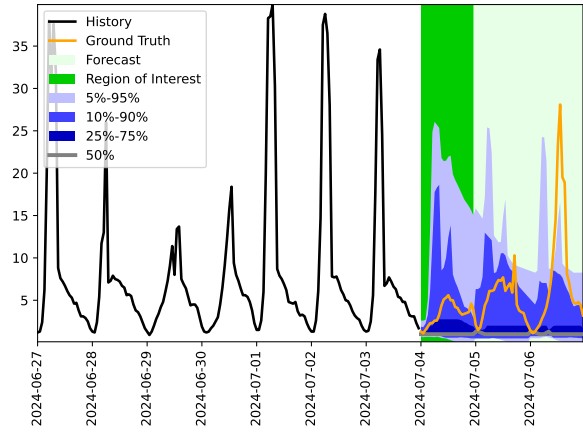

(c) Context-aided Forecast (RCRPS=0.148)

Figure 41: Model: Qwen2.5-3B-Inst. The reasoning trace is evaluated as **Correct Reasoning**. The forecast changes by -9.7% and is hence evaluated as **No Improvement with Context**.

I'll incorporate the given context into the forecasting process as follows:
1. Understanding the historical data: I'll analyze the historical time series data to identify any patterns, trends, or seasonal cycles that can be used to inform the forecast.
2. Accounting for the holiday: Given that July 4th is a holiday, which typically results in reduced traffic, I'll adjust the forecast accordingly. This may involve applying a lower growth rate or a more conservative estimate to the forecasted values.
3. Incorporating additional context: The context does not provide any specific additional information, but I'll consider the typical daily patterns in traffic, such as the morning and evening commutes.

(a) Reasoning trace

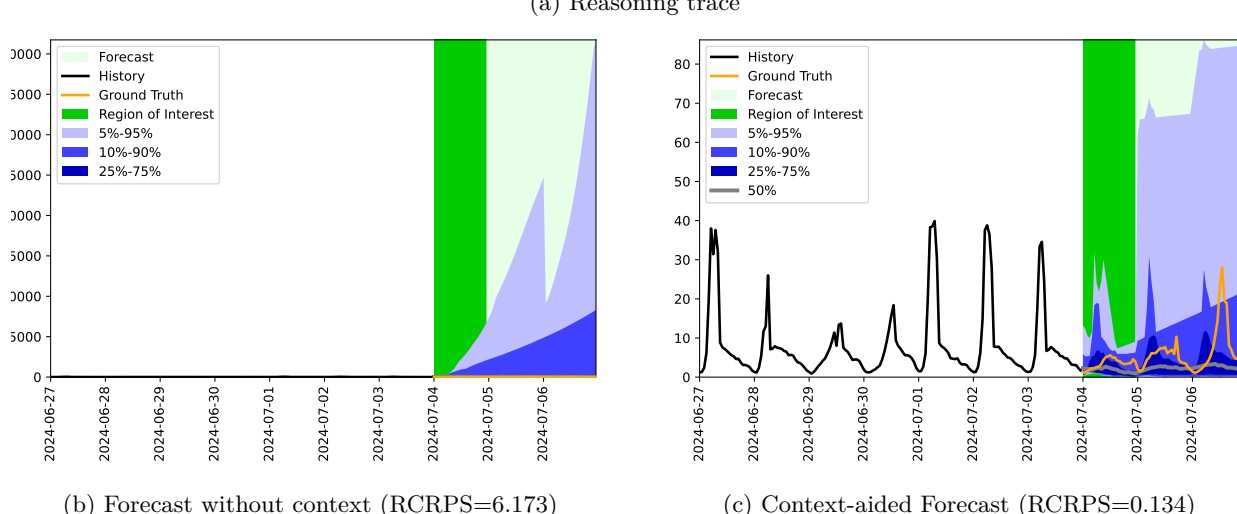

(b) Forecast without context (RCRPS=6.173)

(c) Context-aided Forecast (RCRPS=0.134)

Figure 42: Model: Llama3.2-3B-Inst. The reasoning trace is evaluated as **Correct Reasoning**. The forecast changes by 97.8% and is hence evaluated as **Improvement with Context**.

To incorporate the given context effectively into our forecasting model, we need to consider the following aspects:

1. **Holiday Effect**: Since July 4th is a holiday (Independence Day), we expect lower occupancy rates compared to regular Thursdays. We can observe from the history that typical Thursday patterns show higher occupancy during morning and evening rush hours, whereas holiday patterns usually have reduced traffic throughout the day.

2. **Day of Week Patterns**: Regular days follow certain patterns. Typically, weekdays like Thursday, Friday, and Saturday (though Saturday is usually quieter) have different occupancy levels based on the time of day. For example, mornings and evenings are busier than midday or late night.

3. **Trend Analysis**: From the historical data, we notice that there is a significant drop in occupancy rates as we move towards late hours. The peak times for congestion tend to be around 7 AM to 8 AM and 5 PM to 6 PM on normal days.

Given that July 4th is a holiday, we can adjust the expected occupancy rates to be generally lower across all times of the day. For July 5th and 6th, since they are a Friday and Saturday, respectively, we expect slightly lower than usual traffic but still more than a full holiday.

Therefore, we will adjust the forecasted values by considering these patterns and the impact of the holiday.

(a) Reasoning trace

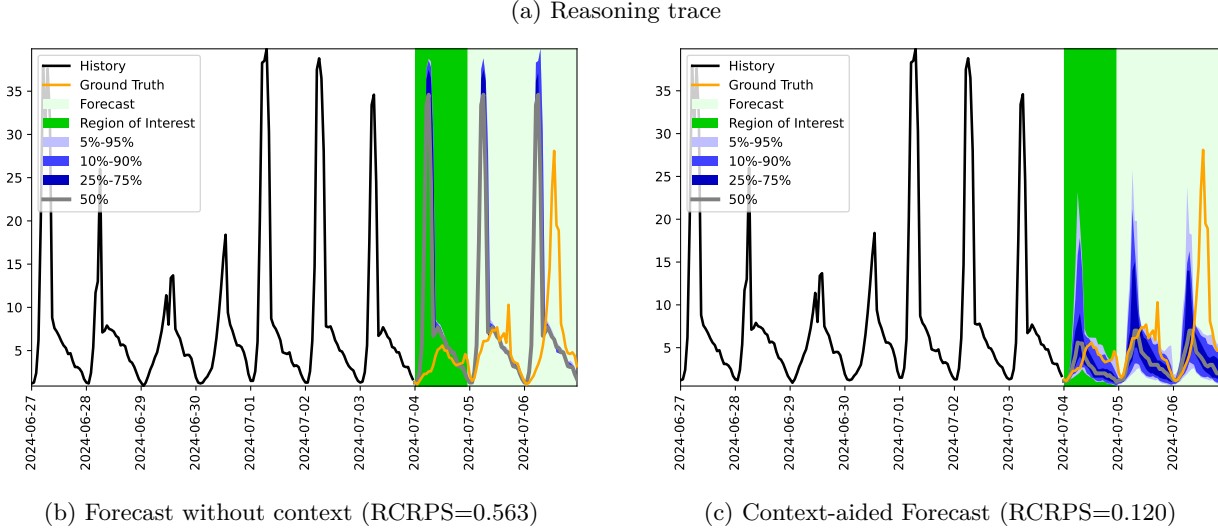

(b) Forecast without context (RCRPS=0.563)    (c) Context-aided Forecast (RCRPS=0.120)

Figure 43: Model: Qwen2.5-14B-Inst. The reasoning trace is evaluated as **Correct Reasoning**. The forecast changes by 78.7% and is hence evaluated as **Improvement with Context**.

1. **Analyzing Holiday Impact**: Since 2024-07-04 is a holiday (Independence Day), traffic is expected to be significantly lower than on a regular day. This is a key consideration for the forecast. 2. **Identifying Patterns**: From the historical data, there are clear patterns during different times of the day. For example, traffic peaks around 6 AM and drops off around 9 AM before picking up again around noon. 3. **Averaging and Adjusting for Holidays**: On holidays, we can look at the average traffic from the day before the holiday (2024-07-03) and adjust for the holiday effect, which we expect to be a substantial reduction. 4. **Carrying Forward Pattern**: For the days after the holiday (2024-07-05 and 2024-07-06), we will extend the pattern observed on 2024-07-03 while accounting for the potential easing of restrictions on weekend traffic.

(a) Reasoning trace

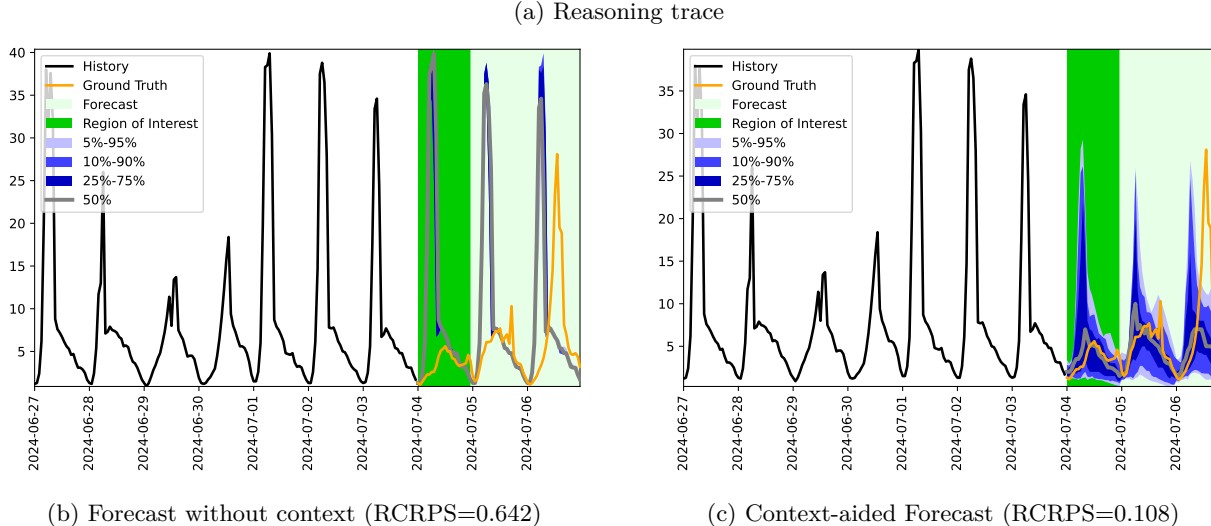

(b) Forecast without context (RCRPS=0.642)  (c) Context-aided Forecast (RCRPS=0.108)

Figure 44: Model: Qwen-2.5-7B-Inst. The reasoning trace is evaluated as **Correct Reasoning**. The forecast changes by 83.1% and is hence evaluated as **Improvement with Context**.

# D   Additional Details on CorDP

## D.1   CorDP Prompt

We use the following prompt for the CorDP method, where **{base_forecasts}** are replaced by the forecasts of the quantitative forecaster in the format: (timestep1, value1), (timestep2, value2), ... (timestepN, valueN) where N is the prediction length. **{history}** is replaced by the respective numerical history for the task instance in the format (timestamp, value), **{context}** is replaced by the respective textual context for the task instance, and **((pred_time))** is replaced with the prediction timesteps.

```
I have a time series forecasting task for you.

Here is some context about the task. Make sure to factor in any background knowledge,
satisfy any constraints, and respect any scenarios.
<context>
{context}
</context>

Here is a historical time series in (timestamp, value) format:
<history>
{history}
</history>

And these are the forecasts of my statistical forecasting model in (timestamp, value) format:
<base_forecast>
{base_forecasts}
</base_forecast>

My statistical forecasting model does not support taking in context as part of its input. I
    would like you to correct its forecasts to incorporate the context wherever necessary, and
    return the corrected context-aware forecast.
Return the corrected forecast in (timestamp, value) format in between <corrected_forecast> and
    </corrected_forecast> tags.
Do not include any other information (e.g., comments) in the forecast.
```

| Model | Direct Prompt (DP) | Median Corrector (Median-CorDP) | | | SampleWise Corrector (SampleWise-CorDP) | | |
|---|---|---|---|---|---|---|---|
| | | LAG-LLAMA | CHRONOS LARGE | ARIMA | LAG-LLAMA | CHRONOS LARGE | ARIMA |
| Qwen2.5-0.5B-Inst | $0.339 \pm 0.010$ | $0.302 \pm 0.001$ | $0.553 \pm 0.000$ | $0.336 \pm 0.001$ | $\mathbf{0.235 \pm 0.006}$ | $0.438 \pm 0.014$ | $0.272 \pm 0.004$ |
| Qwen2.5-1.5B-Inst | $0.317 \pm 0.020$ | $0.296 \pm 0.002$ | $0.538 \pm 0.005$ | $0.323 \pm 0.002$ | $\mathbf{0.232 \pm 0.005}$ | $0.478 \pm 0.007$ | $0.278 \pm 0.006$ |
| Qwen2.5-3B-Inst | $0.269 \pm 0.015$ | $0.391 \pm 0.004$ | $0.420 \pm 0.004$ | $0.274 \pm 0.005$ | $\mathbf{0.219 \pm 0.005}$ | $0.388 \pm 0.008$ | $0.243 \pm 0.004$ |
| Qwen2.5-7B-Inst | $0.285 \pm 0.006$ | $\mathbf{0.125 \pm 0.002}$ | $0.198 \pm 0.006$ | $0.182 \pm 0.004$ | $0.135 \pm 0.004$ | $0.180 \pm 0.006$ | $0.146 \pm 0.004$ |
| Qwen2.5-14B-Inst | $\mathbf{0.162 \pm 0.005}$ | $0.288 \pm 0.002$ | $0.247 \pm 0.002$ | $0.236 \pm 0.005$ | $0.206 \pm 0.007$ | $0.221 \pm 0.004$ | $0.205 \pm 0.005$ |
| Qwen2.5-32B-Inst | $\mathbf{0.116 \pm 0.001}$ | $0.213 \pm 0.002$ | $0.156 \pm 0.002$ | $0.187 \pm 0.002$ | $0.145 \pm 0.005$ | $0.132 \pm 0.002$ | $0.137 \pm 0.005$ |
| Qwen2.5-72B-Inst | $\mathbf{0.115 \pm 0.004}$ | $0.158 \pm 0.003$ | $0.169 \pm 0.002$ | $0.141 \pm 0.003$ | $0.138 \pm 0.006$ | $0.140 \pm 0.004$ | $0.125 \pm 0.004$ |
| Llama-3.2-1B-Inst | $0.336 \pm 0.026$ | $0.281 \pm 0.004$ | $0.414 \pm 0.013$ | $0.311 \pm 0.002$ | $\mathbf{0.234 \pm 0.006}$ | $0.507 \pm 0.003$ | $0.269 \pm 0.005$ |
| Llama-3.2-3B-Inst | $0.281 \pm 0.013$ | $0.243 \pm 0.003$ | $0.368 \pm 0.006$ | $0.262 \pm 0.004$ | $\mathbf{0.214 \pm 0.005}$ | $0.362 \pm 0.007$ | $0.243 \pm 0.004$ |
| Llama-3-8B-Inst | $0.255 \pm 0.008$ | $0.167 \pm 0.005$ | $0.189 \pm 0.004$ | $0.164 \pm 0.003$ | $\mathbf{0.149 \pm 0.005}$ | $0.176 \pm 0.006$ | $0.150 \pm 0.005$ |
| Llama3.3-70B-Inst | $\mathbf{0.105 \pm 0.003}$ | $0.211 \pm 0.001$ | $0.163 \pm 0.001$ | $0.205 \pm 0.001$ | $0.164 \pm 0.005$ | $0.126 \pm 0.003$ | $0.152 \pm 0.003$ |
| Llama3.1-405B-Inst | $0.126 \pm 0.004$ | $0.212 \pm 0.004$ | $0.146 \pm 0.003$ | $0.168 \pm 0.003$ | $0.131 \pm 0.006$ | $\mathbf{0.117 \pm 0.003}$ | $0.144 \pm 0.003$ |
| GPT-4o | $0.123 \pm 0.004$ | $0.212 \pm 0.005$ | $0.118 \pm 0.001$ | $0.185 \pm 0.002$ | $0.156 \pm 0.006$ | $\mathbf{0.108 \pm 0.003}$ | $0.124 \pm 0.002$ |
| GPT-4o-mini | $0.263 \pm 0.005$ | $0.277 \pm 0.002$ | $0.270 \pm 0.002$ | $0.317 \pm 0.001$ | $\mathbf{0.224 \pm 0.006}$ | $0.241 \pm 0.003$ | $0.255 \pm 0.003$ |
| Base Quantitative Forecaster | - | $0.224 \pm 0.005$ | $0.536 \pm 0.003$ | $0.272 \pm 0.004$ | $0.224 \pm 0.005$ | $0.536 \pm 0.003$ | $0.272 \pm 0.004$ |

Table 8: Results (RoI CRPS) on RoI tasks in CiK. The best-performing method with each model in every group is in **bold**.

| Model | Direct Prompt (DP) | Median Corrector (Median-CorDP) | | | SampleWise Corrector (SampleWise-CorDP) | | |
|---|---|---|---|---|---|---|---|
| | | LAG-LLAMA | CHRONOS LARGE | ARIMA | LAG-LLAMA | CHRONOS LARGE | ARIMA |
| Qwen2.5-0.5B-Inst | $0.243 \pm 0.103$ | $\mathbf{0.116 \pm 0.007}$ | $0.501 \pm 0.008$ | $0.675 \pm 0.025$ | $0.236 \pm 0.028$ | $0.861 \pm 0.204$ | $0.716 \pm 0.044$ |
| Qwen2.5-1.5B-Inst | $0.706 \pm 0.147$ | $\mathbf{0.185 \pm 0.047}$ | $0.488 \pm 0.008$ | $0.680 \pm 0.022$ | $0.794 \pm 0.065$ | $0.485 \pm 0.010$ | $1.185 \pm 0.043$ |
| Qwen2.5-3B-Inst | $0.234 \pm 0.056$ | $0.483 \pm 0.008$ | $0.478 \pm 0.005$ | $0.469 \pm 0.024$ | $0.418 \pm 0.107$ | $0.474 \pm 0.005$ | $0.422 \pm 0.118$ |
| Qwen2.5-7B-Inst | $0.470 \pm 0.078$ | $0.507 \pm 0.009$ | $0.947 \pm 0.004$ | $0.547 \pm 0.026$ | $0.523 \pm 0.015$ | $\mathbf{0.146 \pm 0.065}$ | $0.537 \pm 0.036$ |
| Qwen2.5-14B-Inst | $0.039 \pm 0.015$ | $0.001 \pm 0.003$ | $\mathbf{0.000 \pm 0.005}$ | $0.051 \pm 0.009$ | $0.457 \pm 0.015$ | $0.455 \pm 0.010$ | $0.466 \pm 0.030$ |
| Qwen2.5-32B-Inst | $0.479 \pm 0.019$ | $0.001 \pm 0.009$ | $\mathbf{0.000 \pm 0.005}$ | $0.000 \pm 0.027$ | $0.758 \pm 0.012$ | $0.455 \pm 0.008$ | $0.455 \pm 0.028$ |
| Qwen2.5-72B-Inst | $0.032 \pm 0.028$ | $0.304 \pm 0.006$ | $\mathbf{0.000 \pm 0.006}$ | $0.003 \pm 0.007$ | $0.004 \pm 0.008$ | $0.000 \pm 0.008$ | $0.001 \pm 0.024$ |
| Llama-3.2-1B-Inst | $0.275 \pm 0.092$ | $\mathbf{0.084 \pm 0.011}$ | $0.482 \pm 0.015$ | $0.499 \pm 0.068$ | $0.905 \pm 0.027$ | $0.924 \pm 0.013$ | $1.168 \pm 0.053$ |
| Llama-3.2-3B-Inst | $1.030 \pm 0.090$ | $\mathbf{0.112 \pm 0.032}$ | $0.519 \pm 0.018$ | $0.502 \pm 0.081$ | $0.884 \pm 0.091$ | $0.487 \pm 0.016$ | $1.003 \pm 0.116$ |
| Llama-3-8B-Inst | $0.169 \pm 0.172$ | $\mathbf{0.061 \pm 0.011}$ | $0.481 \pm 0.015$ | $0.438 \pm 0.034$ | $0.609 \pm 0.029$ | $0.476 \pm 0.012$ | $0.943 \pm 0.033$ |
| Llama3.3-70B-Inst | $\mathbf{0.000 \pm 0.024}$ | $0.000 \pm 0.003$ | $0.001 \pm 0.007$ | $0.000 \pm 0.022$ | $0.002 \pm 0.010$ | $0.000 \pm 0.006$ | $0.000 \pm 0.022$ |
| Llama3.1-405B-Inst | $0.004 \pm 0.009$ | $0.060 \pm 0.031$ | $0.303 \pm 0.008$ | $0.006 \pm 0.025$ | $0.042 \pm 0.016$ | $\mathbf{0.000 \pm 0.009}$ | $0.228 \pm 0.027$ |
| GPT-4o | $0.455 \pm 0.029$ | $\mathbf{0.000 \pm 0.008}$ | $0.000 \pm 0.010$ | $0.000 \pm 0.021$ | $0.001 \pm 0.008$ | $0.000 \pm 0.010$ | $0.000 \pm 0.028$ |
| GPT-4o-mini | $\mathbf{0.001 \pm 0.032}$ | $0.006 \pm 0.004$ | $0.018 \pm 0.006$ | $0.002 \pm 0.003$ | $0.245 \pm 0.008$ | $0.019 \pm 0.009$ | $0.017 \pm 0.034$ |
| Base Quantitative Forecaster | - | $0.204 \pm 0.037$ | $0.487 \pm 0.010$ | $0.843 \pm 0.050$ | $0.204 \pm 0.037$ | $0.487 \pm 0.010$ | $0.843 \pm 0.050$ |

Table 11: Results (Constraint violation CRPS) on tasks with constraints. The best-performing method with each model in every group is in **bold**.

| Model | Direct Prompt (DP) | Median Corrector (Median-CorDP) | | | SampleWise Corrector (SampleWise-CorDP) | | |
|---|---|---|---|---|---|---|---|
| | | LAG-LLAMA | CHRONOS LARGE | ARIMA | LAG-LLAMA | CHRONOS LARGE | ARIMA |
| Qwen2.5-0.5B-Inst | $0.129 \pm 0.010$ | $0.283 \pm 0.001$ | $0.142 \pm 0.000$ | $0.206 \pm 0.001$ | $0.211 \pm 0.006$ | $\mathbf{0.111 \pm 0.014}$ | $0.159 \pm 0.004$ |
| Qwen2.5-1.5B-Inst | $0.224 \pm 0.020$ | $0.268 \pm 0.002$ | $0.140 \pm 0.005$ | $0.198 \pm 0.002$ | $0.193 \pm 0.005$ | $\mathbf{0.113 \pm 0.007}$ | $0.160 \pm 0.006$ |
| Qwen2.5-3B-Inst | $0.186 \pm 0.015$ | $0.251 \pm 0.004$ | $0.129 \pm 0.004$ | $0.179 \pm 0.005$ | $0.179 \pm 0.005$ | $\mathbf{0.114 \pm 0.008}$ | $0.134 \pm 0.004$ |
| Qwen2.5-7B-Inst | $0.164 \pm 0.006$ | $0.225 \pm 0.002$ | $0.137 \pm 0.006$ | $0.182 \pm 0.004$ | $0.167 \pm 0.004$ | $\mathbf{0.127 \pm 0.006}$ | $0.146 \pm 0.004$ |
| Qwen2.5-14B-Inst | $\mathbf{0.146 \pm 0.005}$ | $0.306 \pm 0.002$ | $0.212 \pm 0.002$ | $0.219 \pm 0.005$ | $0.210 \pm 0.007$ | $0.188 \pm 0.004$ | $0.200 \pm 0.005$ |
| Qwen2.5-32B-Inst | $0.140 \pm 0.001$ | $0.238 \pm 0.002$ | $0.143 \pm 0.002$ | $0.194 \pm 0.002$ | $0.164 \pm 0.005$ | $\mathbf{0.112 \pm 0.002}$ | $0.131 \pm 0.005$ |
| Qwen2.5-72B-Inst | $\mathbf{0.138 \pm 0.004}$ | $0.265 \pm 0.003$ | $0.192 \pm 0.002$ | $0.200 \pm 0.003$ | $0.181 \pm 0.006$ | $0.155 \pm 0.004$ | $0.158 \pm 0.004$ |
| Llama-3.2-1B-Inst | $0.248 \pm 0.026$ | $0.260 \pm 0.004$ | $0.107 \pm 0.013$ | $0.191 \pm 0.002$ | $0.191 \pm 0.006$ | $\mathbf{0.104 \pm 0.003}$ | $0.159 \pm 0.005$ |
| Llama-3.2-3B-Inst | $0.162 \pm 0.013$ | $0.213 \pm 0.003$ | $0.116 \pm 0.006$ | $0.152 \pm 0.004$ | $0.177 \pm 0.005$ | $\mathbf{0.107 \pm 0.007}$ | $0.136 \pm 0.004$ |
| Llama-3-8B-Inst | $\mathbf{0.163 \pm 0.008}$ | $0.257 \pm 0.005$ | $0.238 \pm 0.004$ | $0.208 \pm 0.003$ | $0.232 \pm 0.005$ | $0.198 \pm 0.006$ | $0.189 \pm 0.005$ |
| Llama3.3-70B-Inst | $0.182 \pm 0.003$ | $0.277 \pm 0.001$ | $0.157 \pm 0.001$ | $0.194 \pm 0.001$ | $0.205 \pm 0.005$ | $\mathbf{0.132 \pm 0.003}$ | $0.154 \pm 0.003$ |
| Llama3.1-405B-Inst | $0.150 \pm 0.004$ | $0.248 \pm 0.004$ | $0.170 \pm 0.003$ | $0.174 \pm 0.003$ | $0.163 \pm 0.006$ | $\mathbf{0.133 \pm 0.003}$ | $0.141 \pm 0.003$ |
| GPT-4o | $\mathbf{0.106 \pm 0.004}$ | $0.246 \pm 0.005$ | $0.140 \pm 0.001$ | $0.190 \pm 0.002$ | $0.159 \pm 0.006$ | $0.114 \pm 0.003$ | $0.145 \pm 0.002$ |
| GPT-4o-mini | $0.150 \pm 0.005$ | $0.282 \pm 0.002$ | $0.141 \pm 0.002$ | $0.198 \pm 0.001$ | $0.198 \pm 0.006$ | $\mathbf{0.117 \pm 0.003}$ | $0.150 \pm 0.003$ |
| Base Quantitative Forecaster | - | $0.202 \pm 0.005$ | $0.115 \pm 0.003$ | $0.159 \pm 0.004$ | $0.202 \pm 0.005$ | $0.115 \pm 0.003$ | $0.159 \pm 0.004$ |

Table 9: Results (non-RoI CRPS) on RoI tasks in CiK. The best-performing method with each model in every group is in **bold**.

| Model | Direct Prompt (DP) | Median Corrector (Median-CorDP) | | | SampleWise Corrector (SampleWise-CorDP) | | |
|---|---|---|---|---|---|---|---|
| | | LAG-LLAMA | CHRONOS LARGE | ARIMA | LAG-LLAMA | CHRONOS LARGE | ARIMA |
| Qwen2.5-0.5B-Inst | $0.836 \pm 0.046$ | $0.864 \pm 0.003$ | $1.110 \pm 0.006$ | $1.094 \pm 0.090$ | $\mathbf{0.679 \pm 0.013}$ | $0.895 \pm 0.127$ | $0.953 \pm 0.092$ |
| Qwen2.5-1.5B-Inst | $0.851 \pm 0.026$ | $\mathbf{0.525 \pm 0.021}$ | $0.672 \pm 0.005$ | $0.969 \pm 0.011$ | $0.733 \pm 0.030$ | $0.595 \pm 0.007$ | $1.059 \pm 0.021$ |
| Qwen2.5-3B-Inst | $0.558 \pm 0.027$ | $0.606 \pm 0.008$ | $0.638 \pm 0.006$ | $0.849 \pm 0.014$ | $\mathbf{0.533 \pm 0.048}$ | $0.587 \pm 0.007$ | $0.731 \pm 0.053$ |
| Qwen2.5-7B-Inst | $\mathbf{0.521 \pm 0.009}$ | $0.584 \pm 0.006$ | $0.964 \pm 0.013$ | $0.939 \pm 0.013$ | $0.538 \pm 0.011$ | $0.571 \pm 0.034$ | $0.808 \pm 0.019$ |
| Qwen2.5-14B-Inst | $\mathbf{0.310 \pm 0.010}$ | $0.328 \pm 0.004$ | $0.406 \pm 0.009$ | $0.556 \pm 0.007$ | $0.470 \pm 0.009$ | $0.551 \pm 0.009$ | $0.654 \pm 0.015$ |
| Qwen2.5-32B-Inst | $0.580 \pm 0.013$ | $\mathbf{0.263 \pm 0.007}$ | $0.355 \pm 0.009$ | $0.423 \pm 0.013$ | $0.416 \pm 0.008$ | $0.486 \pm 0.011$ | $0.604 \pm 0.014$ |
| Qwen2.5-72B-Inst | $\mathbf{0.253 \pm 0.015}$ | $0.392 \pm 0.014$ | $0.479 \pm 0.017$ | $0.603 \pm 0.015$ | $0.320 \pm 0.016$ | $0.441 \pm 0.016$ | $0.552 \pm 0.017$ |
| Llama-3.2-1B-Inst | $\mathbf{0.467 \pm 0.041}$ | $0.477 \pm 0.007$ | $0.687 \pm 0.008$ | $0.857 \pm 0.030$ | $0.765 \pm 0.014$ | $0.857 \pm 0.008$ | $0.983 \pm 0.025$ |
| Llama-3.2-3B-Inst | $1.004 \pm 0.040$ | $\mathbf{0.422 \pm 0.018}$ | $0.600 \pm 0.014$ | $0.821 \pm 0.037$ | $0.722 \pm 0.043$ | $0.551 \pm 0.012$ | $0.985 \pm 0.052$ |
| Llama-3-8B-Inst | $0.771 \pm 0.043$ | $\mathbf{0.385 \pm 0.006}$ | $0.615 \pm 0.008$ | $0.833 \pm 0.007$ | $0.586 \pm 0.015$ | $0.561 \pm 0.006$ | $0.953 \pm 0.016$ |
| Llama3.3-70B-Inst | $0.289 \pm 0.011$ | $0.306 \pm 0.004$ | $0.313 \pm 0.006$ | $0.456 \pm 0.010$ | $\mathbf{0.249 \pm 0.006}$ | $0.273 \pm 0.006$ | $0.419 \pm 0.011$ |
| Llama3.1-405B-Inst | $\mathbf{0.196 \pm 0.005}$ | $0.310 \pm 0.014$ | $0.272 \pm 0.006$ | $0.316 \pm 0.012$ | $0.235 \pm 0.009$ | $0.241 \pm 0.006$ | $0.288 \pm 0.013$ |
| GPT-4o | $0.455 \pm 0.014$ | $0.270 \pm 0.006$ | $0.316 \pm 0.007$ | $0.468 \pm 0.012$ | $\mathbf{0.201 \pm 0.007}$ | $0.254 \pm 0.007$ | $0.330 \pm 0.014$ |
| GPT-4o-mini | $0.513 \pm 0.017$ | $0.422 \pm 0.011$ | $0.431 \pm 0.006$ | $0.692 \pm 0.009$ | $\mathbf{0.363 \pm 0.014}$ | $0.375 \pm 0.008$ | $0.559 \pm 0.019$ |
| Base Quantitative Forecaster | - | $0.497 \pm 0.018$ | $0.605 \pm 0.006$ | $0.921 \pm 0.023$ | $0.497 \pm 0.018$ | $0.605 \pm 0.006$ | $0.921 \pm 0.023$ |

Table 10: Results (RCRPS) on tasks with a full RoI in CiK. The best-performing method with each model in every group is in **bold**.

## D.2 Results on various groups of tasks

We now look into results aggregated across the various kinds of tasks in the CiK benchmark: Table 8, Table 9 showcases performance of methods within and outside the region of interest (RoI) respectively for tasks that have an RoI, Table 10 shows performance across tasks where the entire prediction window is the RoI, and Table 11 shows constraint RCRPS across tasks with constraints. We find that SampleWise-CorDP has an advantage on tasks with an RoI, achieving the best performance in most models, both within and outside the RoI. Median-CorDP however has a clear advantage on tasks where the shape of the entire forecast is influenced by the context, which make up most of the benchmark, achieving the best performance in half the models, and trailing closely with DP in the other. These results also indicate that DP methods are still consistently strong in tasks where the entire prediction is influenced by the context. Median-CorDP overwhelmingly outperforms DP and bags the best performance in tasks with constraints, sometimes achieving perfect performance with large models. This shows that when choosing between CorDP methods, the kind of tasks that will be encountered is an important factor to consider.

## D.3 IC-CorDP: CorDP with an in-context example

To evaluate if the proposed CorDP and IC-DP methods can be combined to yield further gains, we evaluate a hybrid method, which we call IC-CorDP: this method uses CorDP as the foundation: the goal is to output a forecast given the history, context and a base forecast; when combined with IC-DP, it uses an in-context example that contains the history, context, base forecast and ground truth of the example. We abbreviate this hybrid method as IC-CorDP (In-Context Corrector Direct Prompt). We use the Median-CorDP for this experiment (and hence call this method IC-Median-CorDP), and run experiments with a subset of LLMs. As in CorDP, we test it with multiple base forecasters.

The results are in Table 12. IC-Median-CorDP improves performance compared to Median-CorDP across LLMs across all sizes, and across multiple base quantitative forecasters that the LLM bootstraps over. The levels of gains achieved with IC-Median-CorDP depend on the LLM and the base quantitative forecaster. This shows that there is clear potential in combining the two strategies to improve performance.

| Model | Direct Prompt (DP) | Median Corrector (Median-CorDP) | | | In-Context Median Corrector (IC-Median-CorDP) | | |
|---|---|---|---|---|---|---|---|
| | | Lag-Llama | Chronos Large | ARIMA | Lag-Llama | Chronos Large | ARIMA |
| Llama3.2-1B-Inst | $0.396 \pm 0.027$ | $0.394 \pm 0.004$ | $0.515 \pm 0.007$ | $0.612 \pm 0.018$ | $\mathbf{0.315 \pm 0.004}$ | $0.390 \pm 0.031$ | $0.480 \pm 0.010$ |
| Llama3.2-3B-Inst | $0.687 \pm 0.025$ | $0.344 \pm 0.011$ | $0.455 \pm 0.009$ | $0.573 \pm 0.022$ | $\mathbf{0.334 \pm 0.008}$ | $0.354 \pm 0.011$ | $0.478 \pm 0.016$ |
| Qwen2.5-0.5B-Inst | $0.592 \pm 0.027$ | $0.633 \pm 0.002$ | $0.801 \pm 0.003$ | $0.761 \pm 0.054$ | $\mathbf{0.358 \pm 0.005}$ | $1.734 \pm 0.008$ | $0.548 \pm 0.010$ |
| Qwen2.5-1.5B-Inst | $0.616 \pm 0.018$ | $0.426 \pm 0.013$ | $0.537 \pm 0.003$ | $0.682 \pm 0.006$ | $\mathbf{0.305 \pm 0.004}$ | $0.390 \pm 0.028$ | $0.334 \pm 0.009$ |
| Qwen2.5-3B-Inst | $0.424 \pm 0.017$ | $0.490 \pm 0.005$ | $0.491 \pm 0.004$ | $0.597 \pm 0.009$ | $\mathbf{0.326 \pm 0.008}$ | $0.475 \pm 0.009$ | $0.399 \pm 0.013$ |
| Qwen2.5-7B-Inst | $0.401 \pm 0.006$ | $0.419 \pm 0.004$ | $0.641 \pm 0.008$ | $0.633 \pm 0.008$ | $\mathbf{0.322 \pm 0.008}$ | $0.334 \pm 0.009$ | $0.449 \pm 0.010$ |
| Qwen2.5-14B-Inst | $\mathbf{0.247 \pm 0.006}$ | $0.315 \pm 0.003$ | $0.334 \pm 0.006$ | $0.423 \pm 0.004$ | $0.256 \pm 0.006$ | $0.293 \pm 0.006$ | $0.336 \pm 0.010$ |
| Qwen2.5-32B-Inst | $0.397 \pm 0.008$ | $\mathbf{0.248 \pm 0.004}$ | $0.272 \pm 0.005$ | $0.329 \pm 0.008$ | $0.261 \pm 0.005$ | $0.261 \pm 0.007$ | $0.383 \pm 0.009$ |
| Qwen2.5-72B-Inst | $0.202 \pm 0.009$ | $0.319 \pm 0.008$ | $0.358 \pm 0.010$ | $0.428 \pm 0.009$ | $0.233 \pm 0.005$ | $\mathbf{0.180 \pm 0.005}$ | $0.400 \pm 0.008$ |
| Llama3.1-405B-Inst | $\mathbf{0.173 \pm 0.003}$ | $0.278 \pm 0.009$ | $0.226 \pm 0.004$ | $0.257 \pm 0.008$ | $0.227 \pm 0.006$ | $0.308 \pm 0.006$ | $0.243 \pm 0.012$ |
| Base Quantitative Forecaster | - | $0.382 \pm 0.011$ | $0.492 \pm 0.004$ | $0.636 \pm 0.014$ | $0.382 \pm 0.011$ | $0.492 \pm 0.004$ | $0.636 \pm 0.014$ |

Table 12: Aggregate results of the hybrid method IC-CorDP on CiK, accompanied by standard errors. The best performing method for each model is in **bold**.

### D.4   Example Forecasts

#### D.4.1   Task: *ElectricityIncreaseInPredictionWithSplitContext*

**Context:**
Background: This is the electricity consumption recorded in Kilowatt (kW) in city A.
Constraints: None
Scenario: Suppose that there is a heat wave in city A from 2013-05-28 12:00:00 for 2 hours, which would typically lead to excessive use of air conditioning, and 10 times the usual electricity being consumed. But in this case, residents sought to conserve energy and used lesser air conditioning, resulting in excessive usage of only 4 times the usual electricity.

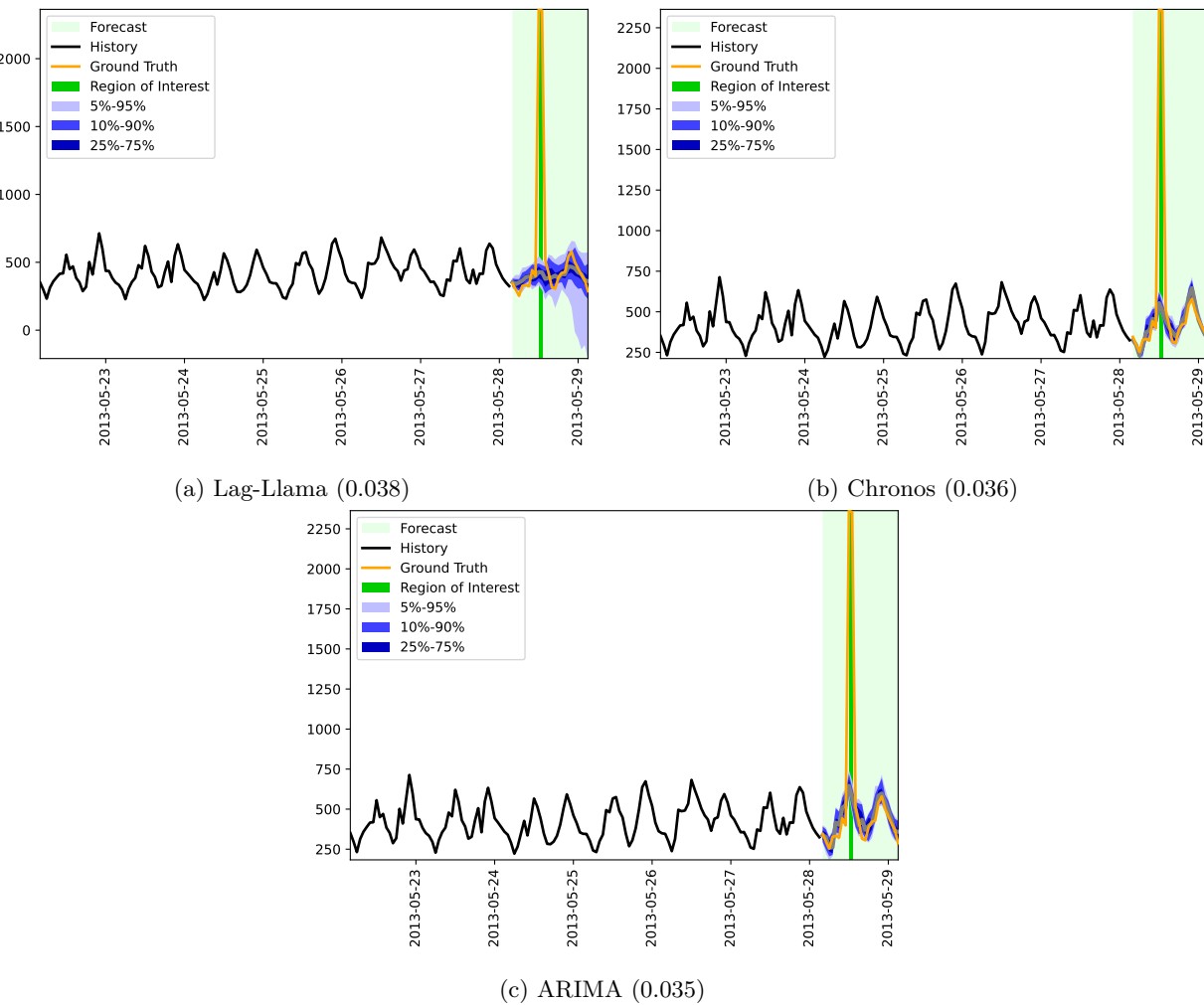

(a) Lag-Llama (0.038)

(b) Chronos (0.036)

(c) ARIMA (0.035)

Figure 45: Forecasts of Lag-Llama, Chronos, and ARIMA on the *ElectricityIncreaseInPredictionWithSplit-Context* task (with RCRPS in brackets)

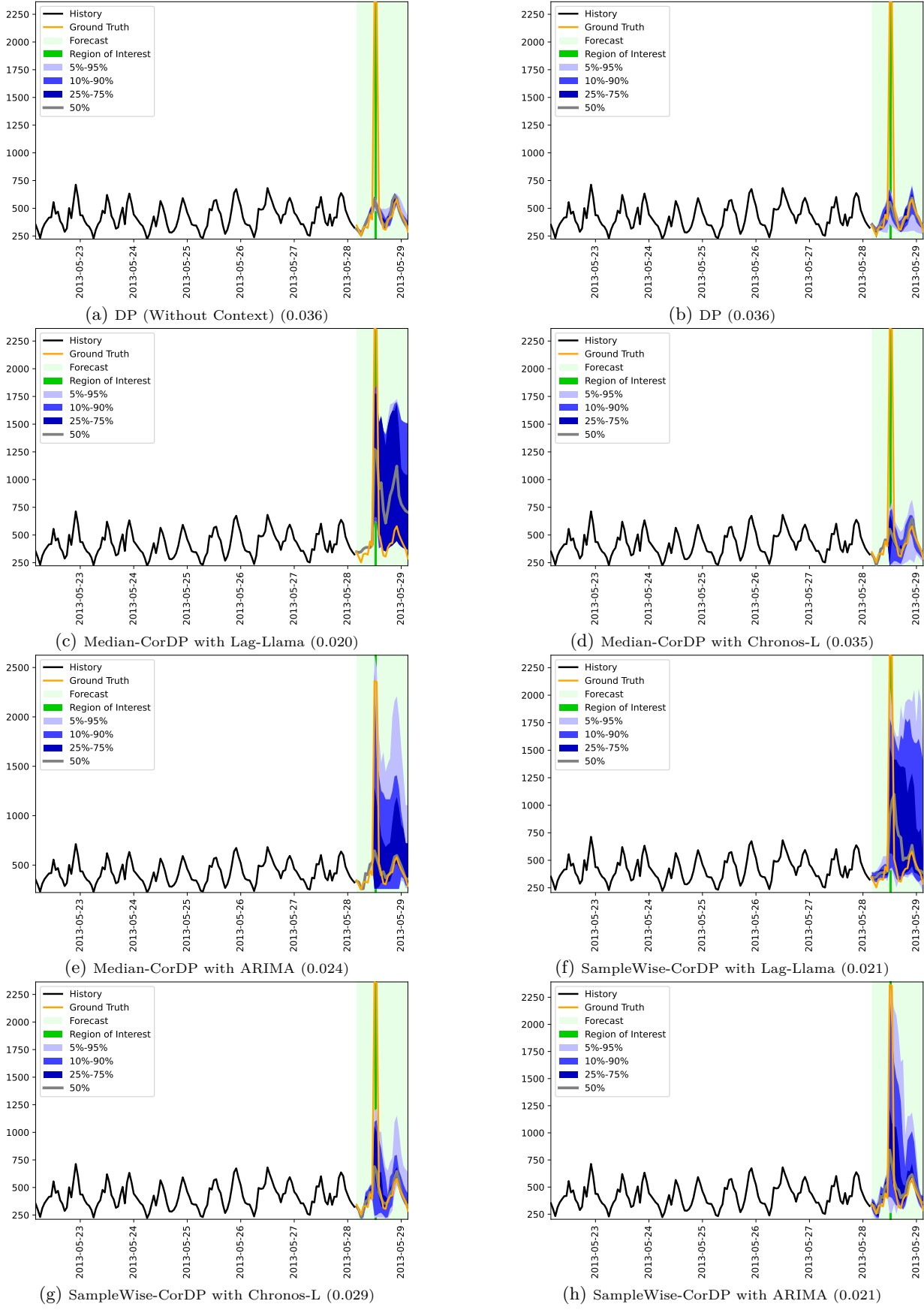

(a) DP (Without Context) (0.036)

(b) DP (0.036)

(c) Median-CorDP with Lag-Llama (0.020)

(d) Median-CorDP with Chronos-L (0.035)

(e) Median-CorDP with ARIMA (0.024)

(f) SampleWise-CorDP with Lag-Llama (0.021)

(g) SampleWise-CorDP with Chronos-L (0.029)

(h) SampleWise-CorDP with ARIMA (0.021)

Figure 46: Forecasts of model Qwen2.5-7B-Inst on task *ElectricityIncreaseInPredictionWithSplitContext* (with RCRPS in brackets)

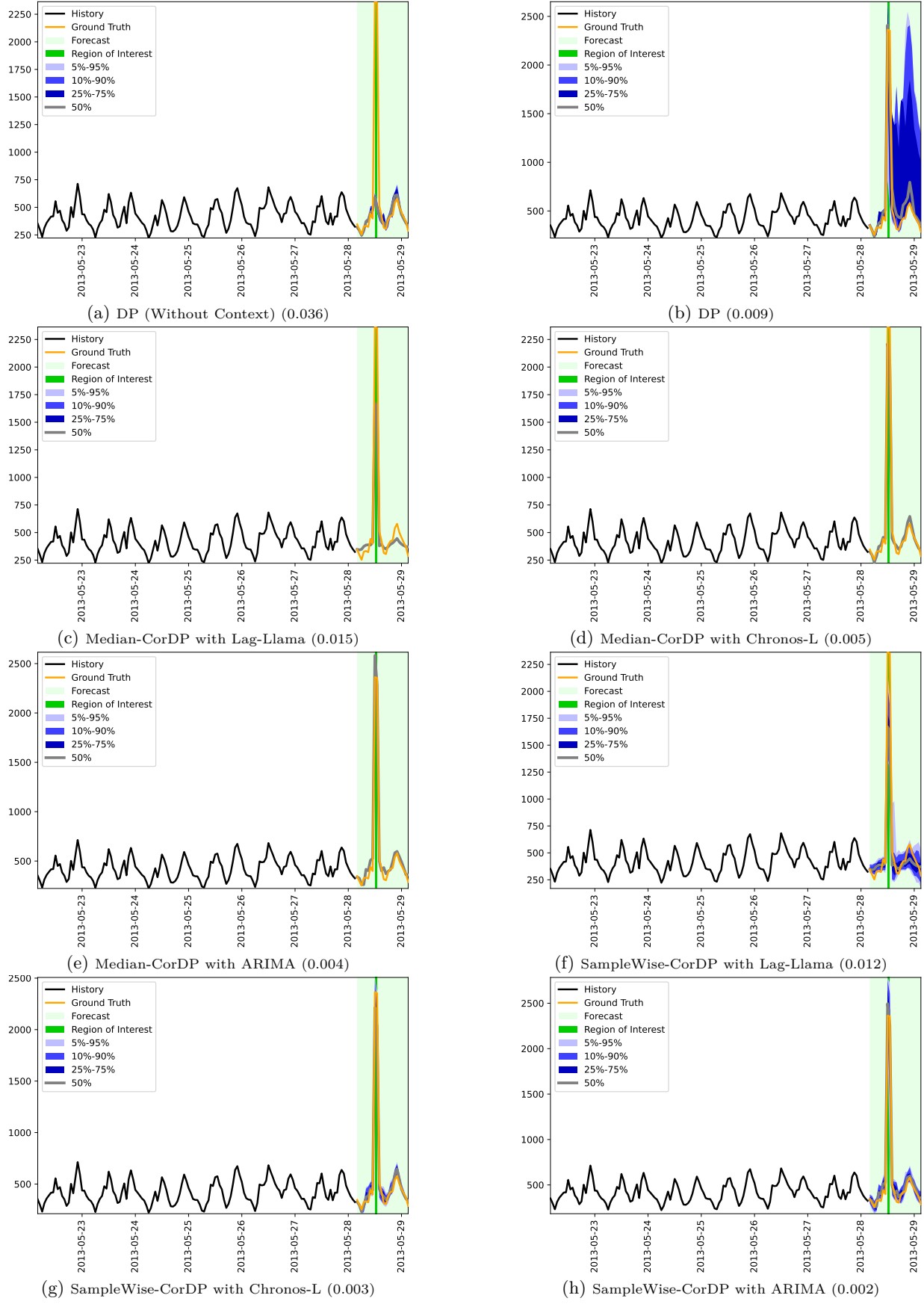

Figure 47: Forecasts of model Qwen2.5-32B-Inst on task *ElectricityIncreaseInPredictionWithSplitContext* (with RCRPS in brackets)

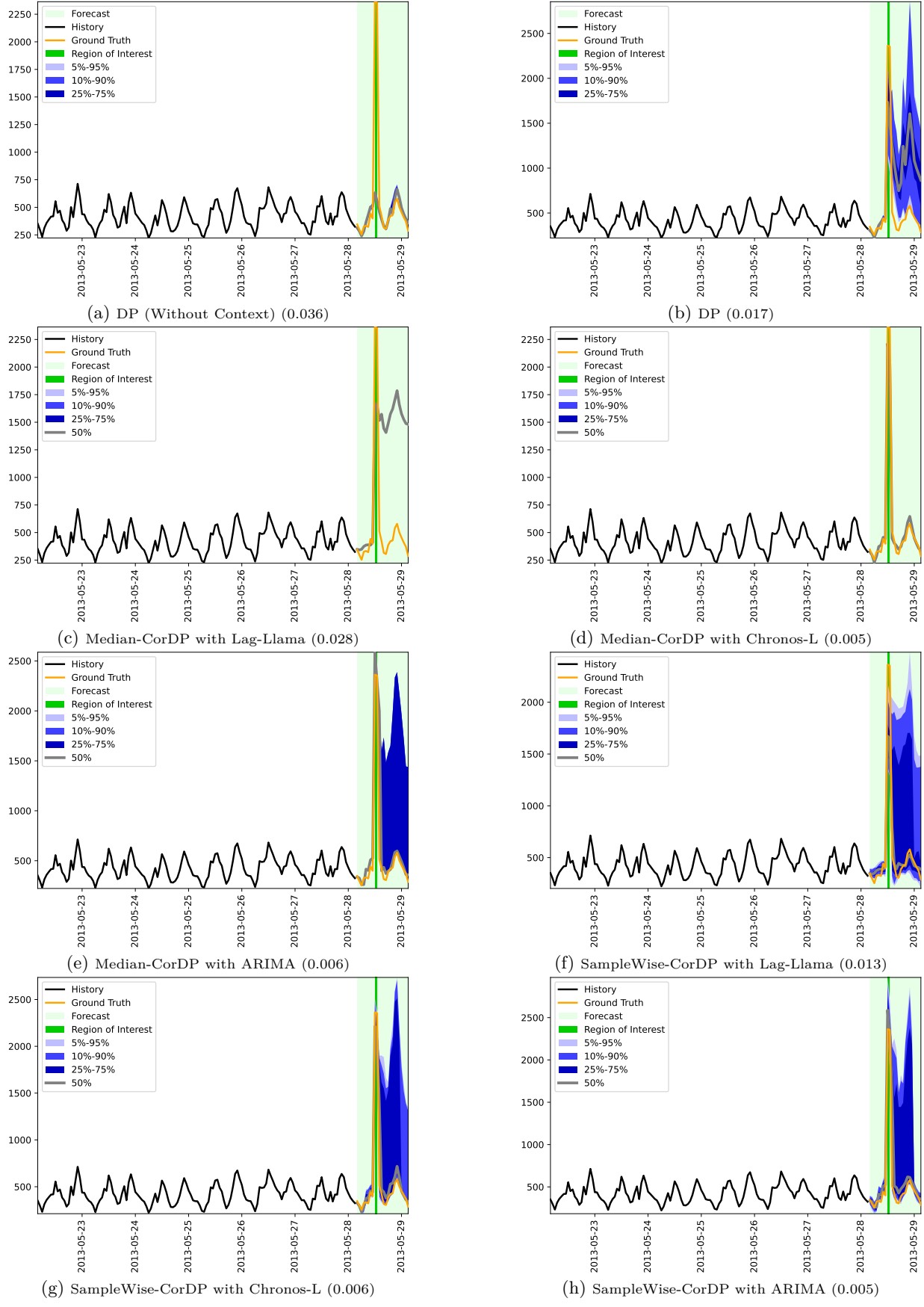

Figure 48: Forecasts of model Llama3.3-70B-Inst on task *ElectricityIncreaseInPredictionWithSplitContext* (with RCRPS in brackets)

**D.4.2  Task: *IncreasedWithdrawalScenario***

**Context:**
Background: This is the number of cash withdrawals from an automated teller machine (ATM) in an arbitrary location in England.
Constraints: None
Scenario: Suppose that there is a carnival from 1996-11-22 00:00:00, for 11 days leading to more people in the area, and 4 times the number of usual withdrawals during that period.

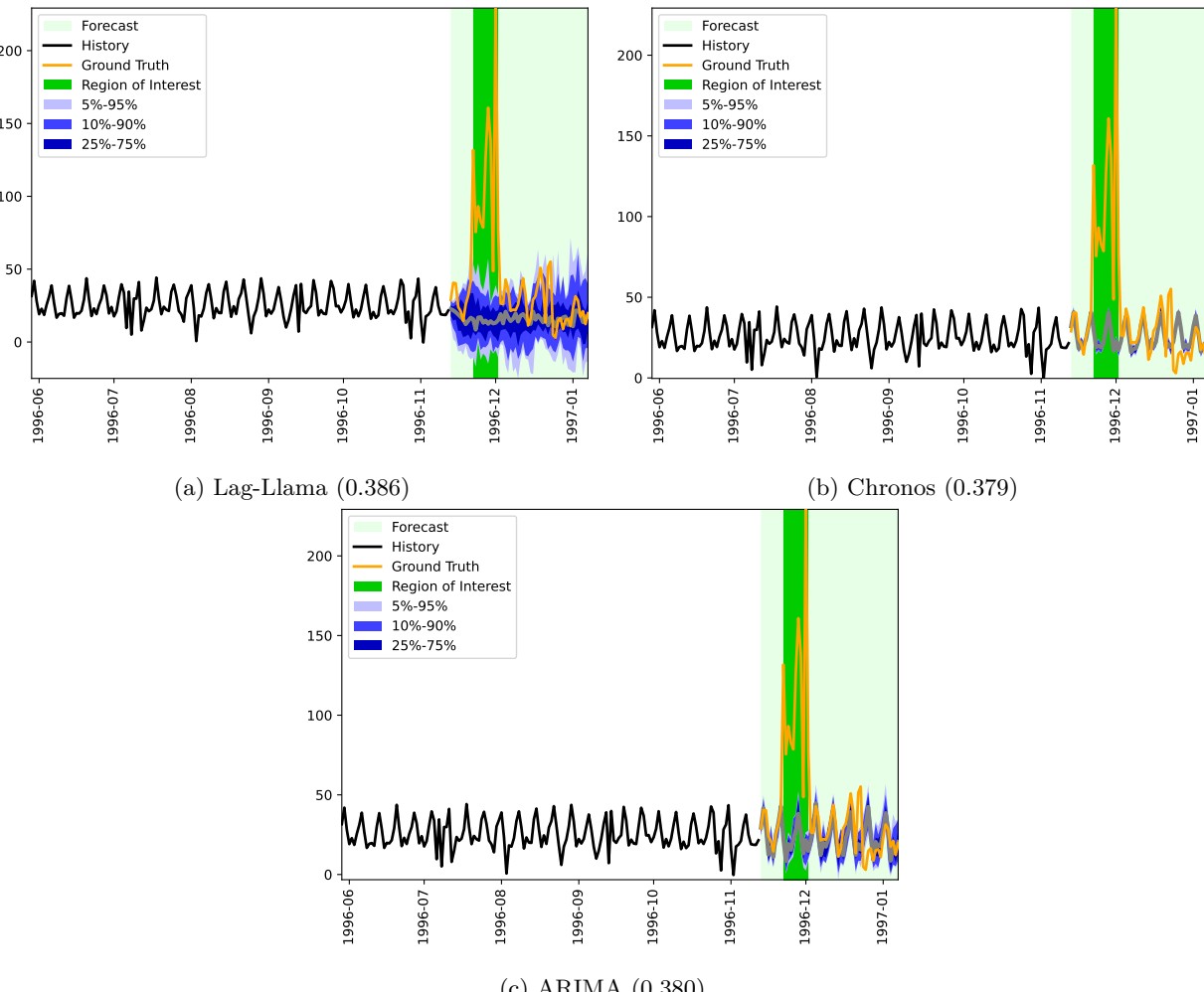

(a) Lag-Llama (0.386)  (b) Chronos (0.379)

(c) ARIMA (0.380)

Figure 49: Forecasts of Lag-Llama, Chronos, and ARIMA on the *IncreasedWithdrawalScenario* task (with RCRPS in brackets)

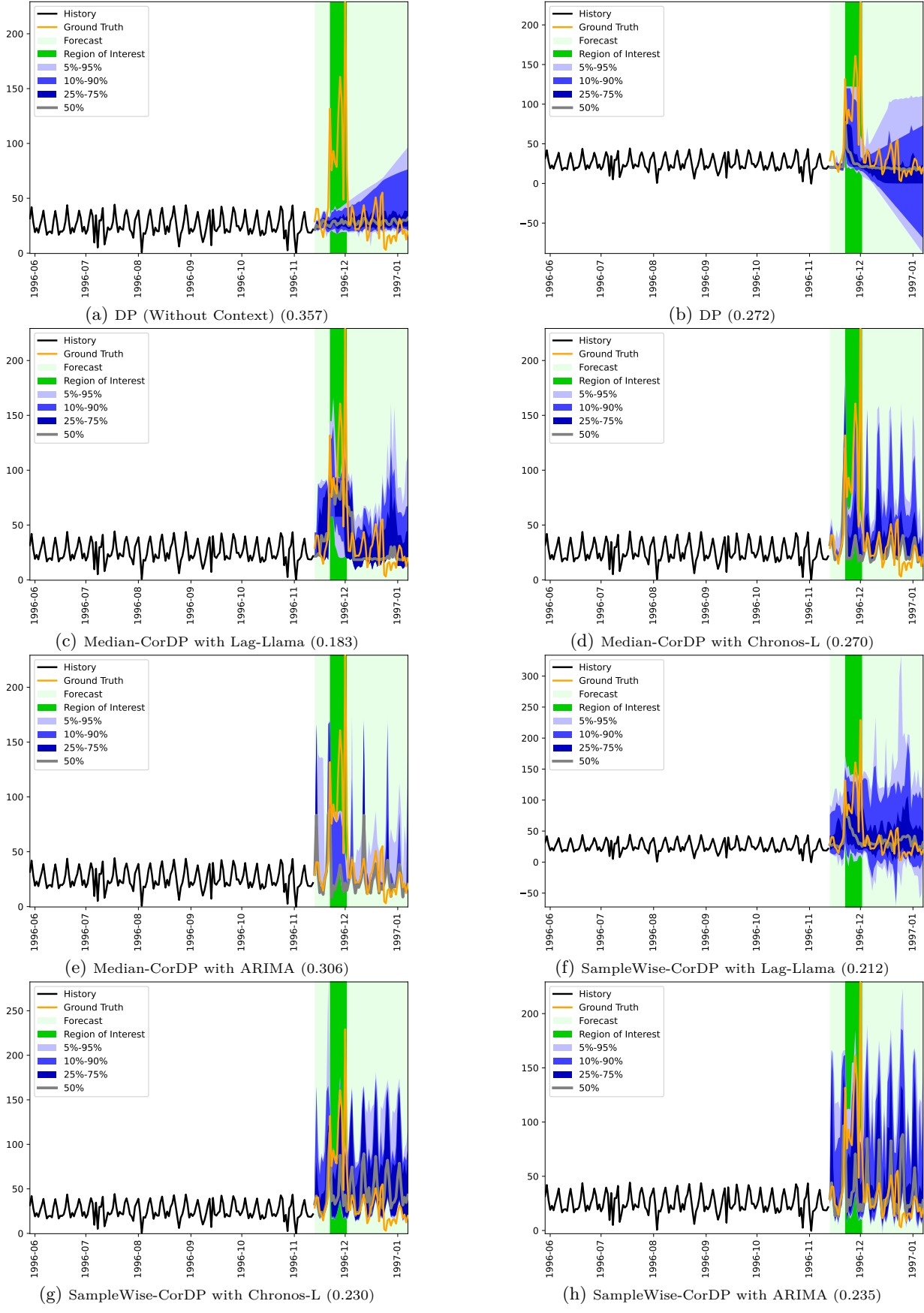

Figure 50: Forecasts of model Qwen2.5-7B-Inst on task *IncreasedWithdrawalScenario* (with RCRPS in brackets)

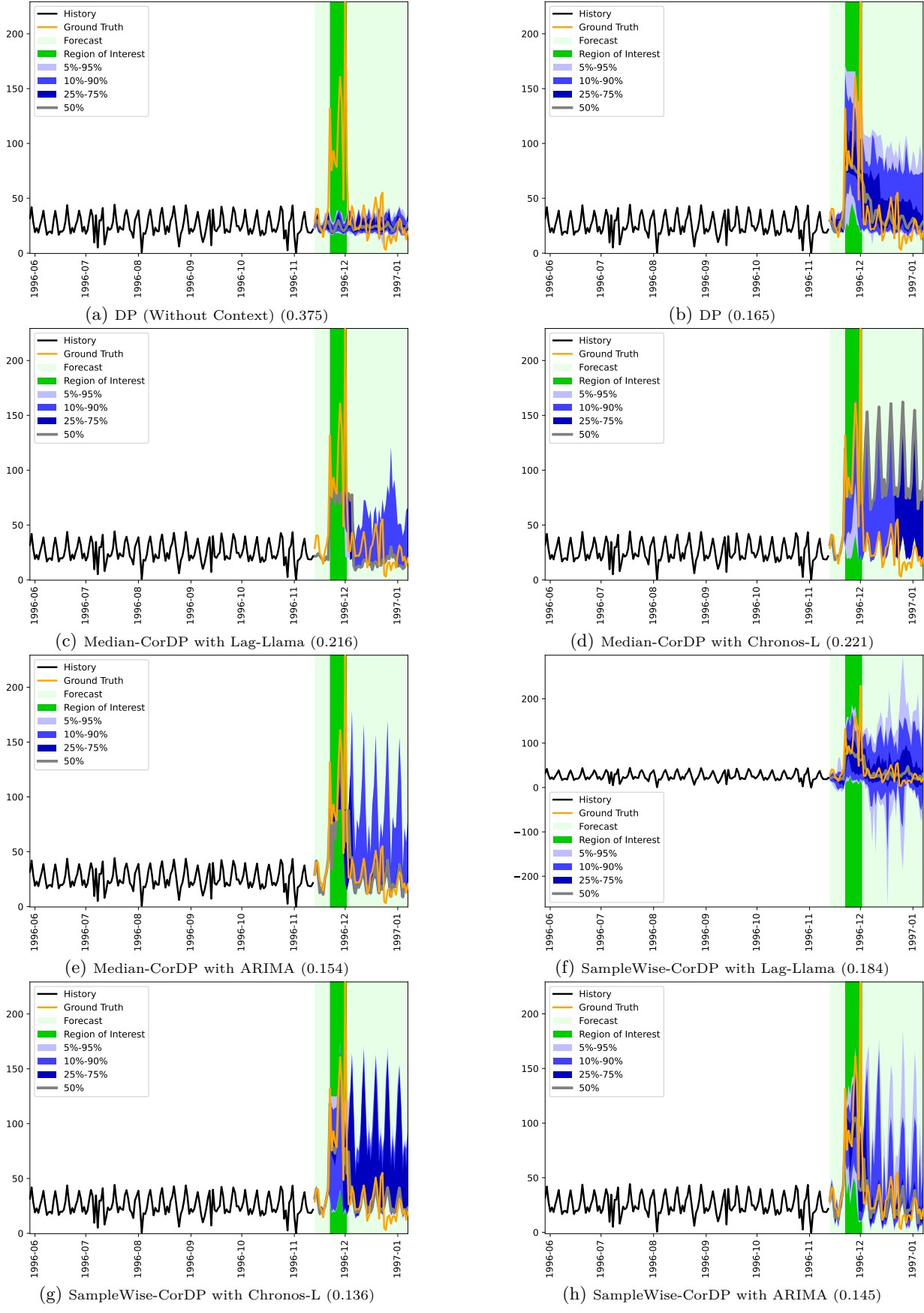

Figure 51: Forecasts of model Qwen2.5-32B-Inst on task *IncreasedWithdrawalScenario* (with RCRPS in brackets)

### D.4.3 Task: *ATMBuildingClosed*

> **Context:**
> Background: This is the number of cash withdrawals from an automated teller machine (ATM) in an arbitrary location in England.
> Constraints: None
> Scenario: Consider that the building which contains the ATM is closed from 1996-11-24 00:00:00, for 10 days.

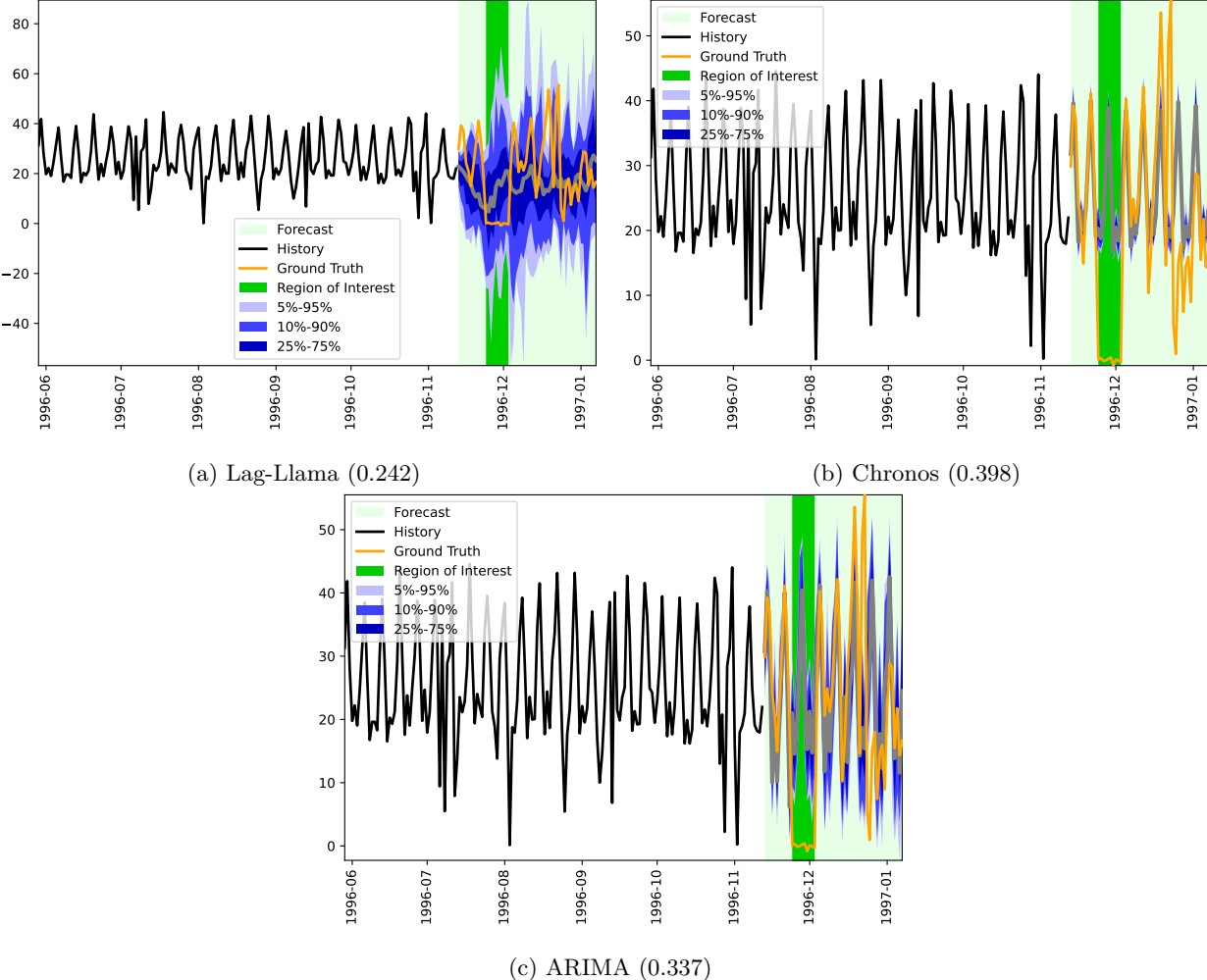

(a) Lag-Llama (0.242)  (b) Chronos (0.398)

(c) ARIMA (0.337)

Figure 52: Forecasts of Lag-Llama, Chronos, and ARIMA on the *ATMBuildingClosedTask* task (with RCRPS in brackets)

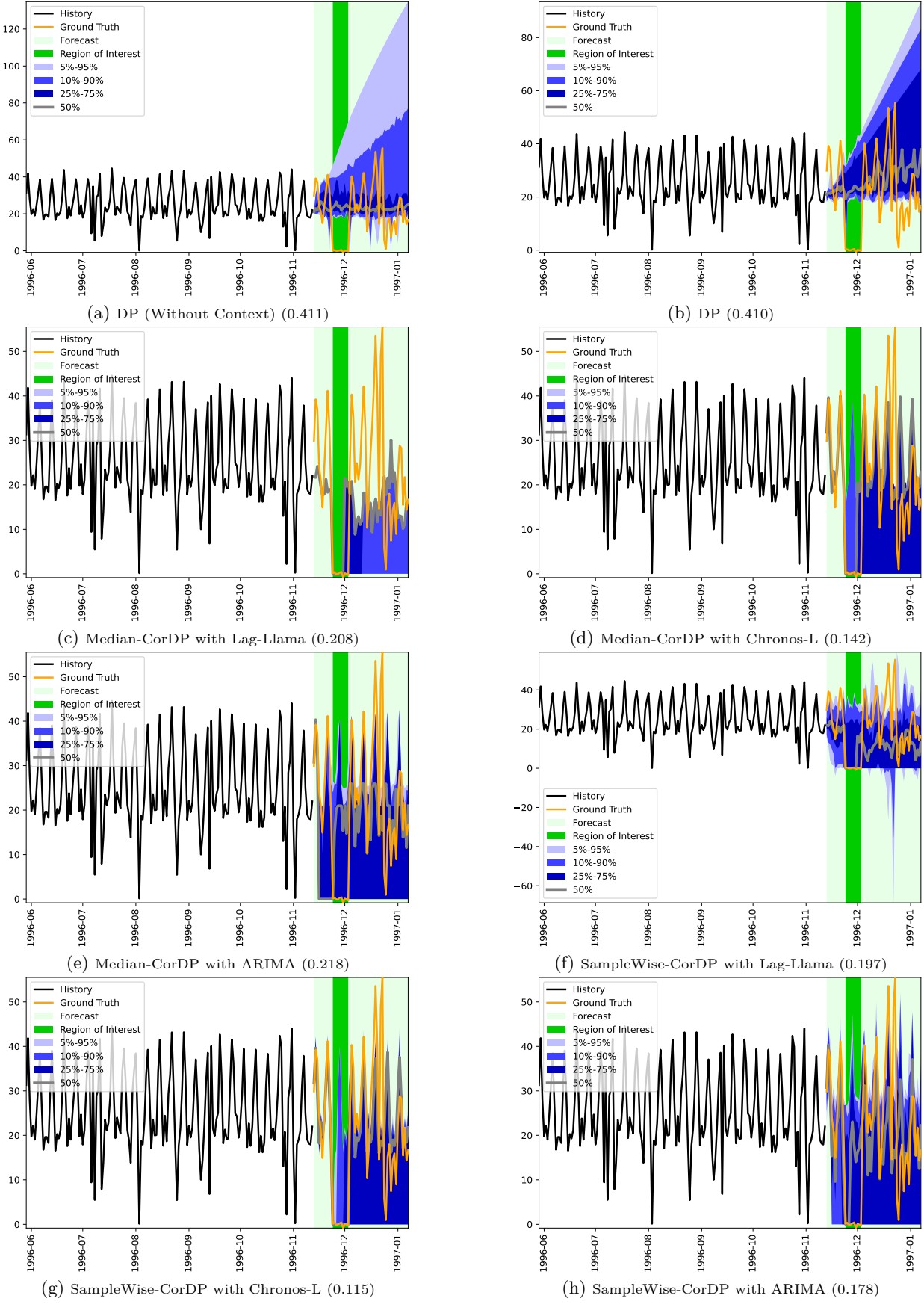

Figure 53: Forecasts of model Qwen2.5-7B-Inst on task *ATMBuildingClosedTask* (with RCRPS in brackets)

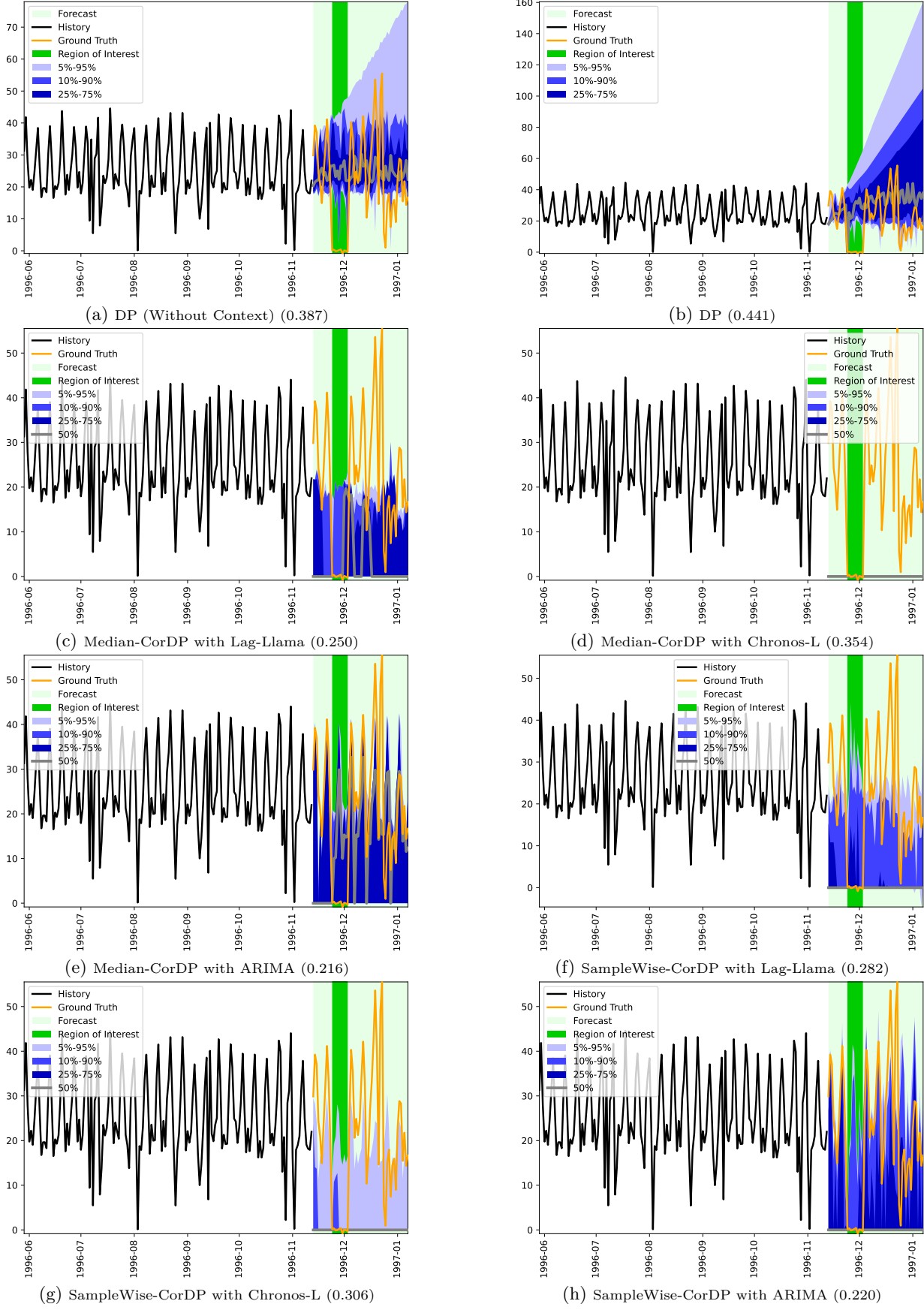

Figure 54: Forecasts of model Llama3-8B-Inst on task *ATMBuildingClosed* (with RCRPS in brackets)

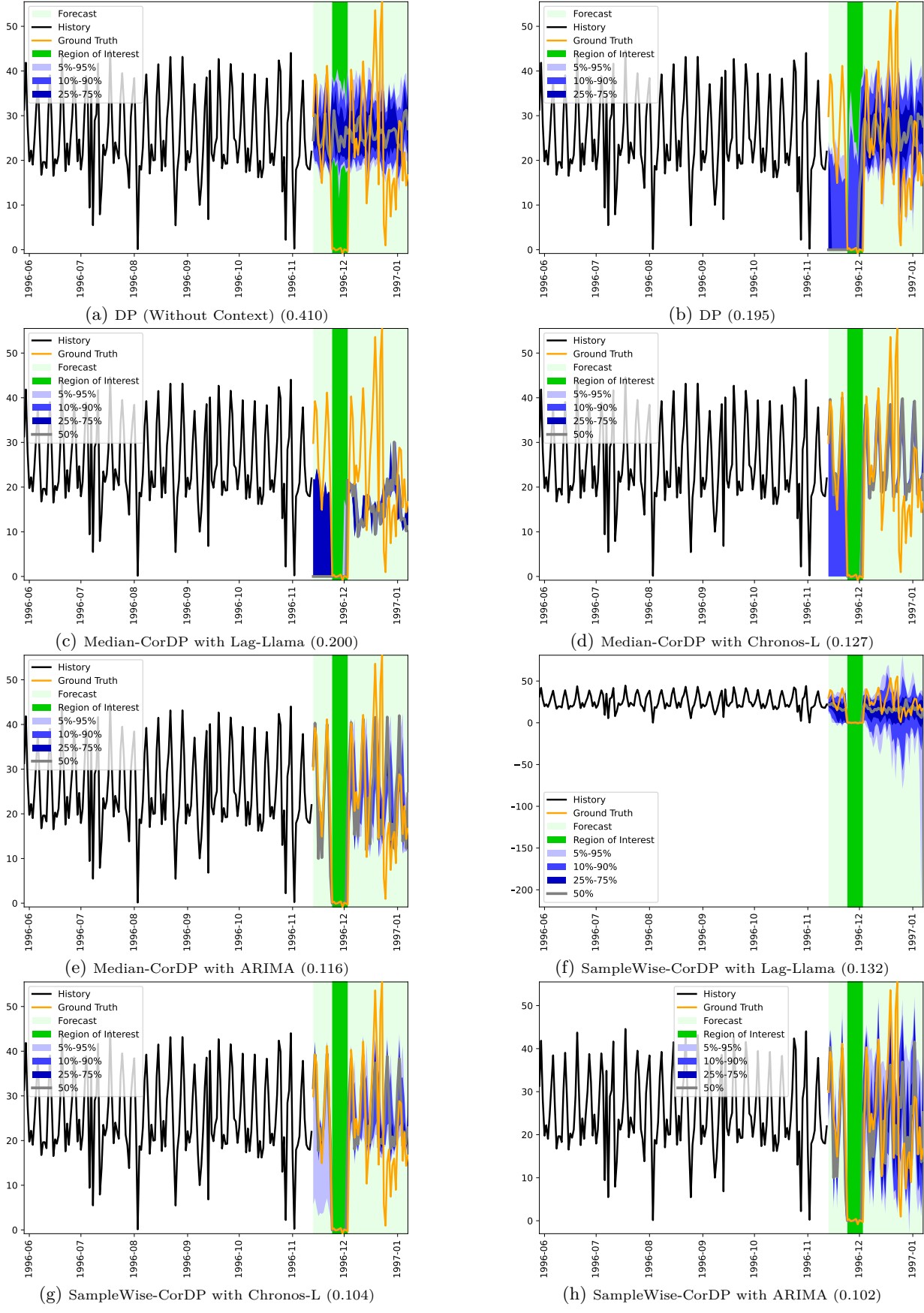

Figure 55: Forecasts of model Llama3.1-405B-Inst on task *ATMBuildingClosed* (with RCRPS in brackets)

### D.4.4 Task: *ZenithInfoHalfDaySolarForecastTask*

---

**Context:**
Background: This series contains the amount of sunlight (in Watts per squared meter) arriving on a horizontal surface, for a location in Florida, United States. Over the previous 90 days, the maximum sunlight happened on average at 12:25:33.
Constraints: None
Scenario: None

---

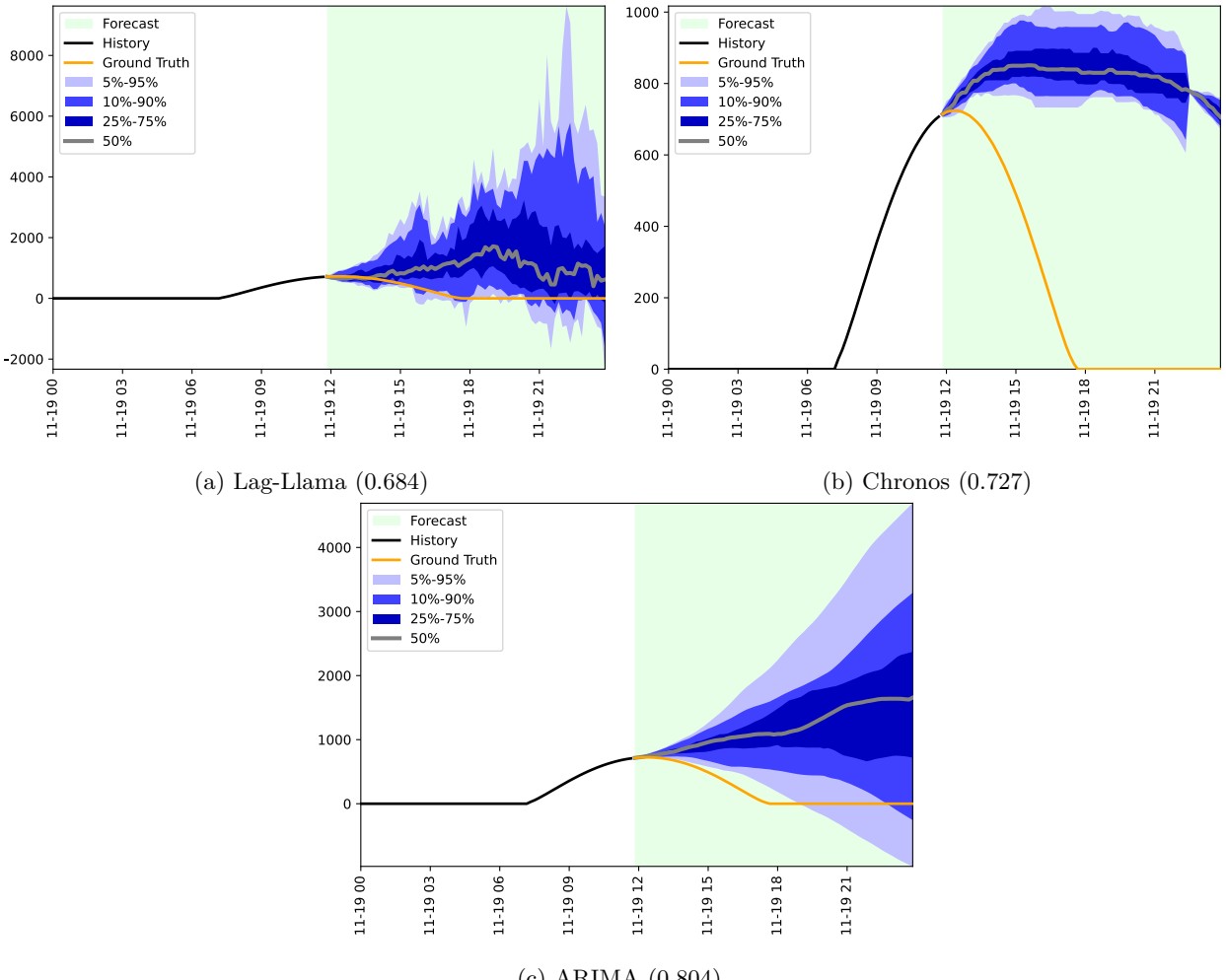

(a) Lag-Llama (0.684)

(b) Chronos (0.727)

(c) ARIMA (0.804)

Figure 56: Forecasts of Lag-Llama, Chronos, and ARIMA on the *ZenithInfoHalfDaySolarForecastTask* task (with RCRPS in brackets)

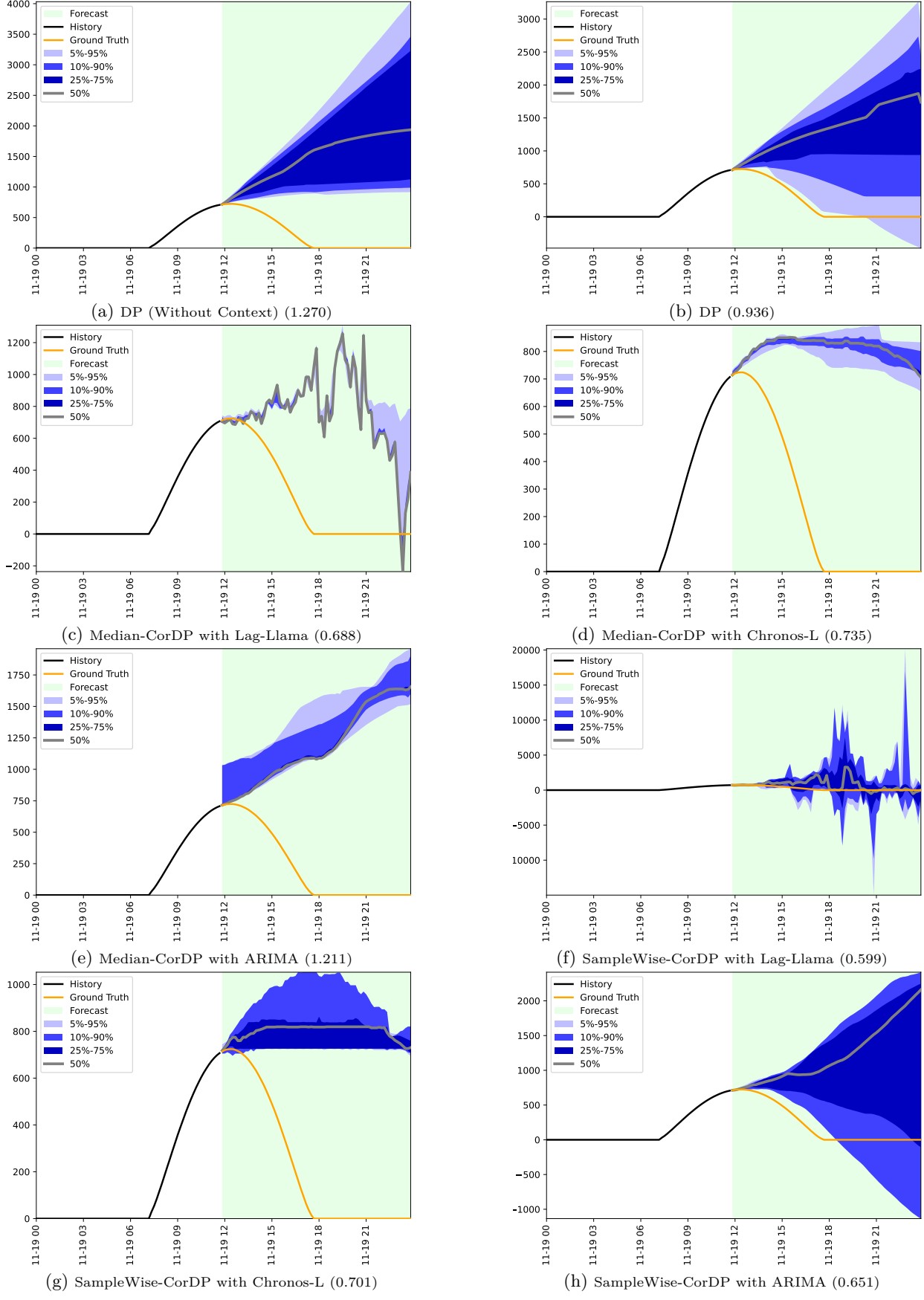

Figure 57: Forecasts of model Llama3.2-3B-Inst on task *ZenithInfoHalfDaySolarForecastTask* (with RCRPS in brackets)

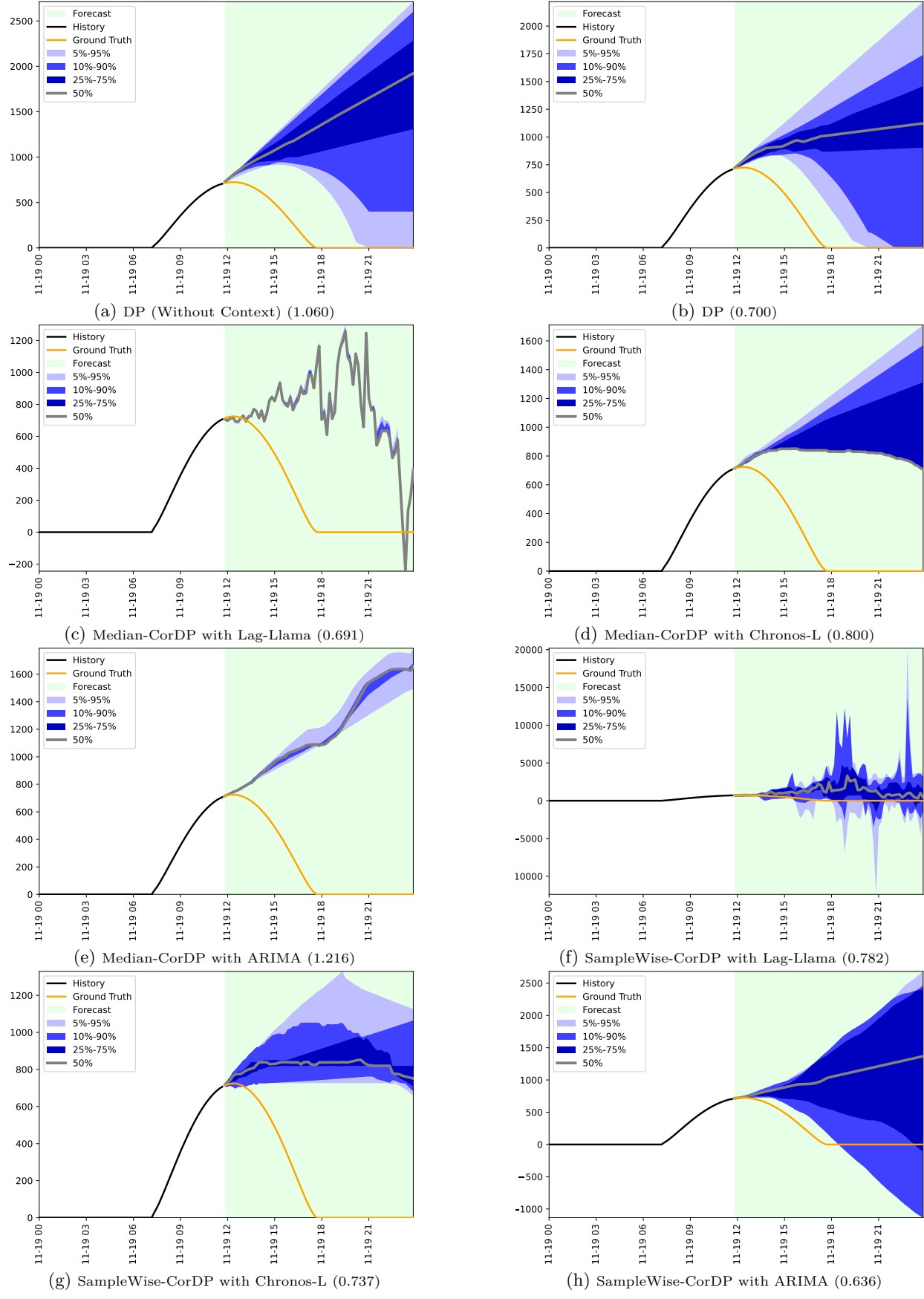

Figure 58: Forecasts of model Llama3-8B-Inst on task *ZenithInfoHalfDaySolarForecastTask* (with RCRPS in brackets)

**D.4.5  Task: *BoundedPredConstraintsBasedOnPredQuantilesTask***

**Context:**
Background: None
Constraints: Suppose that in the forecast, the values are bounded above by 6.29.
Scenario: None

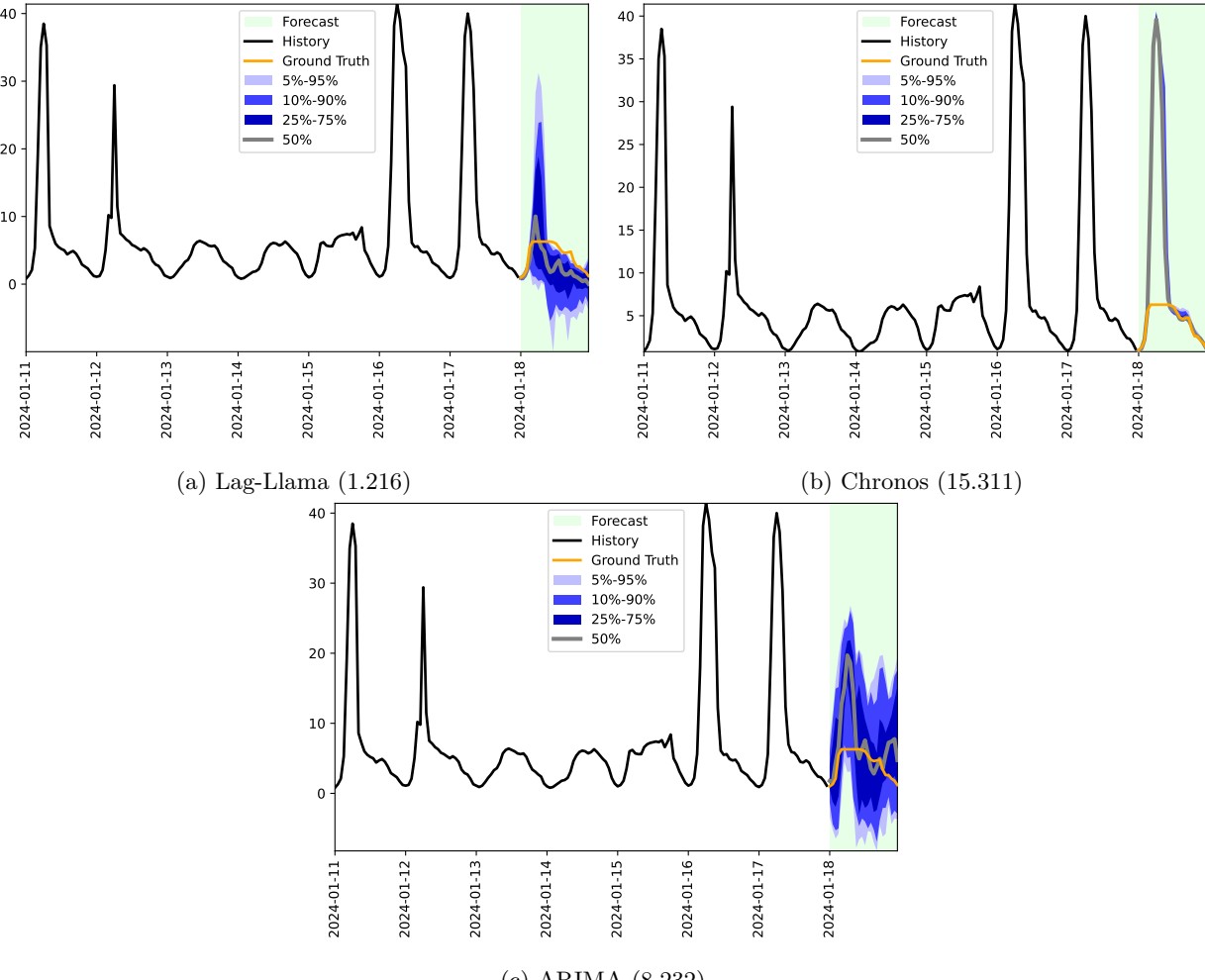

(a) Lag-Llama (1.216)

(b) Chronos (15.311)

(c) ARIMA (8.232)

Figure 59: Forecasts of Lag-Llama, Chronos, and ARIMA on the *BoundedPredConstraintsBasedOnPredQuantilesTask* task (with RCRPS in brackets)

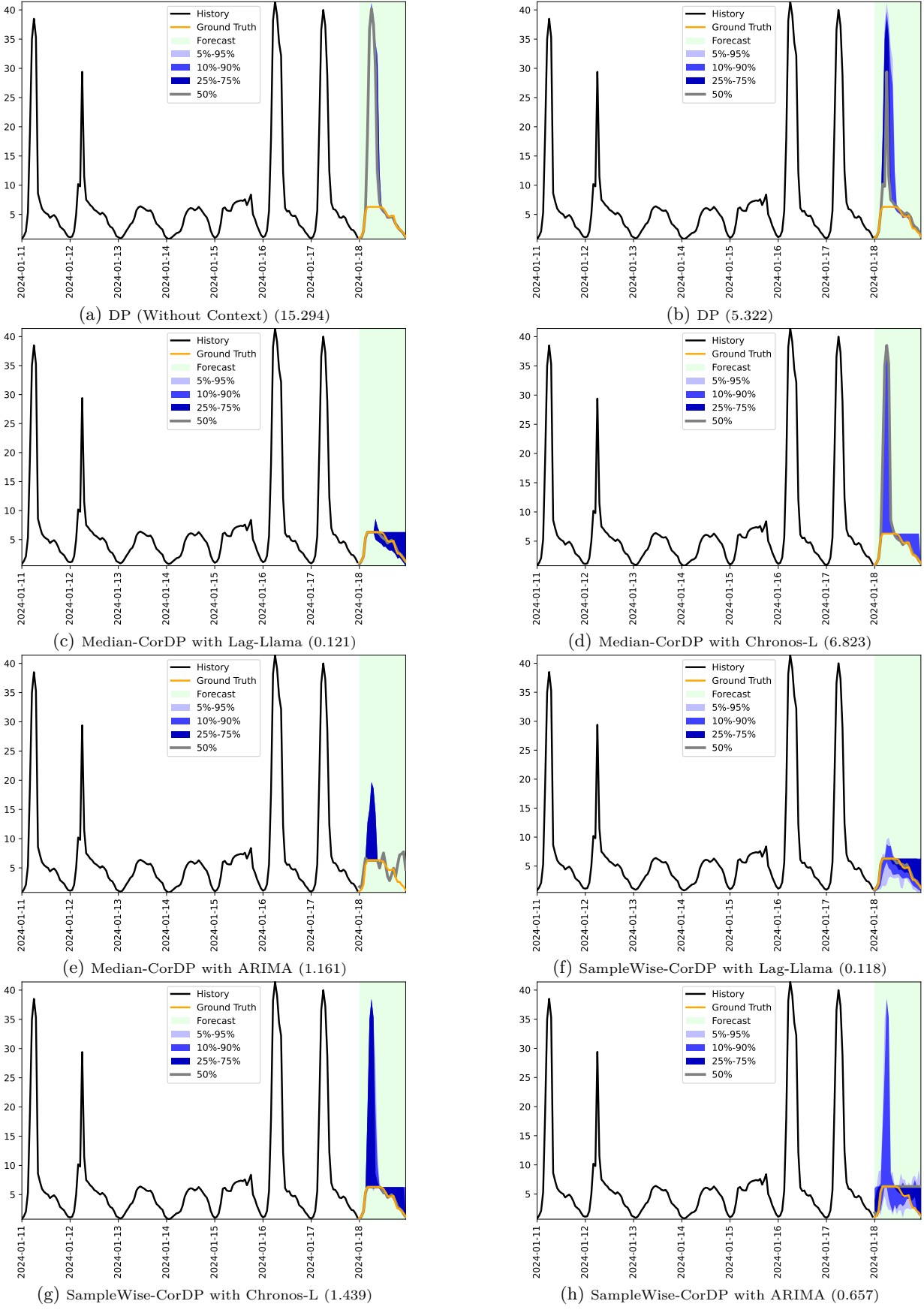

Figure 60: Forecasts of model Qwen2.5-7B-Inst on task *BoundedPredConstraintsBasedOnPredQuantilesTask* (with RCRPS in brackets)

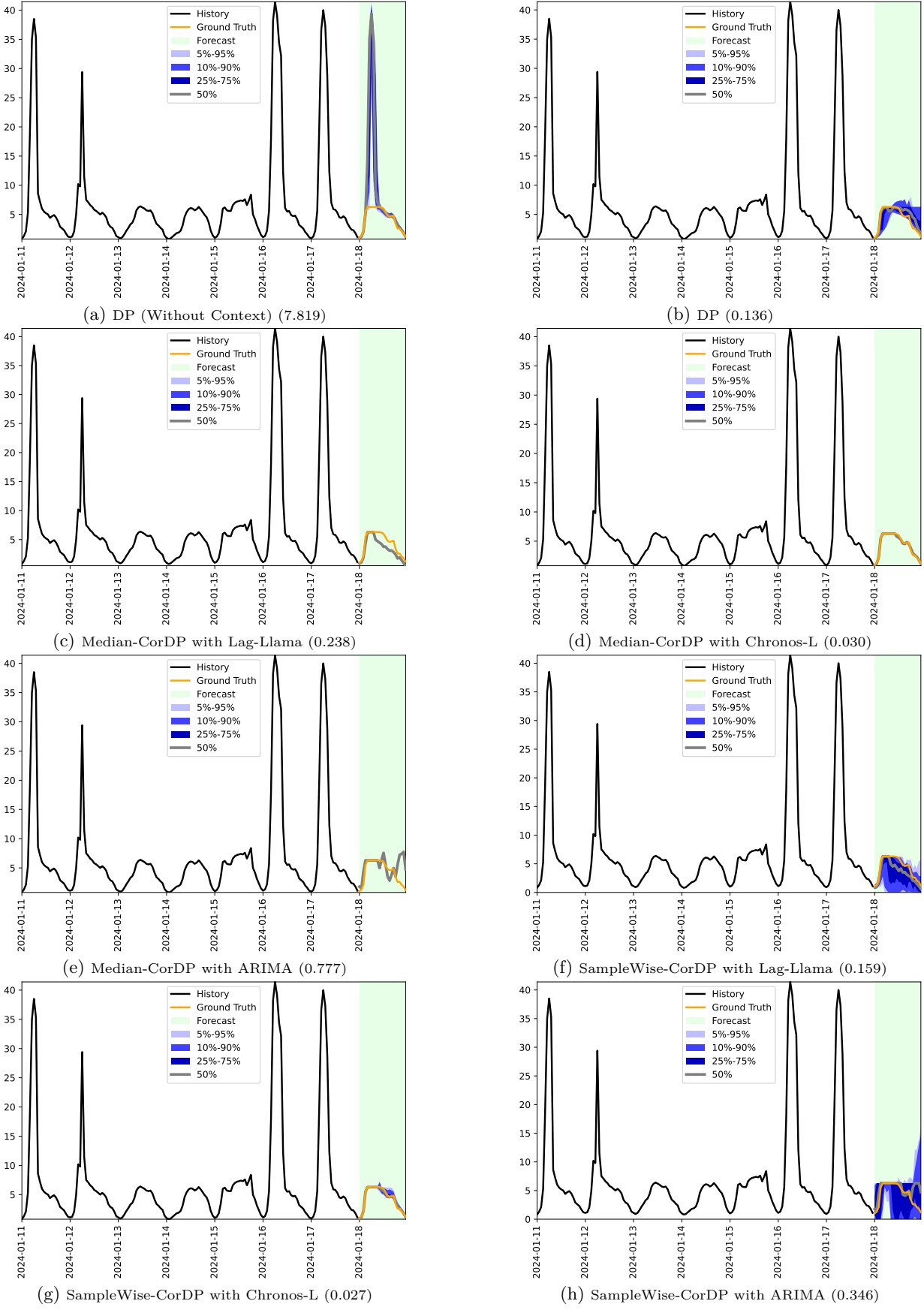

Figure 61: Forecasts of model Qwen2.5-14B-Inst on task *BoundedPredConstraintsBasedOnPredQuantilesTask* (with RCRPS in brackets)

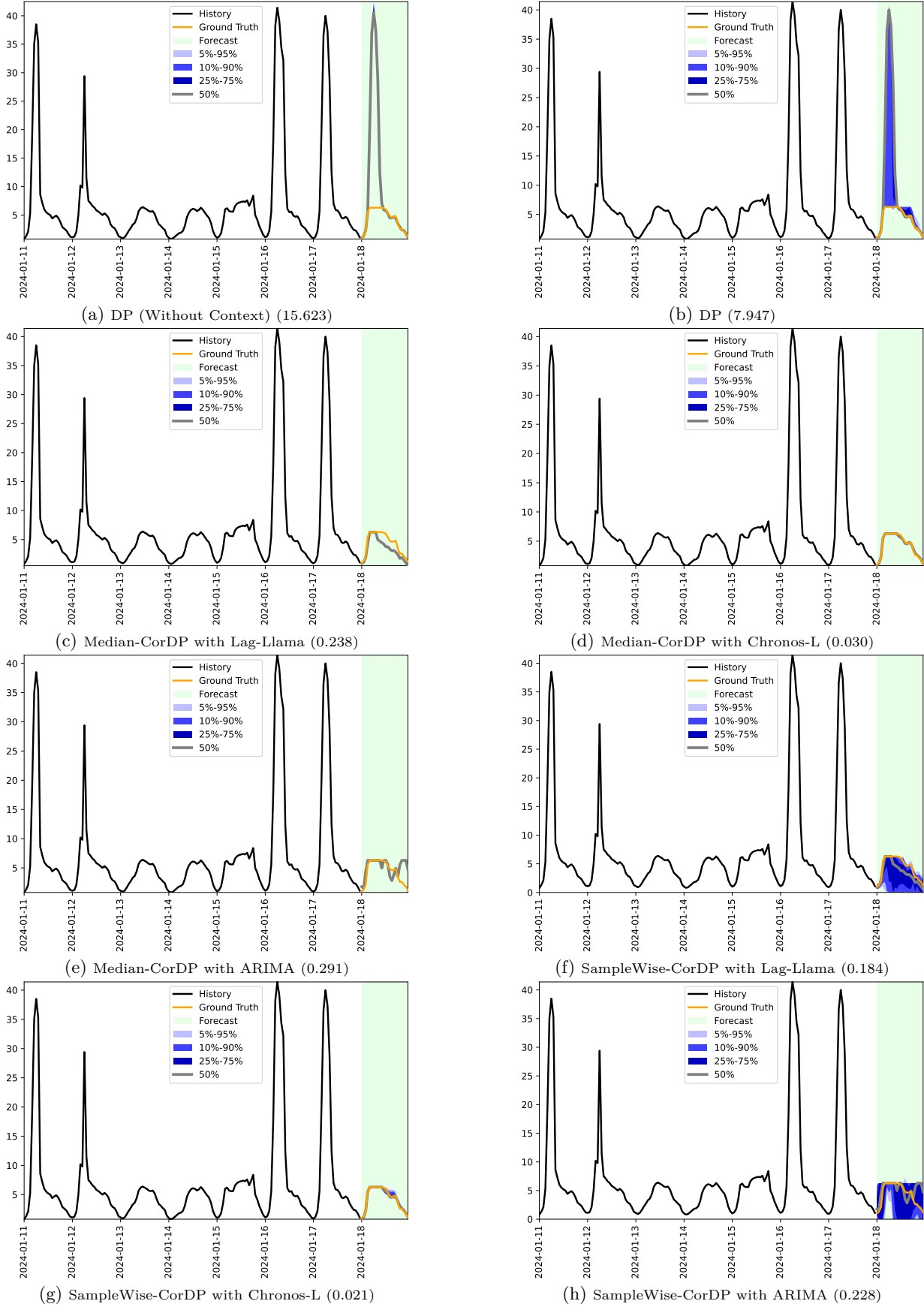

Figure 62: Forecasts of model Qwen2.5-32B-Inst on task *BoundedPredConstraintsBasedOnPredQuantilesTask* (with RCRPS in brackets)

# E   Additional Details on IC-DP

## E.1   IC-DP Prompt

We use the following prompt for the IC-DP method, where **{example_task_instance.background}**, **{example_task_instance.constraints}**, and **{example_task_instance.scenario}** are replaced by the background, constraints and scenario portions of the context of the example task respectively. **{example_task_history}**, **{example_pred_time}** and **{example_task_future}** are replaced by the history, prediction timestamps and the ground truth future values of the examples tasks respectively. **{history}** is replaced by the respective numerical history for the task instance in the format (timestamp, value), and **{context}** is replaced by the respective textual context for the task instance, and **((pred_time))** is replaced with the prediction timesteps. Although this prompt is specialized to the CiK benchmark where contexts are made up of background, constraints and scenario parts, the prompt can be generalized to use any kind of text context.

```
I have a context-aided time series forecasting task for you, where you will be given the
    history of a time series and additional context information, and prediction timesteps for
    which a forecast is required. You are expected to factor in any background knowledge,
satisfy any constraints, and respect any scenarios given in the context, and output the
    forecast.
in (timestamp, value) format in between <forecast> and </forecast> tags. You are to not
    include any other information (e.g., comments) in the forecast.

Here is the prompt for an example task:

Here is the context:
<context>\nBackground: {example_task_instance.background}\nConstraints:
    {example_task_instance.constraints}\nScenario:
    {example_task_instance.scenario}\n\n</context>\n\nHere is a historical time series in
    (timestamp, value) format:\n<history>{example_task_history}</history>\n\nNow please
    predict the value at the following timestamps: {example_pred_time}.\n

The expected output would be:
<forecast>{example_task_future}</forecast>

Note how the context was incorporated in the forecast. You are expected to do the same.
Here is the problem for which you need to return a forecast:

Here is some context about the task.
<context>
{context}
</context>

Here is a historical time series in (timestamp, value) format:
<history>
{history}
</history>

Now please predict the value at the following timestamps: {pred_time}.

Return the forecast in (timestamp, value) format in between <forecast> and </forecast> tags.
Do not include any other information (e.g., comments) in the forecast.
```

### E.2  Aggregate Results

Table 13 displays the aggregate results of models on CiK, comparing IC-DP and DP.

| Model | DP | IC-DP |
|---|---|---|
| Llama3.2-1B-Inst | $0.396 \pm 0.027$ | $\mathbf{0.337 \pm 0.009}$ |
| Llama3.2-3B-Inst | $0.687 \pm 0.025$ | $\mathbf{0.476 \pm 0.018}$ |
| Qwen2.5-0.5B-Inst | $0.592 \pm 0.027$ | $\mathbf{0.305 \pm 0.006}$ |
| Qwen2.5-1.5B-Inst | $0.616 \pm 0.018$ | $\mathbf{0.273 \pm 0.008}$ |
| Qwen2.5-3B-Inst | $0.424 \pm 0.017$ | $\mathbf{0.298 \pm 0.011}$ |
| Qwen2.5-7B-Inst | $0.401 \pm 0.006$ | $\mathbf{0.264 \pm 0.012}$ |
| Qwen2.5-14B-Inst | $\mathbf{0.247 \pm 0.006}$ | $0.270 \pm 0.005$ |
| Qwen2.5-32B-Inst | $0.397 \pm 0.008$ | $\mathbf{0.245 \pm 0.027}$ |
| Qwen2.5-72B-Inst | $0.202 \pm 0.009$ | $\mathbf{0.180 \pm 0.014}$ |
| Llama3.3-70B-Inst | $0.230 \pm 0.006$ | $\mathbf{0.168 \pm 0.006}$ |
| Llama3.1-405B-Inst | $0.173 \pm 0.003$ | $\mathbf{0.129 \pm 0.004}$ |
| GPT-4o | $0.317 \pm 0.009$ | $\mathbf{0.164 \pm 0.005}$ |
| GPT-4o-mini | $0.389 \pm 0.010$ | $\mathbf{0.253 \pm 0.004}$ |

Table 13: Results of models with IC-DP on CiK. The best-performing method with each model is in **bold**.

### E.3  Example Forecasts

### E.3.1 Task: *ElectricityIncreaseInPredictionWithDistractorWithDates*

Background: This is the electricity consumption recorded in Kilowatt (kW) in city A.
Constraints: None.
Scenario: There was a festival in neighbouring cities B and C that resulted in 10 times the usual electricity being consumed there from 2013-05-28 12:00:00 for 2 hours. But this did not affect electricity consumption in city A. Suppose that there is a heat wave in city A from 2013-05-28 12:00:00 for 2 hours, leading to excessive use of air conditioning, and 4 times the usual electricity being consumed.

Figure 63: Context

Background: This is the electricity consumption recorded in Kilowatt (kW) in city A.
Constraints: None
Scenario: A brief technical issue in the electricity grid in a nearby city caused a major dip of 75% from 2014-03-24 13:00:00 for 2 hours. This issue has affected many nearby cities, but not this city.Suppose that there is a heat wave in city A from 2014-03-24 13:00:00 for 2 hours, leading to excessive use of air conditioning, and 4 times the usual electricity being consumed.

(a) Context of the In-Context Example Task used with IC-DP experiments

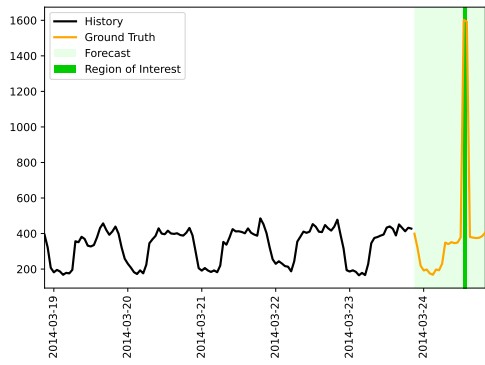

(b) Historical and Future Data of the In-Context Example Task used with IC-DP experiments

Figure 64: In-Context Example Task used with IC-DP experiments

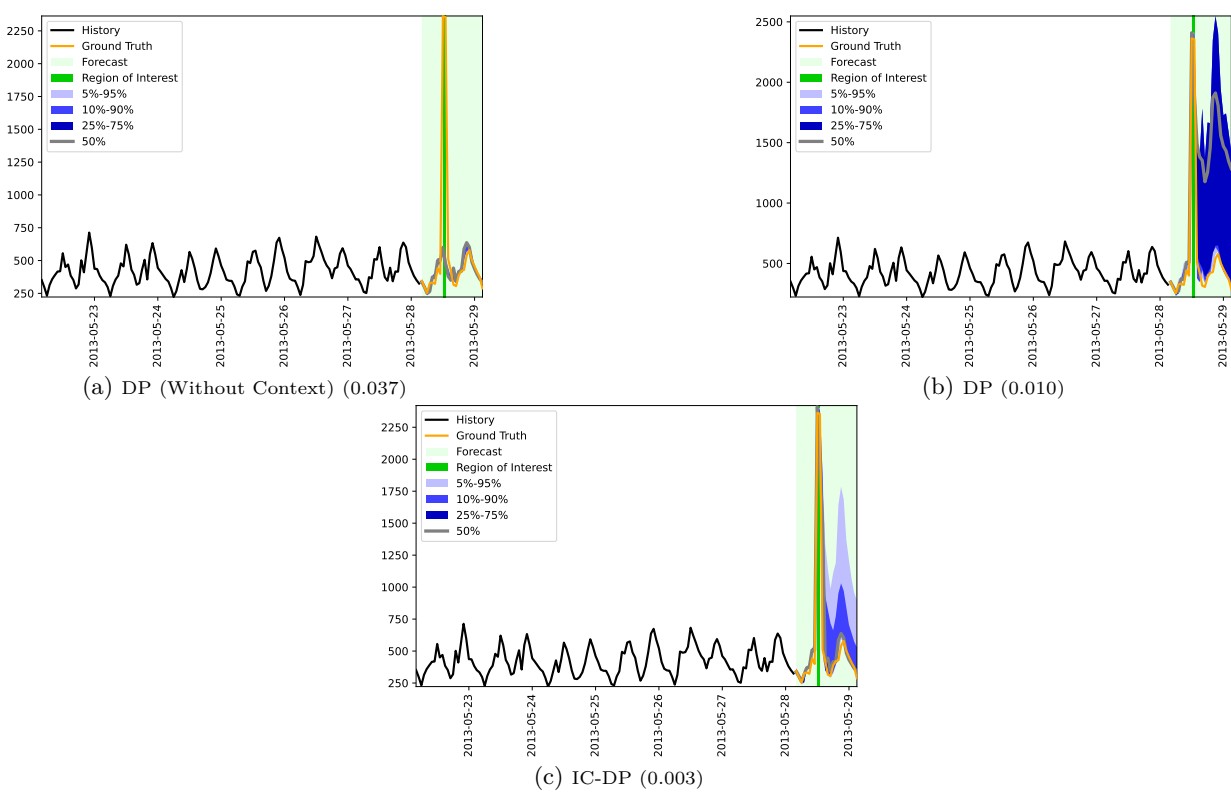

Figure 65: Forecasts of model Qwen2.5-72B-Inst on task *ElectricityIncreaseInPredictionWithDistractorWith-Dates* (with RCRPS in brackets)

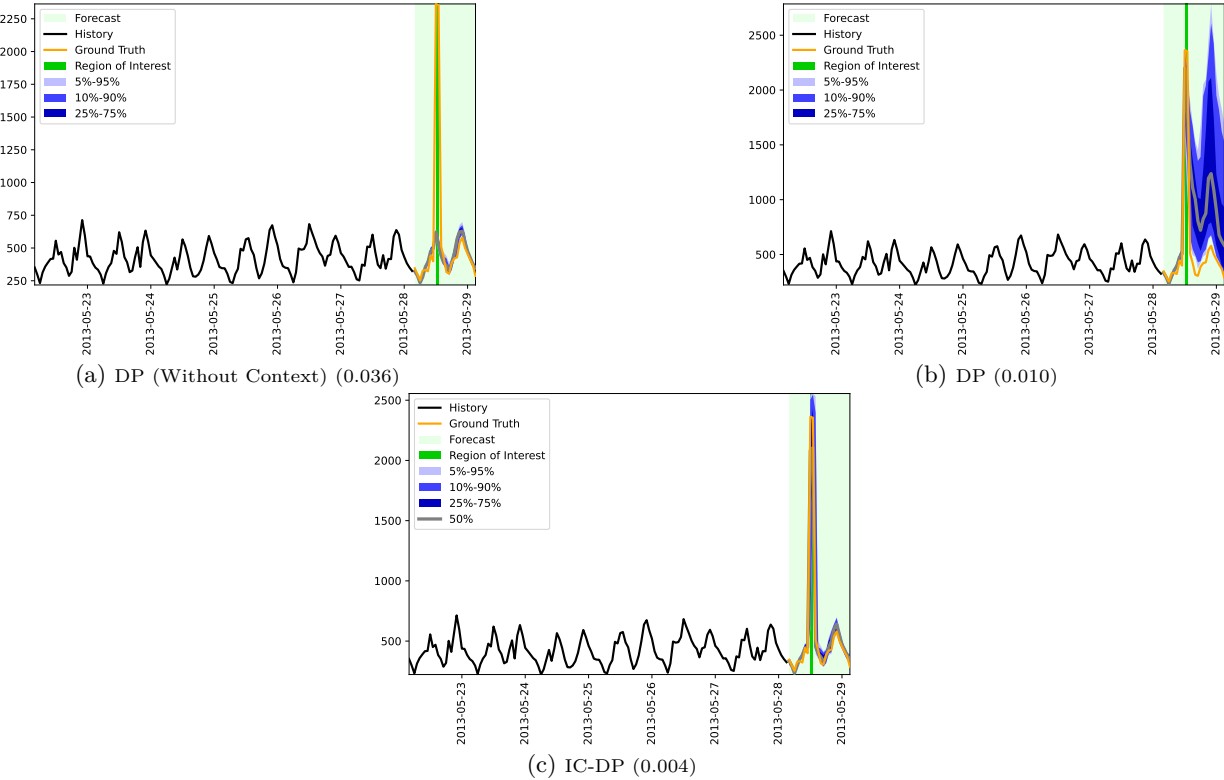

Figure 66: Forecasts of model Llama3.1-405B-Inst on task *ElectricityIncreaseInPredictionWithDistractor-WithDates* (with RCRPS in brackets)

### E.3.2 Task: *SensorTrendAccumulationTask*

Background: This series represents the occupancy rate (%) captured by a highway sensor. The sensor had a calibration problem starting from 2024-01-11 12:00:00 which resulted in an additive trend in the series that increases by 0.0874 at every hour. At timestep 2024-01-18 00:00:00, the sensor was repaired and this additive trend will disappear.
Constraints: None
Scenario: None

Figure 67: Context

Background: This series represents the occupancy rate (%) captured by a highway sensor. The sensor had a calibration problem starting from 2024-02-12 13:00:00 which resulted in an additive trend in the series that increases by 0.0489 at every hour. At timestep 2024-02-16 20:00:00, the sensor was repaired and this additive trend will disappear.
Constraints: None
Scenario: None

(a) Context of the In-Context Example Task used with IC-DP experiments

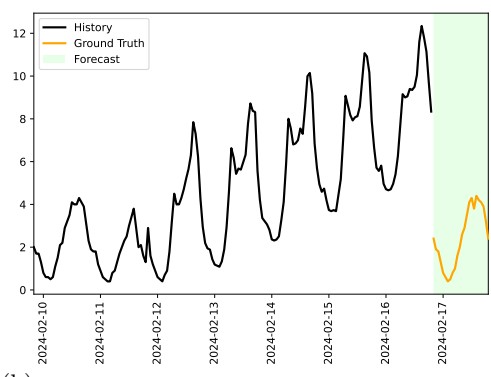

(b) Historical and Future Data of the In-Context Example Task used with IC-DP experiments

Figure 68: In-Context Example Task used with IC-DP experiments

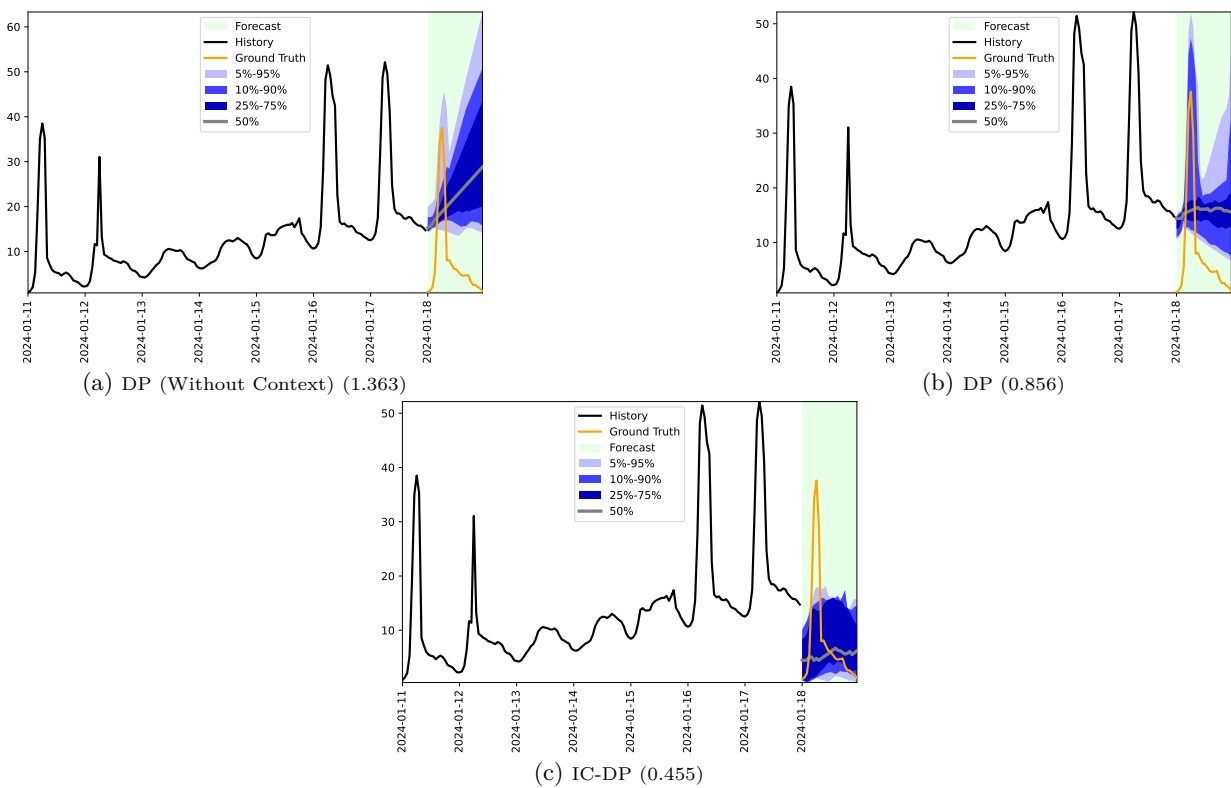

(a) DP (Without Context) (1.363)

(b) DP (0.856)

(c) IC-DP (0.455)

Figure 69: Forecasts of model Llama3-8B-Inst on task *SensorTrendAccumulationTask* (with RCRPS in brackets)

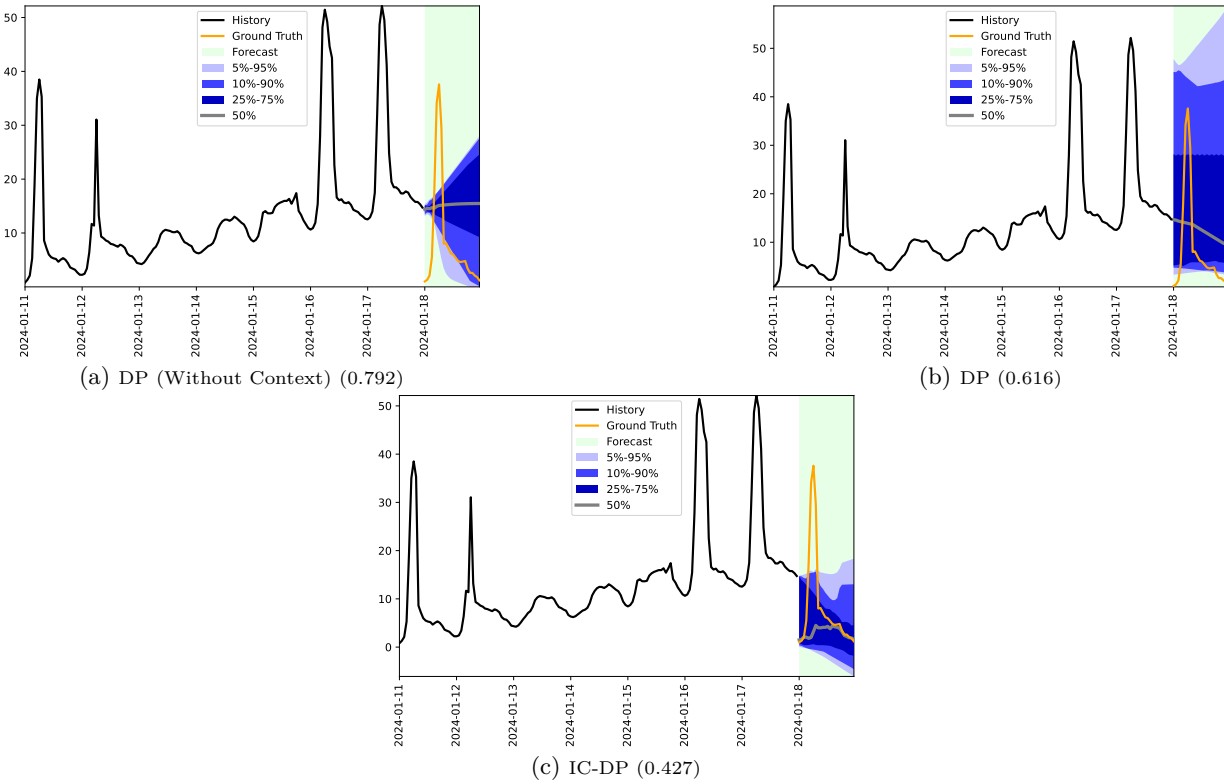

(a) DP (Without Context) (0.792)

(b) DP (0.616)

(c) IC-DP (0.427)

Figure 70: Forecasts of model Qwen2.5-3B-Inst on task *SensorTrendAccumulationTask* (with RCRPS in brackets)

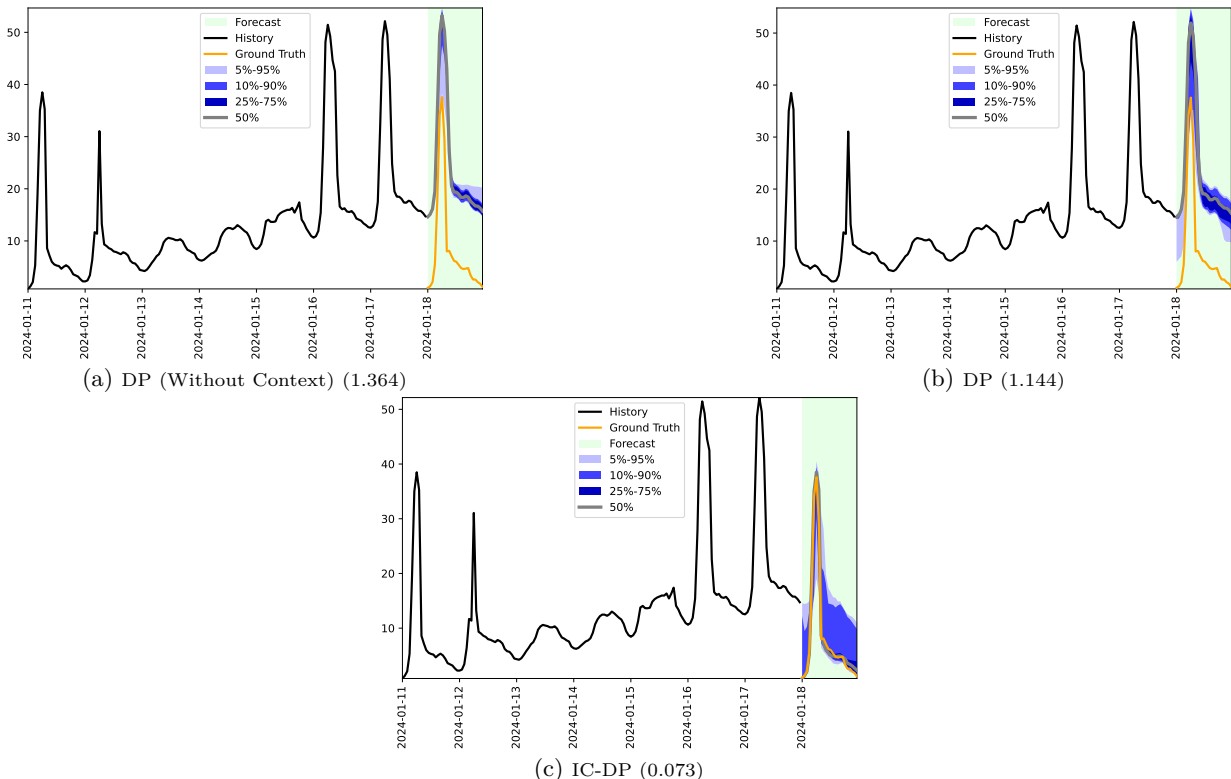

(a) DP (Without Context) (1.364)

(b) DP (1.144)

(c) IC-DP (0.073)

Figure 71: Forecasts of model Llama3.1-405B-Inst on task *SensorTrendAccumulationTask* (with RCRPS in brackets)

### E.3.3 Task: *BoundedPredConstraintsBasedOnPredQuantilesTask*

Background: None
Constraints: Suppose that in the forecast, the values are bounded above by 6.29.
Scenario: None

Figure 72: Context

Background: None
Constraints: Suppose that in the forecast, the values are bounded below by 1.57, the values are bounded above by 3.53.
Scenario: None

(a) Context of the In-Context Example Task used with IC-DP experiments

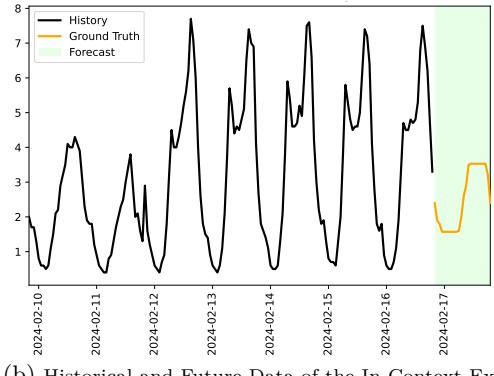

(b) Historical and Future Data of the In-Context Example Task used with IC-DP experiments

Figure 73: In-Context Example Task used with IC-DP experiments

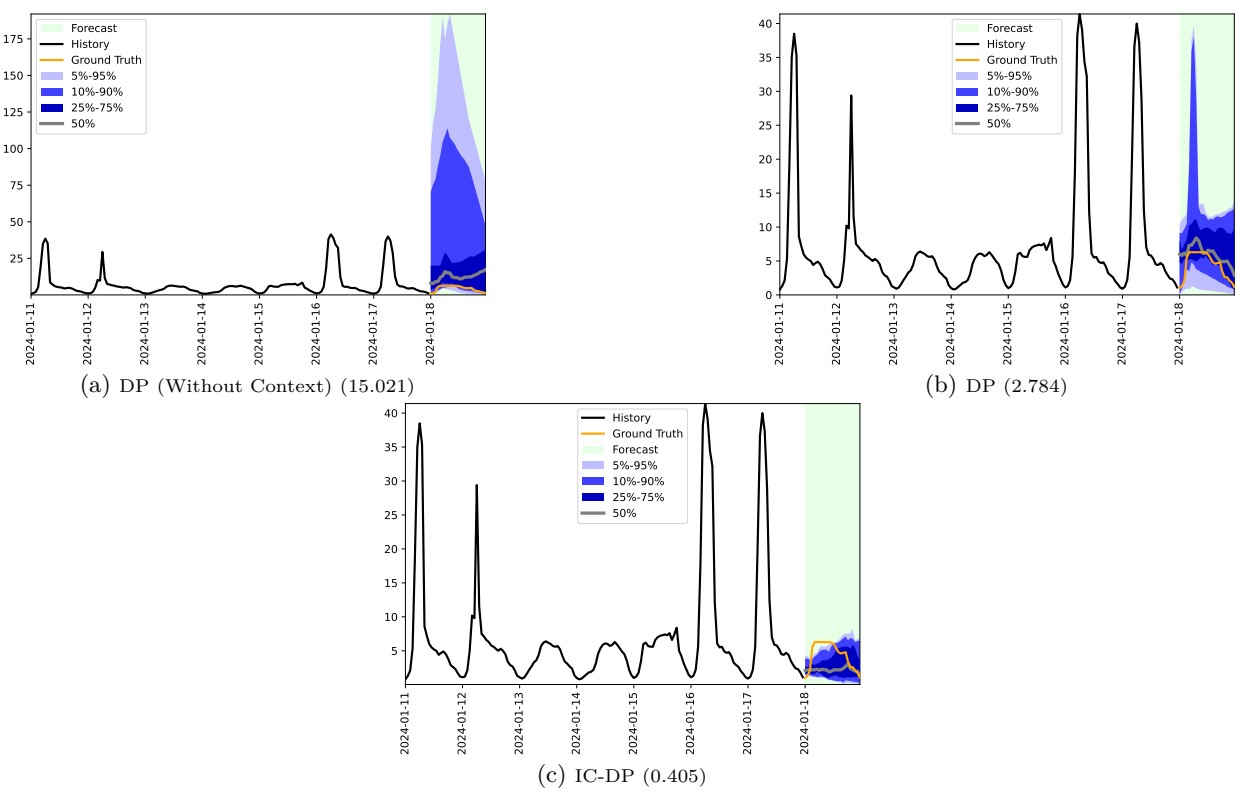

Figure 74: Forecasts of model Llama3.2-1B-Inst on task *BoundedPredConstraintsBasedOnPredQuantilesTask* (with RCRPS in brackets)

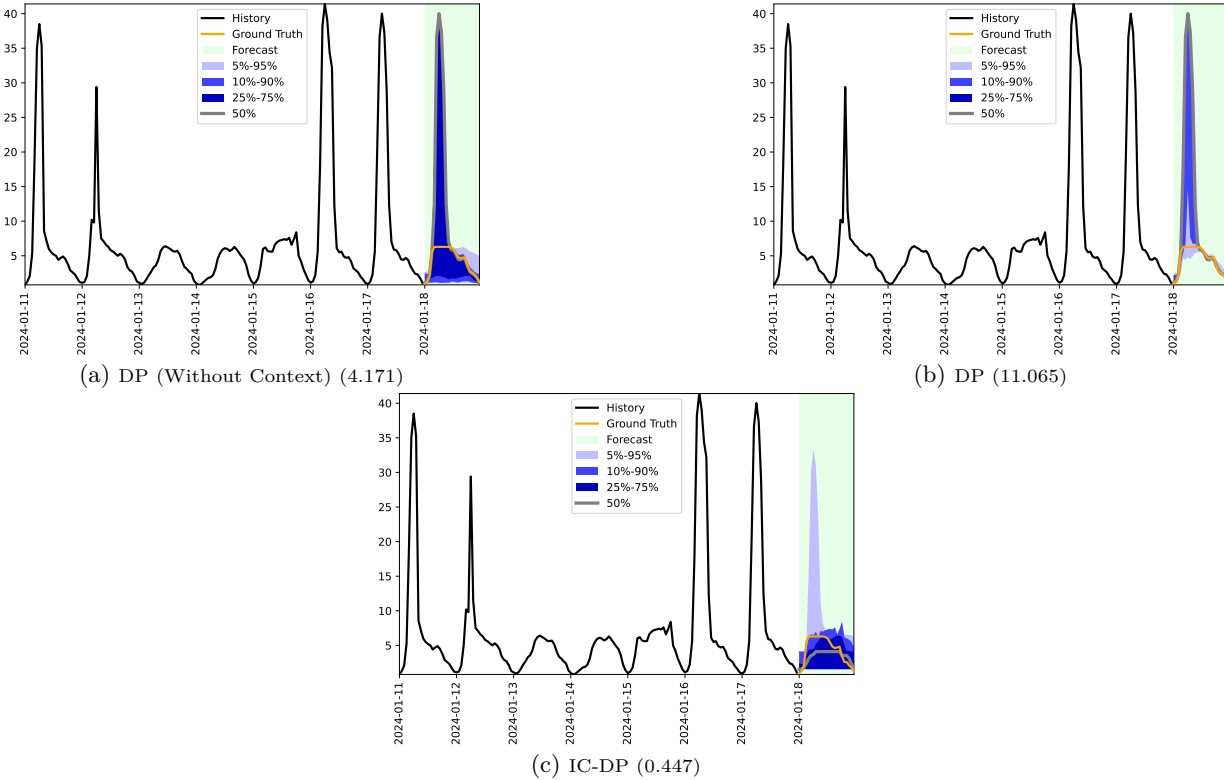

Figure 75: Forecasts of model Qwen2.5-1.5B-Inst on task *BoundedPredConstraintsBasedOnPredQuantilesTask* (with RCRPS in brackets)

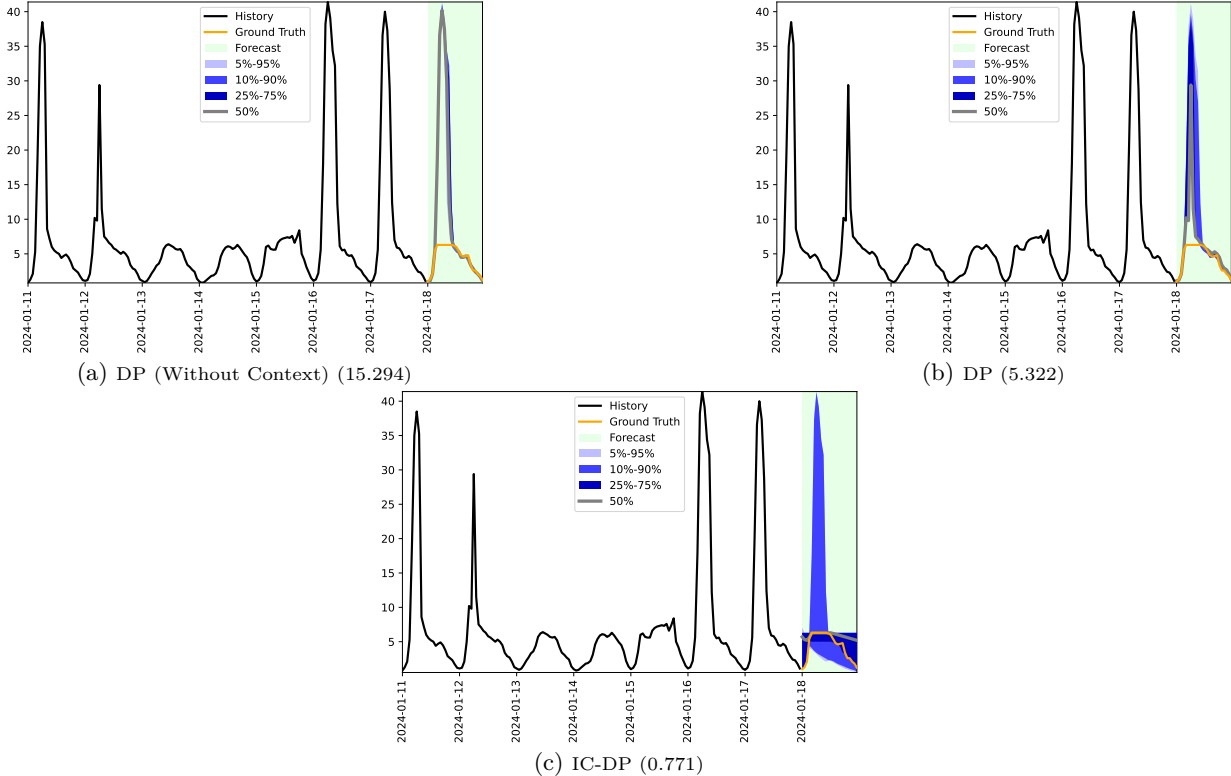

(a) DP (Without Context) (15.294)

(b) DP (5.322)

(c) IC-DP (0.771)

Figure 76: Forecasts of model Qwen2.5-7B-Inst on task *BoundedPredConstraintsBasedOnPredQuantilesTask* (with RCRPS in brackets)

# F    Additional Details on RouteDP

## F.1    RouteDP Prompt

To predict the difficulty of a task, we use the below prompt, where **{direct_prompt}** is replaced by the instantiated Direct Prompt (DP) prompt, which contains the context, historical time series and prediction timesteps of the task, as used in Williams et al. (2025).

```
{direct_prompt}
You are given a forecasting task with full contextual information.
Please rate the task as easy or hard.
Difficulty:
```

Given all 71 tasks from the CiK benchmark, we first run the designated Router LLM to predict the difficulty of a task. In particular, we use constrained decoding to limit the outputs to either "easy" or "hard".

Then, to route tasks, given a $k$ number of tasks to send to the large model, we dispatch the top-$k$ tasks considered hardest according to $P(hard)$ to the larger LLM, and dispatch the rest to the main model.

## F.2    Extended Results

| Main Model | Router | Percentage of tasks sent to large model | | | | | |
|---|---|---|---|---|---|---|---|
| | | 0% | 20% | 40% | 60% | 80% | 100% |
| Qwen2.5-0.5B-Inst | | | | | | | |
| | Qwen2.5-0.5B-Inst | 0.592 ± 0.027 | 0.316 ± 0.027 | 0.222 ± 0.005 | 0.206 ± 0.005 | 0.199 ± 0.004 | 0.173 ± 0.003 |
| | Qwen2.5-1.5B-Inst | 0.592 ± 0.027 | 0.504 ± 0.009 | 0.449 ± 0.007 | 0.404 ± 0.004 | 0.407 ± 0.004 | 0.173 ± 0.003 |
| | Qwen2.5-3B-Inst | 0.592 ± 0.027 | 0.507 ± 0.026 | 0.490 ± 0.026 | 0.393 ± 0.003 | 0.282 ± 0.003 | 0.173 ± 0.003 |
| | Qwen2.5-7B-Inst | 0.592 ± 0.027 | 0.510 ± 0.010 | 0.437 ± 0.007 | 0.412 ± 0.004 | 0.181 ± 0.004 | 0.173 ± 0.003 |
| | Qwen2.5-14B-Inst | 0.592 ± 0.027 | 0.581 ± 0.027 | 0.439 ± 0.027 | 0.324 ± 0.027 | 0.187 ± 0.004 | 0.173 ± 0.003 |
| | Qwen2.5-32B-Inst | 0.592 ± 0.027 | 0.383 ± 0.010 | 0.368 ± 0.008 | 0.230 ± 0.006 | 0.196 ± 0.004 | 0.173 ± 0.003 |
| | Qwen2.5-72B-Inst | 0.592 ± 0.027 | 0.509 ± 0.010 | 0.395 ± 0.009 | 0.287 ± 0.009 | 0.243 ± 0.009 | 0.173 ± 0.003 |
| Qwen2.5-1.5B-Inst | | | | | | | |
| | Qwen2.5-0.5B-Inst | 0.616 ± 0.018 | 0.436 ± 0.016 | 0.258 ± 0.005 | 0.231 ± 0.005 | 0.210 ± 0.004 | 0.173 ± 0.003 |
| | Qwen2.5-1.5B-Inst | 0.616 ± 0.018 | 0.466 ± 0.016 | 0.300 ± 0.016 | 0.212 ± 0.005 | 0.210 ± 0.005 | 0.173 ± 0.003 |
| | Qwen2.5-3B-Inst | 0.616 ± 0.018 | 0.375 ± 0.009 | 0.349 ± 0.009 | 0.196 ± 0.004 | 0.181 ± 0.004 | 0.173 ± 0.003 |
| | Qwen2.5-7B-Inst | 0.616 ± 0.018 | 0.481 ± 0.018 | 0.288 ± 0.016 | 0.235 ± 0.005 | 0.188 ± 0.004 | 0.173 ± 0.003 |
| | Qwen2.5-14B-Inst | 0.616 ± 0.018 | 0.598 ± 0.017 | 0.536 ± 0.016 | 0.523 ± 0.015 | 0.214 ± 0.004 | 0.173 ± 0.003 |
| | Qwen2.5-32B-Inst | 0.616 ± 0.018 | 0.441 ± 0.017 | 0.356 ± 0.017 | 0.256 ± 0.009 | 0.210 ± 0.004 | 0.173 ± 0.003 |
| | Qwen2.5-72B-Inst | 0.616 ± 0.018 | 0.484 ± 0.017 | 0.448 ± 0.017 | 0.445 ± 0.017 | 0.375 ± 0.015 | 0.173 ± 0.003 |
| Qwen2.5-3B-Inst | | | | | | | |
| | Qwen2.5-0.5B-Inst | 0.424 ± 0.017 | 0.338 ± 0.014 | 0.301 ± 0.011 | 0.281 ± 0.011 | 0.208 ± 0.004 | 0.173 ± 0.003 |
| | Qwen2.5-1.5B-Inst | 0.424 ± 0.017 | 0.383 ± 0.014 | 0.309 ± 0.012 | 0.260 ± 0.012 | 0.262 ± 0.012 | 0.173 ± 0.003 |
| | Qwen2.5-3B-Inst | 0.424 ± 0.017 | 0.315 ± 0.015 | 0.254 ± 0.011 | 0.203 ± 0.006 | 0.188 ± 0.005 | 0.173 ± 0.003 |
| | Qwen2.5-7B-Inst | 0.424 ± 0.017 | 0.382 ± 0.015 | 0.285 ± 0.012 | 0.276 ± 0.012 | 0.228 ± 0.010 | 0.173 ± 0.003 |
| | Qwen2.5-14B-Inst | 0.424 ± 0.017 | 0.402 ± 0.017 | 0.340 ± 0.015 | 0.329 ± 0.015 | 0.246 ± 0.011 | 0.173 ± 0.003 |
| | Qwen2.5-32B-Inst | 0.424 ± 0.017 | 0.359 ± 0.014 | 0.326 ± 0.013 | 0.295 ± 0.012 | 0.262 ± 0.011 | 0.173 ± 0.003 |
| | Qwen2.5-72B-Inst | 0.424 ± 0.017 | 0.392 ± 0.015 | 0.345 ± 0.015 | 0.338 ± 0.014 | 0.289 ± 0.013 | 0.173 ± 0.003 |
| Qwen2.5-7B-Inst | | | | | | | |
| | Qwen2.5-0.5B-Inst | 0.401 ± 0.006 | 0.364 ± 0.005 | 0.238 ± 0.004 | 0.229 ± 0.004 | 0.208 ± 0.004 | 0.173 ± 0.003 |
| | Qwen2.5-1.5B-Inst | 0.401 ± 0.006 | 0.263 ± 0.006 | 0.222 ± 0.005 | 0.183 ± 0.004 | 0.181 ± 0.004 | 0.173 ± 0.003 |
| | Qwen2.5-3B-Inst | 0.401 ± 0.006 | 0.338 ± 0.004 | 0.314 ± 0.004 | 0.179 ± 0.004 | 0.174 ± 0.004 | 0.173 ± 0.003 |
| | Qwen2.5-7B-Inst | 0.401 ± 0.006 | 0.260 ± 0.006 | 0.199 ± 0.005 | 0.191 ± 0.004 | 0.188 ± 0.004 | 0.173 ± 0.003 |
| | Qwen2.5-14B-Inst | 0.401 ± 0.006 | 0.384 ± 0.006 | 0.351 ± 0.006 | 0.343 ± 0.005 | 0.194 ± 0.004 | 0.173 ± 0.003 |
| | Qwen2.5-32B-Inst | 0.401 ± 0.006 | 0.260 ± 0.006 | 0.240 ± 0.005 | 0.231 ± 0.004 | 0.206 ± 0.004 | 0.173 ± 0.003 |
| | Qwen2.5-72B-Inst | 0.401 ± 0.006 | 0.267 ± 0.006 | 0.246 ± 0.006 | 0.244 ± 0.006 | 0.214 ± 0.005 | 0.173 ± 0.003 |
| Qwen2.5-14B-Inst | | | | | | | |
| | Qwen2.5-0.5B-Inst | 0.247 ± 0.006 | 0.208 ± 0.004 | 0.202 ± 0.004 | 0.199 ± 0.004 | 0.194 ± 0.004 | 0.173 ± 0.003 |
| | Qwen2.5-1.5B-Inst | 0.247 ± 0.006 | 0.246 ± 0.006 | 0.220 ± 0.006 | 0.196 ± 0.006 | 0.199 ± 0.006 | 0.173 ± 0.003 |
| | Qwen2.5-3B-Inst | 0.247 ± 0.006 | 0.206 ± 0.006 | 0.204 ± 0.006 | 0.192 ± 0.006 | 0.189 ± 0.004 | 0.173 ± 0.003 |
| | Qwen2.5-7B-Inst | 0.247 ± 0.006 | 0.234 ± 0.007 | 0.197 ± 0.006 | 0.198 ± 0.006 | 0.179 ± 0.003 | 0.173 ± 0.003 |
| | Qwen2.5-14B-Inst | 0.247 ± 0.006 | 0.237 ± 0.006 | 0.203 ± 0.006 | 0.200 ± 0.004 | 0.176 ± 0.003 | 0.173 ± 0.003 |
| | Qwen2.5-32B-Inst | 0.247 ± 0.006 | 0.230 ± 0.005 | 0.220 ± 0.005 | 0.195 ± 0.003 | 0.193 ± 0.003 | 0.173 ± 0.003 |
| | Qwen2.5-72B-Inst | 0.247 ± 0.006 | 0.238 ± 0.006 | 0.218 ± 0.005 | 0.203 ± 0.004 | 0.202 ± 0.004 | 0.173 ± 0.003 |
| Qwen2.5-32B-Inst | | | | | | | |
| | Qwen2.5-0.5B-Inst | 0.397 ± 0.008 | 0.296 ± 0.005 | 0.171 ± 0.003 | 0.172 ± 0.004 | 0.172 ± 0.003 | 0.173 ± 0.003 |
| | Qwen2.5-1.5B-Inst | 0.397 ± 0.008 | 0.278 ± 0.008 | 0.272 ± 0.008 | 0.264 ± 0.007 | 0.266 ± 0.007 | 0.173 ± 0.003 |
| | Qwen2.5-3B-Inst | 0.397 ± 0.008 | 0.390 ± 0.007 | 0.384 ± 0.007 | 0.265 ± 0.007 | 0.218 ± 0.006 | 0.173 ± 0.003 |
| | Qwen2.5-7B-Inst | 0.397 ± 0.008 | 0.276 ± 0.008 | 0.273 ± 0.008 | 0.265 ± 0.007 | 0.175 ± 0.003 | 0.173 ± 0.003 |
| | Qwen2.5-14B-Inst | 0.397 ± 0.008 | 0.397 ± 0.008 | 0.361 ± 0.007 | 0.310 ± 0.005 | 0.177 ± 0.003 | 0.173 ± 0.003 |
| | Qwen2.5-32B-Inst | 0.397 ± 0.008 | 0.240 ± 0.007 | 0.237 ± 0.007 | 0.185 ± 0.003 | 0.181 ± 0.004 | 0.173 ± 0.003 |
| | Qwen2.5-72B-Inst | 0.397 ± 0.008 | 0.284 ± 0.008 | 0.236 ± 0.007 | 0.191 ± 0.005 | 0.193 ± 0.006 | 0.173 ± 0.003 |
| Qwen2.5-72B-Inst | | | | | | | |
| | Qwen2.5-0.5B-Inst | 0.202 ± 0.009 | 0.167 ± 0.007 | 0.156 ± 0.004 | 0.158 ± 0.004 | 0.165 ± 0.004 | 0.173 ± 0.003 |
| | Qwen2.5-1.5B-Inst | 0.202 ± 0.009 | 0.184 ± 0.006 | 0.181 ± 0.006 | 0.185 ± 0.006 | 0.194 ± 0.006 | 0.173 ± 0.003 |
| | Qwen2.5-3B-Inst | 0.202 ± 0.009 | 0.207 ± 0.009 | 0.210 ± 0.009 | 0.189 ± 0.006 | 0.178 ± 0.004 | 0.173 ± 0.003 |
| | Qwen2.5-7B-Inst | 0.202 ± 0.009 | 0.187 ± 0.006 | 0.185 ± 0.006 | 0.192 ± 0.006 | 0.175 ± 0.003 | 0.173 ± 0.003 |
| | Qwen2.5-14B-Inst | 0.202 ± 0.009 | 0.207 ± 0.009 | 0.202 ± 0.009 | 0.190 ± 0.008 | 0.175 ± 0.003 | 0.173 ± 0.003 |
| | Qwen2.5-32B-Inst | 0.202 ± 0.009 | 0.183 ± 0.005 | 0.186 ± 0.005 | 0.180 ± 0.004 | 0.175 ± 0.004 | 0.173 ± 0.003 |
| | Qwen2.5-72B-Inst | 0.202 ± 0.009 | 0.198 ± 0.006 | 0.188 ± 0.005 | 0.185 ± 0.004 | 0.180 ± 0.004 | 0.173 ± 0.003 |

Table 14: Average RCRPS of main models routed with different routers, as the percentage of tasks sent to the large model increases. The means are accompanied by standard errors.

| Router | Qwen2.5-0.5B-Inst | Qwen2.5 1.5B | Qwen2.5 3B | Qwen2.5 7B | Qwen2.5 14B | Qwen2.5 32B | Qwen2.5-72B-Inst |
|---|---|---|---|---|---|---|---|
| Qwen2.5-0.5B-Inst | 66.76 | 48.83 | 13.50 | 29.05 | 22.03 | 68.59 | 67.59 |
| Qwen2.5-1.5B-Inst | 1.40 | 40.53 | 4.67 | 55.13 | 2.63 | 19.65 | 12.61 |
| Qwen2.5-3B-Inst | 3.10 | 46.41 | 45.41 | 23.05 | 22.23 | 3.47 | 1.23 |
| Qwen2.5-7B-Inst | 7.95 | 35.33 | 7.41 | 55.72 | 15.73 | 26.63 | 6.71 |
| Qwen2.5-14B-Inst | 10.12 | 4.62 | 0.00 | 5.11 | 12.14 | 7.66 | 3.77 |
| Qwen2.5-32B-Inst | 36.96 | 39.04 | 6.52 | 51.15 | 7.79 | 58.45 | 14.86 |
| Qwen2.5-72B-Inst | 9.73 | 3.35 | 0.00 | 29.19 | 0.49 | 34.67 | 3.34 |

Table 15: Area captured by each router for each main model, between the main model's own random and ideal routing curves. Values of different routers across the same main model are comparable.

### F.3 Plots showcasing the area captured by different router models

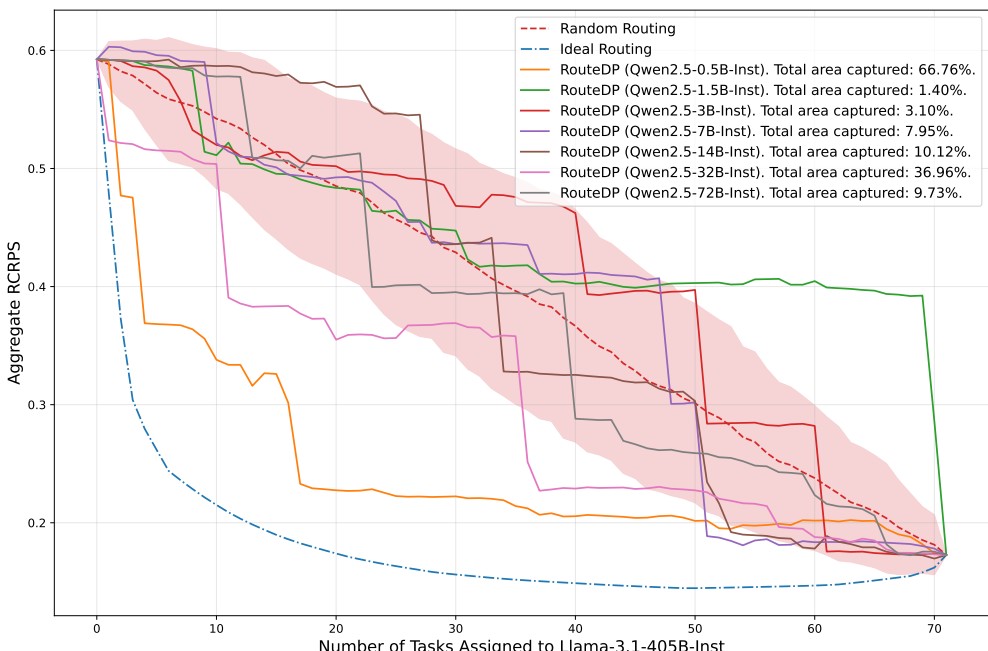

Figure 77: Random, ideal and router curves with Qwen2.5-0.5B-Inst as the main model. The shaded region represents the distribution of 100 random assignment trajectories.

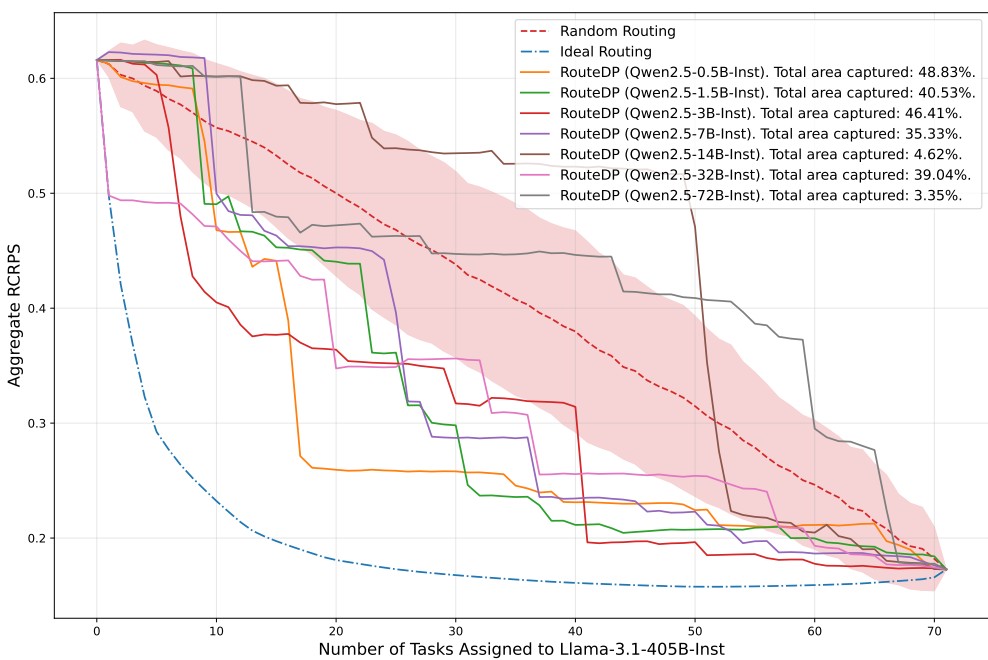

Figure 78: Random, ideal and router curves with Qwen2.5-1.5B-Inst as the main model. The shaded region represents the distribution of 100 random assignment trajectories.

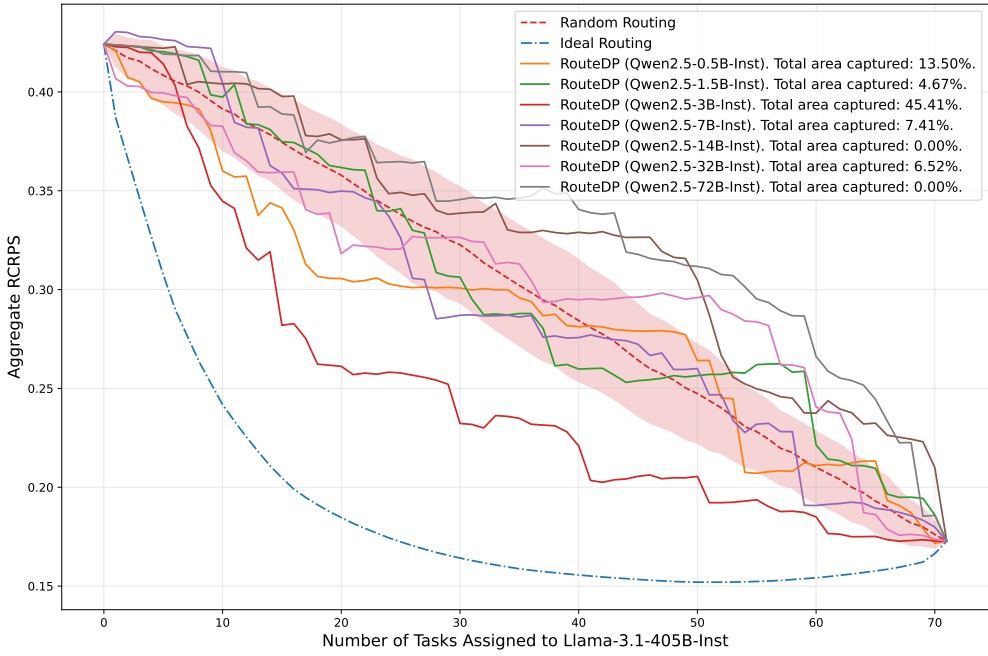

Figure 79: Random, ideal and router curves with Qwen2.5-3B-Inst as the main model. The shaded region represents the distribution of 100 random assignment trajectories.

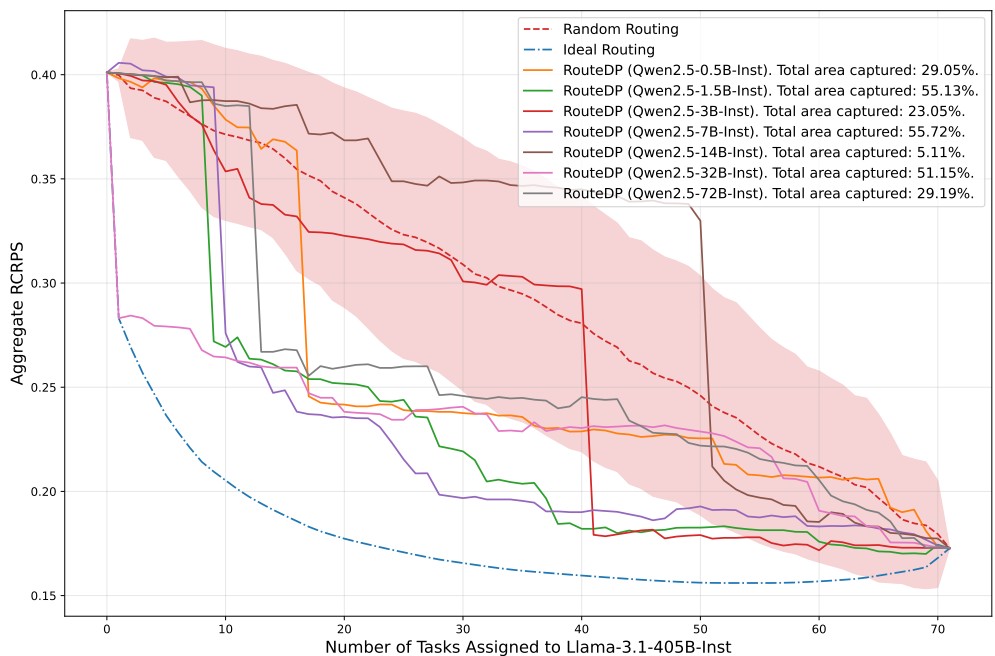

Figure 80: Random, ideal and router curves with Qwen2.5-7B-Inst as the main model. The shaded region represents the distribution of 100 random assignment trajectories.

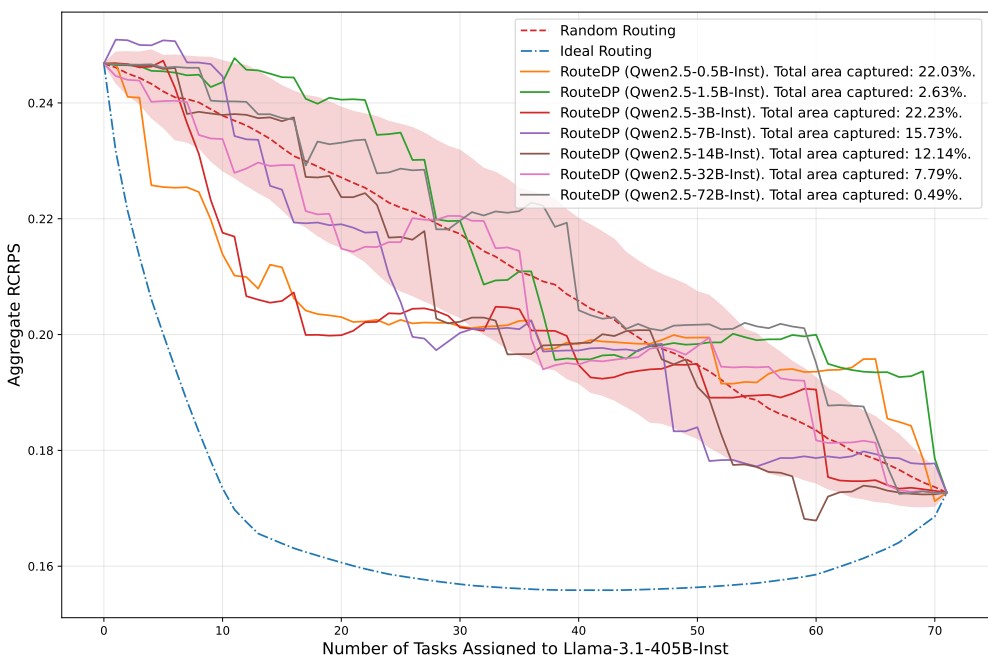

Figure 81: Random, ideal and router curves with Qwen2.5-14B-Inst as the main model. The shaded region represents the distribution of 100 random assignment trajectories.

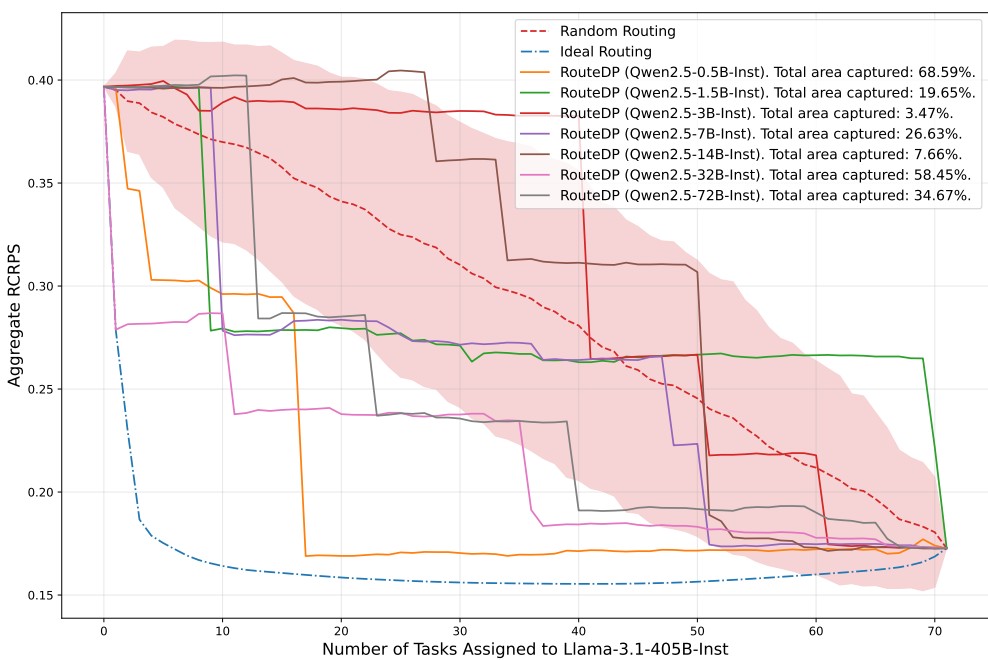

Figure 82: Random, ideal and router curves with Qwen2.5-32B-Inst as the main model. The shaded region represents the distribution of 100 random assignment trajectories.

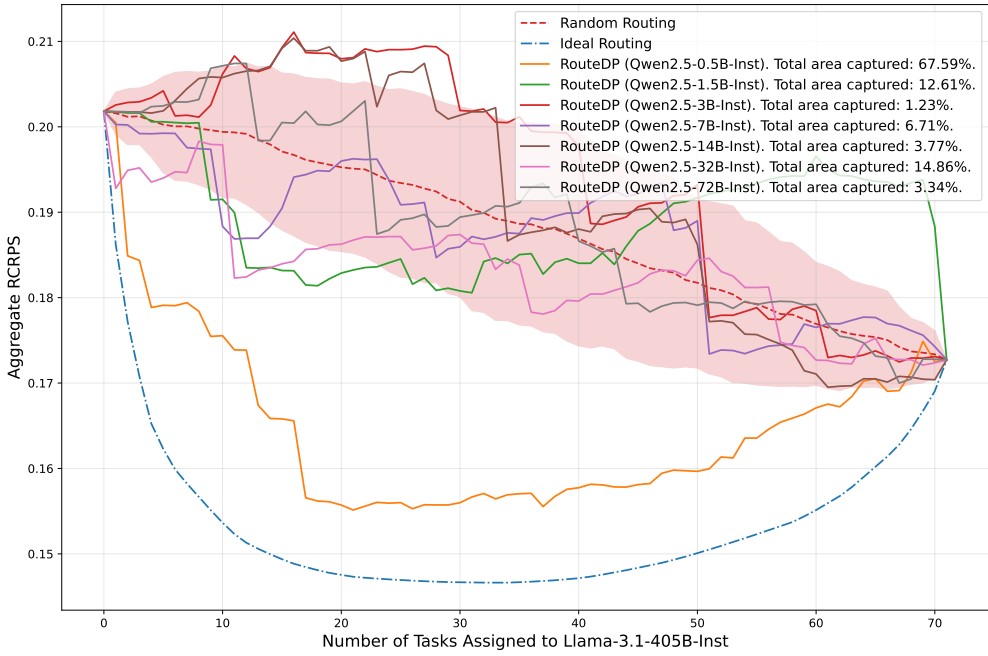

Figure 83: Random, ideal and router curves with Qwen2.5-72B-Inst as the main model. The shaded region represents the distribution of 100 random assignment trajectories.

## F.4 RouteDP with other methods

The advantage of the proposed RouteDP method is that the difficulty scores predicted by the router can, in principle, be used route tasks to models irrespective of what method the downstream model uses to

obtain forecasts. To evaluate if RouteDP empirically improves the performance of downstream models for which methods other than DP were used to obtain forecasts, we test it with models that used IC-DP and CorDP to obtain forecasts. We call these methods Route-IC-DP and Route-Cor-DP respectively, indicating that the routing is done on IC-DP and CorDP forecasts respectively. For CorDP, we evaluate it with the SampleWiseCorDP method using Lag-Llama as the base forecaster.

Results with IC-DP are in Table 16, and results with CorDP are in Table 17. We observe improvements similar to with DP, across several router models and main models. This shows that the difficulties predicted by the router model may not depend on the downstream strategy (DP, IC-DP, CorDP etc.) employed. Example routing curves with Qwen2.5-0.5B-Inst as the main model and the router are provided in Figure 84 and Figure 85 respectively.

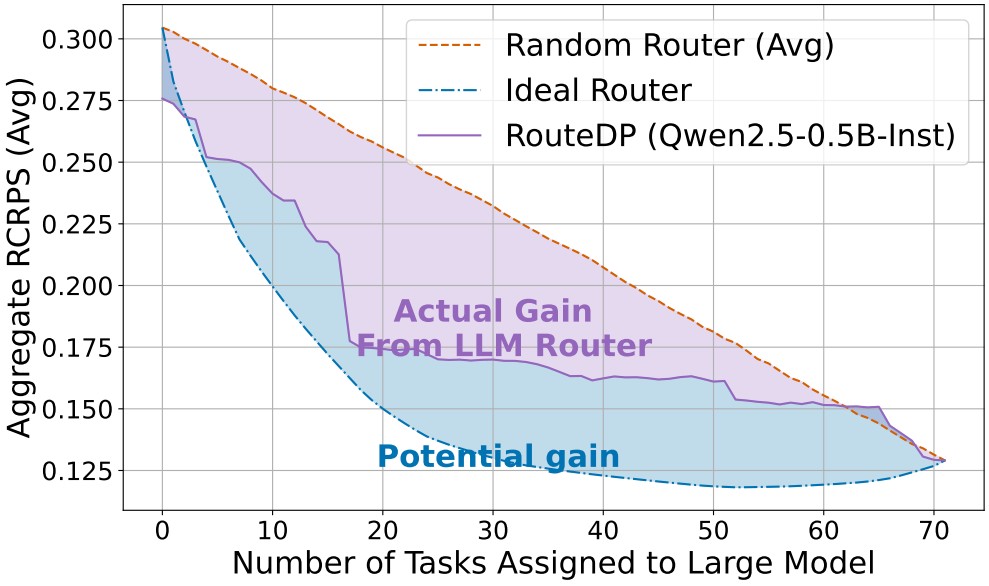

Figure 84: Route-IC-DP: Random, ideal and router curves with Qwen2.5-0.5B-Inst as the main model. The shaded region represents the distribution of 100 random assignment trajectories.

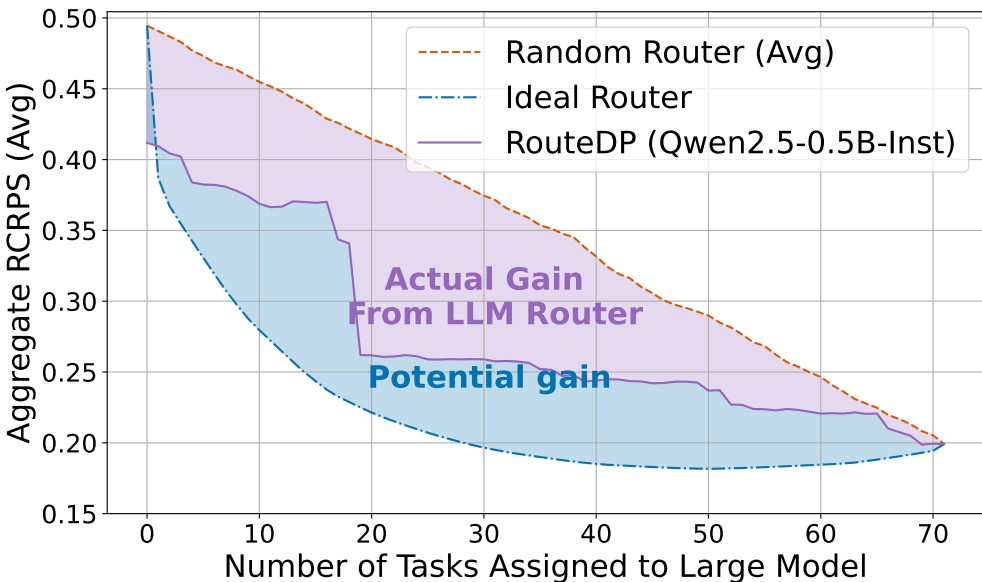

Figure 85: Route-CorDP: Random, ideal and router curves with Qwen2.5-0.5B-Inst as the main model. The shaded region represents the distribution of 100 random assignment trajectories.

| Main Model | Router | Percentage of tasks sent to large model | | | | | |
|---|---|---|---|---|---|---|---|
| | | 0% | 20% | 40% | 60% | 80% | 100% |
| Qwen2.5-0.5B-Inst | | | | | | | |
| | Qwen2.5-0.5B-Inst | 0.305 ± 0.006 | 0.225 ± 0.005 | 0.181 ± 0.004 | 0.160 ± 0.004 | 0.154 ± 0.004 | 0.129 ± 0.004 |
| | Qwen2.5-1.5B-Inst | 0.305 ± 0.006 | 0.250 ± 0.006 | 0.198 ± 0.005 | 0.171 ± 0.004 | 0.167 ± 0.004 | 0.129 ± 0.004 |
| | Qwen2.5-3B-Inst | 0.305 ± 0.006 | 0.241 ± 0.004 | 0.224 ± 0.004 | 0.157 ± 0.004 | 0.143 ± 0.004 | 0.129 ± 0.004 |
| | Qwen2.5-7B-Inst | 0.305 ± 0.006 | 0.265 ± 0.006 | 0.198 ± 0.005 | 0.185 ± 0.004 | 0.143 ± 0.004 | 0.129 ± 0.004 |
| | Qwen2.5-14B-Inst | 0.305 ± 0.006 | 0.297 ± 0.006 | 0.239 ± 0.006 | 0.222 ± 0.006 | 0.160 ± 0.004 | 0.129 ± 0.004 |
| | Qwen2.5-32B-Inst | 0.305 ± 0.006 | 0.244 ± 0.006 | 0.222 ± 0.005 | 0.191 ± 0.004 | 0.146 ± 0.004 | 0.129 ± 0.004 |
| | Qwen2.5-72B-Inst | 0.305 ± 0.006 | 0.274 ± 0.006 | 0.254 ± 0.006 | 0.236 ± 0.006 | 0.172 ± 0.006 | 0.129 ± 0.004 |
| Qwen2.5-1.5B-Inst | | | | | | | |
| | Qwen2.5-0.5B-Inst | 0.273 ± 0.008 | 0.216 ± 0.007 | 0.186 ± 0.005 | 0.171 ± 0.005 | 0.160 ± 0.004 | 0.129 ± 0.004 |
| | Qwen2.5-1.5B-Inst | 0.273 ± 0.008 | 0.243 ± 0.007 | 0.193 ± 0.005 | 0.165 ± 0.004 | 0.166 ± 0.004 | 0.129 ± 0.004 |
| | Qwen2.5-3B-Inst | 0.273 ± 0.008 | 0.197 ± 0.004 | 0.184 ± 0.005 | 0.153 ± 0.004 | 0.142 ± 0.004 | 0.129 ± 0.004 |
| | Qwen2.5-7B-Inst | 0.273 ± 0.008 | 0.237 ± 0.007 | 0.189 ± 0.005 | 0.178 ± 0.004 | 0.138 ± 0.004 | 0.129 ± 0.004 |
| | Qwen2.5-14B-Inst | 0.273 ± 0.008 | 0.260 ± 0.007 | 0.212 ± 0.007 | 0.197 ± 0.006 | 0.155 ± 0.003 | 0.129 ± 0.004 |
| | Qwen2.5-32B-Inst | 0.273 ± 0.008 | 0.216 ± 0.006 | 0.198 ± 0.005 | 0.171 ± 0.003 | 0.148 ± 0.003 | 0.129 ± 0.004 |
| | Qwen2.5-72B-Inst | 0.273 ± 0.008 | 0.244 ± 0.006 | 0.207 ± 0.006 | 0.196 ± 0.006 | 0.161 ± 0.005 | 0.129 ± 0.004 |
| Qwen2.5-3B-Inst | | | | | | | |
| | Qwen2.5-0.5B-Inst | 0.298 ± 0.011 | 0.247 ± 0.010 | 0.222 ± 0.009 | 0.220 ± 0.009 | 0.197 ± 0.009 | 0.129 ± 0.004 |
| | Qwen2.5-1.5B-Inst | 0.298 ± 0.011 | 0.255 ± 0.009 | 0.211 ± 0.009 | 0.151 ± 0.004 | 0.148 ± 0.004 | 0.129 ± 0.004 |
| | Qwen2.5-3B-Inst | 0.298 ± 0.011 | 0.223 ± 0.011 | 0.194 ± 0.007 | 0.148 ± 0.004 | 0.138 ± 0.004 | 0.129 ± 0.004 |
| | Qwen2.5-7B-Inst | 0.298 ± 0.011 | 0.250 ± 0.010 | 0.153 ± 0.004 | 0.148 ± 0.004 | 0.135 ± 0.004 | 0.129 ± 0.004 |
| | Qwen2.5-14B-Inst | 0.298 ± 0.011 | 0.274 ± 0.011 | 0.205 ± 0.010 | 0.194 ± 0.010 | 0.161 ± 0.009 | 0.129 ± 0.004 |
| | Qwen2.5-32B-Inst | 0.298 ± 0.011 | 0.250 ± 0.009 | 0.223 ± 0.009 | 0.213 ± 0.009 | 0.183 ± 0.009 | 0.129 ± 0.004 |
| | Qwen2.5-72B-Inst | 0.298 ± 0.011 | 0.265 ± 0.010 | 0.207 ± 0.010 | 0.201 ± 0.009 | 0.169 ± 0.009 | 0.129 ± 0.004 |
| Qwen2.5-7B-Inst | | | | | | | |
| | Qwen2.5-0.5B-Inst | 0.264 ± 0.012 | 0.209 ± 0.011 | 0.169 ± 0.003 | 0.164 ± 0.003 | 0.148 ± 0.003 | 0.129 ± 0.004 |
| | Qwen2.5-1.5B-Inst | 0.264 ± 0.012 | 0.218 ± 0.007 | 0.171 ± 0.005 | 0.147 ± 0.004 | 0.140 ± 0.004 | 0.129 ± 0.004 |
| | Qwen2.5-3B-Inst | 0.264 ± 0.012 | 0.197 ± 0.011 | 0.188 ± 0.011 | 0.143 ± 0.004 | 0.133 ± 0.004 | 0.129 ± 0.004 |
| | Qwen2.5-7B-Inst | 0.264 ± 0.012 | 0.215 ± 0.007 | 0.164 ± 0.005 | 0.151 ± 0.004 | 0.137 ± 0.004 | 0.129 ± 0.004 |
| | Qwen2.5-14B-Inst | 0.264 ± 0.012 | 0.246 ± 0.012 | 0.208 ± 0.012 | 0.196 ± 0.012 | 0.140 ± 0.003 | 0.129 ± 0.004 |
| | Qwen2.5-32B-Inst | 0.264 ± 0.012 | 0.231 ± 0.007 | 0.201 ± 0.006 | 0.168 ± 0.003 | 0.146 ± 0.002 | 0.129 ± 0.004 |
| | Qwen2.5-72B-Inst | 0.264 ± 0.012 | 0.242 ± 0.007 | 0.206 ± 0.007 | 0.196 ± 0.007 | 0.166 ± 0.006 | 0.129 ± 0.004 |
| Qwen2.5-14B-Inst | | | | | | | |
| | Qwen2.5-0.5B-Inst | 0.270 ± 0.005 | 0.258 ± 0.003 | 0.131 ± 0.003 | 0.132 ± 0.003 | 0.132 ± 0.003 | 0.129 ± 0.004 |
| | Qwen2.5-1.5B-Inst | 0.270 ± 0.005 | 0.142 ± 0.005 | 0.134 ± 0.005 | 0.134 ± 0.005 | 0.135 ± 0.005 | 0.129 ± 0.004 |
| | Qwen2.5-3B-Inst | 0.270 ± 0.005 | 0.258 ± 0.005 | 0.260 ± 0.005 | 0.132 ± 0.005 | 0.136 ± 0.005 | 0.129 ± 0.004 |
| | Qwen2.5-7B-Inst | 0.270 ± 0.005 | 0.138 ± 0.005 | 0.135 ± 0.005 | 0.133 ± 0.005 | 0.130 ± 0.004 | 0.129 ± 0.004 |
| | Qwen2.5-14B-Inst | 0.270 ± 0.005 | 0.267 ± 0.005 | 0.255 ± 0.003 | 0.254 ± 0.003 | 0.132 ± 0.003 | 0.129 ± 0.004 |
| | Qwen2.5-32B-Inst | 0.270 ± 0.005 | 0.153 ± 0.005 | 0.141 ± 0.005 | 0.129 ± 0.003 | 0.128 ± 0.003 | 0.129 ± 0.004 |
| | Qwen2.5-72B-Inst | 0.270 ± 0.005 | 0.148 ± 0.005 | 0.141 ± 0.005 | 0.133 ± 0.003 | 0.131 ± 0.003 | 0.129 ± 0.004 |
| Qwen2.5-32B-Inst | | | | | | | |
| | Qwen2.5-0.5B-Inst | 0.245 ± 0.027 | 0.189 ± 0.027 | 0.128 ± 0.002 | 0.129 ± 0.003 | 0.124 ± 0.002 | 0.129 ± 0.004 |
| | Qwen2.5-1.5B-Inst | 0.245 ± 0.027 | 0.179 ± 0.003 | 0.177 ± 0.003 | 0.183 ± 0.004 | 0.182 ± 0.004 | 0.129 ± 0.004 |
| | Qwen2.5-3B-Inst | 0.245 ± 0.027 | 0.241 ± 0.027 | 0.243 ± 0.027 | 0.177 ± 0.004 | 0.178 ± 0.004 | 0.129 ± 0.004 |
| | Qwen2.5-7B-Inst | 0.245 ± 0.027 | 0.181 ± 0.003 | 0.183 ± 0.004 | 0.182 ± 0.004 | 0.131 ± 0.004 | 0.129 ± 0.004 |
| | Qwen2.5-14B-Inst | 0.245 ± 0.027 | 0.241 ± 0.027 | 0.187 ± 0.027 | 0.184 ± 0.027 | 0.132 ± 0.002 | 0.129 ± 0.004 |
| | Qwen2.5-32B-Inst | 0.245 ± 0.027 | 0.193 ± 0.003 | 0.180 ± 0.003 | 0.130 ± 0.002 | 0.123 ± 0.002 | 0.129 ± 0.004 |
| | Qwen2.5-72B-Inst | 0.245 ± 0.027 | 0.193 ± 0.003 | 0.181 ± 0.003 | 0.126 ± 0.002 | 0.123 ± 0.002 | 0.129 ± 0.004 |

Table 16: Routing performance when k tasks are sent to large model, with Route-IC-DP

| Main Model | Router | Percentage of tasks sent to large model | | | | | |
| --- | --- | --- | --- | --- | --- | --- | --- |
| | | 0% | 20% | 40% | 60% | 80% | 100% |
| Qwen2.5-0.5B-Inst | | | | | | | |
| | Qwen2.5-0.5B-Inst | 0.494 ± 0.008 | 0.332 ± 0.009 | 0.295 ± 0.007 | 0.269 ± 0.007 | 0.235 ± 0.006 | 0.199 ± 0.006 |
| | Qwen2.5-1.5B-Inst | 0.494 ± 0.008 | 0.467 ± 0.006 | 0.399 ± 0.006 | 0.364 ± 0.006 | 0.356 ± 0.006 | 0.199 ± 0.006 |
| | Qwen2.5-3B-Inst | 0.494 ± 0.008 | 0.408 ± 0.008 | 0.366 ± 0.008 | 0.325 ± 0.005 | 0.313 ± 0.005 | 0.199 ± 0.006 |
| | Qwen2.5-7B-Inst | 0.494 ± 0.008 | 0.445 ± 0.006 | 0.398 ± 0.006 | 0.387 ± 0.007 | 0.230 ± 0.006 | 0.199 ± 0.006 |
| | Qwen2.5-14B-Inst | 0.494 ± 0.008 | 0.477 ± 0.008 | 0.348 ± 0.009 | 0.323 ± 0.008 | 0.243 ± 0.007 | 0.199 ± 0.006 |
| | Qwen2.5-32B-Inst | 0.494 ± 0.008 | 0.420 ± 0.005 | 0.390 ± 0.005 | 0.270 ± 0.006 | 0.242 ± 0.006 | 0.199 ± 0.006 |
| | Qwen2.5-72B-Inst | 0.494 ± 0.008 | 0.454 ± 0.006 | 0.419 ± 0.006 | 0.306 ± 0.006 | 0.270 ± 0.006 | 0.199 ± 0.006 |
| Qwen2.5-1.5B-Inst | | | | | | | |
| | Qwen2.5-0.5B-Inst | 0.522 ± 0.018 | 0.475 ± 0.018 | 0.295 ± 0.007 | 0.267 ± 0.007 | 0.234 ± 0.006 | 0.199 ± 0.006 |
| | Qwen2.5-1.5B-Inst | 0.522 ± 0.018 | 0.355 ± 0.007 | 0.290 ± 0.006 | 0.257 ± 0.007 | 0.250 ± 0.007 | 0.199 ± 0.006 |
| | Qwen2.5-3B-Inst | 0.522 ± 0.018 | 0.439 ± 0.018 | 0.395 ± 0.017 | 0.215 ± 0.006 | 0.207 ± 0.005 | 0.199 ± 0.006 |
| | Qwen2.5-7B-Inst | 0.522 ± 0.018 | 0.449 ± 0.007 | 0.409 ± 0.007 | 0.281 ± 0.007 | 0.231 ± 0.006 | 0.199 ± 0.006 |
| | Qwen2.5-14B-Inst | 0.522 ± 0.018 | 0.507 ± 0.018 | 0.484 ± 0.018 | 0.465 ± 0.018 | 0.364 ± 0.007 | 0.199 ± 0.006 |
| | Qwen2.5-32B-Inst | 0.522 ± 0.018 | 0.308 ± 0.006 | 0.282 ± 0.006 | 0.268 ± 0.006 | 0.244 ± 0.007 | 0.199 ± 0.006 |
| | Qwen2.5-72B-Inst | 0.522 ± 0.018 | 0.459 ± 0.007 | 0.430 ± 0.007 | 0.304 ± 0.007 | 0.267 ± 0.006 | 0.199 ± 0.006 |
| Qwen2.5-3B-Inst | | | | | | | |
| | Qwen2.5-0.5B-Inst | 0.398 ± 0.028 | 0.374 ± 0.028 | 0.267 ± 0.006 | 0.251 ± 0.006 | 0.231 ± 0.006 | 0.199 ± 0.006 |
| | Qwen2.5-1.5B-Inst | 0.398 ± 0.028 | 0.300 ± 0.006 | 0.244 ± 0.006 | 0.220 ± 0.006 | 0.220 ± 0.006 | 0.199 ± 0.006 |
| | Qwen2.5-3B-Inst | 0.398 ± 0.028 | 0.322 ± 0.028 | 0.304 ± 0.028 | 0.201 ± 0.006 | 0.201 ± 0.006 | 0.199 ± 0.006 |
| | Qwen2.5-7B-Inst | 0.398 ± 0.028 | 0.274 ± 0.006 | 0.240 ± 0.006 | 0.237 ± 0.007 | 0.205 ± 0.006 | 0.199 ± 0.006 |
| | Qwen2.5-14B-Inst | 0.398 ± 0.028 | 0.380 ± 0.028 | 0.360 ± 0.029 | 0.349 ± 0.028 | 0.226 ± 0.006 | 0.199 ± 0.006 |
| | Qwen2.5-32B-Inst | 0.398 ± 0.028 | 0.260 ± 0.005 | 0.232 ± 0.004 | 0.227 ± 0.005 | 0.224 ± 0.006 | 0.199 ± 0.006 |
| | Qwen2.5-72B-Inst | 0.398 ± 0.028 | 0.286 ± 0.006 | 0.254 ± 0.006 | 0.252 ± 0.006 | 0.225 ± 0.005 | 0.199 ± 0.006 |
| Qwen2.5-7B-Inst | | | | | | | |
| | Qwen2.5-0.5B-Inst | 0.382 ± 0.007 | 0.374 ± 0.007 | 0.251 ± 0.008 | 0.235 ± 0.007 | 0.214 ± 0.006 | 0.199 ± 0.006 |
| | Qwen2.5-1.5B-Inst | 0.382 ± 0.007 | 0.263 ± 0.007 | 0.238 ± 0.006 | 0.234 ± 0.007 | 0.230 ± 0.007 | 0.199 ± 0.006 |
| | Qwen2.5-3B-Inst | 0.382 ± 0.007 | 0.351 ± 0.006 | 0.334 ± 0.006 | 0.210 ± 0.006 | 0.208 ± 0.006 | 0.199 ± 0.006 |
| | Qwen2.5-7B-Inst | 0.382 ± 0.007 | 0.362 ± 0.006 | 0.352 ± 0.006 | 0.236 ± 0.007 | 0.219 ± 0.007 | 0.199 ± 0.006 |
| | Qwen2.5-14B-Inst | 0.382 ± 0.007 | 0.374 ± 0.006 | 0.369 ± 0.006 | 0.361 ± 0.006 | 0.345 ± 0.007 | 0.199 ± 0.006 |
| | Qwen2.5-32B-Inst | 0.382 ± 0.007 | 0.245 ± 0.006 | 0.228 ± 0.006 | 0.229 ± 0.007 | 0.225 ± 0.007 | 0.199 ± 0.006 |
| | Qwen2.5-72B-Inst | 0.382 ± 0.007 | 0.380 ± 0.007 | 0.360 ± 0.006 | 0.236 ± 0.006 | 0.224 ± 0.006 | 0.199 ± 0.006 |
| Qwen2.5-14B-Inst | | | | | | | |
| | Qwen2.5-0.5B-Inst | 0.364 ± 0.006 | 0.358 ± 0.006 | 0.246 ± 0.007 | 0.243 ± 0.007 | 0.231 ± 0.007 | 0.199 ± 0.006 |
| | Qwen2.5-1.5B-Inst | 0.364 ± 0.006 | 0.239 ± 0.006 | 0.223 ± 0.006 | 0.206 ± 0.006 | 0.204 ± 0.006 | 0.199 ± 0.006 |
| | Qwen2.5-3B-Inst | 0.364 ± 0.006 | 0.334 ± 0.007 | 0.325 ± 0.006 | 0.196 ± 0.006 | 0.198 ± 0.006 | 0.199 ± 0.006 |
| | Qwen2.5-7B-Inst | 0.364 ± 0.006 | 0.348 ± 0.006 | 0.314 ± 0.005 | 0.203 ± 0.006 | 0.198 ± 0.006 | 0.199 ± 0.006 |
| | Qwen2.5-14B-Inst | 0.364 ± 0.006 | 0.351 ± 0.006 | 0.323 ± 0.006 | 0.318 ± 0.005 | 0.324 ± 0.007 | 0.199 ± 0.006 |
| | Qwen2.5-32B-Inst | 0.364 ± 0.006 | 0.227 ± 0.006 | 0.212 ± 0.006 | 0.222 ± 0.006 | 0.219 ± 0.007 | 0.199 ± 0.006 |
| | Qwen2.5-72B-Inst | 0.364 ± 0.006 | 0.351 ± 0.006 | 0.326 ± 0.006 | 0.204 ± 0.005 | 0.196 ± 0.005 | 0.199 ± 0.006 |
| Qwen2.5-32B-Inst | | | | | | | |
| | Qwen2.5-0.5B-Inst | 0.310 ± 0.005 | 0.311 ± 0.006 | 0.199 ± 0.006 | 0.204 ± 0.006 | 0.199 ± 0.006 | 0.199 ± 0.006 |
| | Qwen2.5-1.5B-Inst | 0.310 ± 0.005 | 0.190 ± 0.005 | 0.192 ± 0.005 | 0.200 ± 0.006 | 0.201 ± 0.006 | 0.199 ± 0.006 |
| | Qwen2.5-3B-Inst | 0.310 ± 0.005 | 0.309 ± 0.005 | 0.314 ± 0.005 | 0.195 ± 0.006 | 0.198 ± 0.006 | 0.199 ± 0.006 |
| | Qwen2.5-7B-Inst | 0.310 ± 0.005 | 0.298 ± 0.004 | 0.306 ± 0.005 | 0.194 ± 0.005 | 0.195 ± 0.006 | 0.199 ± 0.006 |
| | Qwen2.5-14B-Inst | 0.310 ± 0.005 | 0.305 ± 0.004 | 0.299 ± 0.004 | 0.294 ± 0.004 | 0.304 ± 0.005 | 0.199 ± 0.006 |
| | Qwen2.5-32B-Inst | 0.310 ± 0.005 | 0.190 ± 0.005 | 0.184 ± 0.005 | 0.188 ± 0.006 | 0.195 ± 0.006 | 0.199 ± 0.006 |
| | Qwen2.5-72B-Inst | 0.310 ± 0.005 | 0.307 ± 0.005 | 0.296 ± 0.004 | 0.173 ± 0.004 | 0.180 ± 0.004 | 0.199 ± 0.006 |

Table 17: Routing performance when k tasks are sent to large model, with Route-CorDP (SampleWise-CorDP, Lag-Llama as base forecaster)

# G   Implementation Details

To evaluate our models on the CiK benchmark, we use the official codebase of CiK at `https://github.com/ServiceNow/context-is-key-forecasting`. We use the same codebase to run model on the Direct Prompt (DP) method and the quantitative baselines benchmarked for CorDP. For completeness, we provide the details here. Code for all proposed methods will be released on acceptance, with Instions to reproduce all experiments.

### G.1 LLMs

We self-host the other models with the respective official HuggingFace models: Llama3.2-1B-Inst (`https://huggingface.co/meta-Llama/Llama3.2-1B-Inst`), Llama3.2-3B-Inst (`https://huggingface.co/meta-Llama/Llama3.2-3B-Inst`), Llama3-8B-Inst (`https://huggingface.co/meta-Llama/Meta-Llama3-8B-Inst`), Qwen2.5-0.5B-Inst (`https://huggingface.co/Qwen2.5-0.5B-Inst`), Qwen2.5-1.5B-Inst (`https://huggingface.co/Qwen2.5-1.5B-Inst`), Qwen2.5-3B-Inst (`https://huggingface.co/Qwen2.5-3B-Inst`), Qwen2.5-7B-Inst (`https://huggingface.co/Qwen2.5-7B-Inst`), Qwen2.5-14B-Inst (`https://huggingface.co/Qwen2.5-14B-Inst`), Qwen2.5-32B-Inst (`https://huggingface.co/Qwen2.5-32B-Inst`). We use an appropriate number of H100 GPUs for each model. This ranged from 1 GPU (Models below 7B), 2 GPUs (7B, 14B Models) and 4 GPUs (32B Models).

Due to compute restrictions, for all our experiments involving Llama3.1-405B-Inst, Llama3.3-70B-Inst and Qwen2.5-72B-Inst, we use OpenRouter endpoints at `https://openrouter.ai/meta-Llama/Llama3.1-405b-Inst`, `https://openrouter.ai/meta-Llama/Llama3.3-70b-Inst` and `https://openrouter.ai/Qwen2.5-72b-Inst` respectively.

For all the above LLMs, we use the below prompt for the Direct Prompt method, as given in `https://github.com/ServiceNow/context-is-key-forecasting`. **context**, **history** and **pred_time** are replaced by the respective textual context, numerical history and timestamps for which a forecast is required.

```
I have a time series forecasting task for you.

Here is some context about the task. Make sure to factor in any background knowledge,
satisfy any constraints, and respect any scenarios.
<context>
((context))
</context>

Here is a historical time series in (timestamp, value) format:
<history>
((history))
</history>

Now please predict the value at the following timestamps: ((pred_time)).

Return the forecast in (timestamp, value) format in between <forecast> and </forecast> tags.
Do not include any other information (e.g., comments) in the forecast.

Example:
<history>
(t1, v1)
(t2, v2)
(t3, v3)
</history>
<forecast>
(t4, v4)
(t5, v5)
</forecast>
```

### G.2 Lag-Llama

We use the publicly available implementation of Lag-Llama (Rasul et al., 2023) following the Instions at `https://github.com/time-series-foundation-models/`, on a single H100 GPU.

### G.3 Chronos

We use the publicly available implementation of Chronos-Large (Ansari et al., 2024) following the Instions at at `https://github.com/amazon-science/chronos-forecasting` on a single H100 GPU.

## G.4 ARIMA

We used the implementation of ARIMA from the `forecast` R package, using `rpy2`. Results are computed using the `auto.arima` method. We reran the model with restricted parameter and disabled seasonality if the ARIMA fit failed.

## H Cost and Inference Time of Methods

We report the inference time and cost of all experiments below. Note that cost only applies to models that were run using LLM APIs such as OpenRouter and OpenAI, and does not apply for models that were hosted locally.

## H.1 DP

| Metric | Total time taken | | Total Token Cost (USD) | | |
|---|---|---|---|---|---|
| Model | Average | Total | Average | Total | |
| With Context | | | | | |
| GPT-4o | 1m 26s | 28m 45s | 0.26 | 4.89 | |
| GPT-4o-mini | 2m 37s | 52m 22s | 0.03 | 0.50 | |
| Llama3-8B-Inst | 2m 10s | 43m 25s | - | - | |
| Llama3.1-405B-Inst | 2m 57s | 59m 2s | - | - | |
| Llama3.2-1B-Inst | 1m 33s | 31m 2s | - | - | |
| Llama3.2-3B-Inst | 2m 5s | 41m 44s | - | - | |
| Llama3.3-70B-Inst | 4m 39s | 1h 33m | - | - | |
| Qwen2.5-72B-Inst | 19m 22s | 6h 27m | 0.02 | 0.45 | |
| Qwen2.5-0.5B-Inst | 14m 55s | 4h 58m | - | - | |
| Qwen2.5-1.5B-Inst | 17m 34s | 5h 51m | - | - | |
| Qwen2.5-14B-Inst | 15m 23s | 5h 7m | - | - | |
| Qwen2.5-32B-Inst | 18m 9s | 6h 3m | - | - | |
| Qwen2.5-3B-Inst | 18m 15s | 6h 5m | - | - | |
| Qwen2.5-7B-Inst | 21m 43s | 7h 14m | - | - | |
| Without Context | | | | | |
| GPT-4o | 1m 25s | 28m 26s | 0.23 | 4.67 | |
| GPT-4o-mini | 2m 35s | 51m 43s | 0.03 | 0.57 | |
| Llama3-8B-Inst | 2m 11s | 43m 54s | - | - | |
| Llama3.1-405B-Inst | 2m 55s | 58m 33s | - | - | |
| Llama3.2-1B-Inst | 1m 35s | 31m 59s | - | - | |
| Llama3.2-3B-Inst | 2m 8s | 42m 50s | - | - | |
| Llama3.3-70B-Inst | 4m 31s | 1h 30m | - | - | |
| Qwen2.5-72B-Inst | 16m 13s | 5h 24m | 0.02 | 0.43 | |
| Qwen2.5-0.5B-Inst | 15m 0s | 5h 0m | - | - | |
| Qwen2.5-1.5B-Inst | 17m 30s | 5h 50m | - | - | |
| Qwen2.5-3B-Inst | 21m 43s | 7h 14m | - | - | |
| Qwen2.5-7B-Inst | 10m 27s | 3h 29m | - | - | |
| Qwen2.5-14B-Inst | 15m 4s | 5h 1m | - | - | |
| Qwen2.5-32B-Inst | 16m 38s | 5h 32m | - | - | |

Table 18: Cost of performing inference with the **DP** method. "Total" values represent the time (or) cost of running the models on all tasks in the CiK benchmark (Williams et al., 2025), "Average" values represent the average time (or) cost of running the models on a single task from the benchmark.

## H.2 ReDP

To produce the ground truth reasoning traces using the method given in Appendix C.2 using GPT-4.1, the total cost was **USD 0.0808**.

To perform the comparison of each model's reasoning trace with the ground truth reasoning trace, for the selected tasks used for the analysis, we incur a total cost of **USD 0.1774**. The cost per-model is provided in Table 19.

| Model | Total Cost (USD) |
|---|---|
| Llama3.3-70B-Inst | 0.0206 |
| Qwen2.5-72B-Inst | 0.0198 |
| Qwen2.5-7B-Inst | 0.0186 |
| Llama3.1-405B-Inst | 0.0185 |
| Qwen2.5-32B-Inst | 0.0177 |
| Qwen2.5-3B-Inst | 0.0158 |
| Qwen2.5-14B-Inst | 0.0157 |
| Llama3.1-8B-Inst | 0.0153 |
| Llama3.2-3B-Inst | 0.0151 |

Table 19: Total cost per model for reasoning correctness evaluation.

We report the time taken and cost incurred per model to produce forecasts using the ReDP method, in Table 20.

| Metric | Total time | | Total cost (USD) | |
|---|---|---|---|---|
| Stat | Avg | Total | Avg | Total |
| Model | | | | |
| Llama3.2-3B-Inst | 16m 35s | 4h 58m | - | - |
| Llama3.1-405B-Inst | 27m 57s | 9h 19m | 0.06 | 1.22 |
| Llama3.1-8B-Inst | 13m 16s | 2h 12m | 0.01 | 0.12 |
| Llama3.3-70B-Inst | 18m 44s | 6h 14m | 0.01 | 0.27 |
| Qwen2.5-72B-Inst | 19m 59s | 6h 39m | 0.02 | 0.46 |
| Qwen2.5-7B-Inst | 13m 52s | 4h 23m | 0.01 | 0.19 |
| Qwen2.5-14B-Inst | 37m 35s | 12h 31m | - | - |
| Qwen2.5-32B-Inst | 44m 3s | 14h 41m | - | - |
| Qwen2.5-3B-Inst | 1h 1m | 16h 25m | - | - |

Table 20: Cost per model for producing a forecast using the ReDP method.

## H.3 IC-DP

We report the time taken and cost incurred per model to produce forecasts using the ICDP method, in Table 21.

| Metric | Total time taken | | Total Token Cost (USD) | | |
|---|---|---|---|---|---|
| Model | Average | Total | Average | Total | |
| GPT-4o | 0m 55s | 1h 5m | 2.01 | 142.83 | |
| GPT-4o-mini | 1m 19s | 1h 34m | 0.01 | 0.95 | |
| Llama3.1-405B-Inst | 8m 34s | 10h 9m | 0.13 | 9.57 | |
| Llama3.3-70B-Inst | 6m 10s | 7h 18m | 0.02 | 1.15 | |
| Llama3-8B-Inst | 0m 45s | 53m 26s | - | - | |
| Llama3.2-1B-Inst | 3m 27s | 4h 5m | - | - | |
| Llama3.2-3B-Inst | 3m 43s | 4h 24m | - | - | |
| Qwen2.5-72B-Inst | 14m 8s | 16h 43m | 0.03 | 1.95 | |
| Qwen2.5-0.5B-Inst | 6m 20s | 7h 30m | - | - | |
| Qwen2.5-1.5B-Inst | 7m 0s | 8h 17m | - | - | |
| Qwen2.5-14B-Inst | 13m 59s | 16h 33m | - | - | |
| Qwen2.5-32B-Inst | 17m 1s | 20h 8m | - | - | |
| Qwen2.5-3B-Inst | 7m 59s | 9h 27m | - | - | |
| Qwen2.5-7B-Inst | 8m 40s | 10h 15m | - | - | |

Table 21: Cost of performing inference with the **IC-DP** method. "Total" values represent the time (or) cost of running the models on all tasks in the CiK benchmark (Williams et al., 2025), "Average" values represent the average time (or) cost of running the models on a single task from the benchmark.

## H.4 Cor-DP

We report the time taken and cost incurred per model to produce forecasts using the CorDP method, in Table 22 and Table 23.

| Metric | Total time taken | | Total Token Cost (USD) | | |
|---|---|---|---|---|---|
| Model | Average | Total | Average | Total | |
| **Base Forecaster: Arima** | | | | | |
| Llama3.1-405B-Inst | 8m 31s | 10h 4m | 0.03 | 2.39 | |
| Llama3.3-70B-Inst | 4m 51s | 5h 44m | - | - | |
| Llama3-8B-Inst | 2m 0s | 2h 22m | - | - | |
| Llama3.2-1B-Inst | 0m 59s | 1h 10m | - | - | |
| Llama3.2-3B-Inst | 1m 29s | 1h 46m | - | - | |
| Qwen2.5-72B-Inst | 14m 24s | 17h 2m | - | - | |
| Qwen2.5-0.5B-Inst | 1m 34s | 1h 51m | - | - | |
| Qwen2.5-1.5B-Inst | 1m 59s | 2h 21m | - | - | |
| Qwen2.5-14B-Inst | 6m 15s | 7h 24m | - | - | |
| Qwen2.5-32B-Inst | 8m 33s | 10h 7m | - | - | |
| Qwen2.5-3B-Inst | 3m 15s | 3h 51m | - | - | |
| Qwen2.5-7B-Inst | 2m 51s | 3h 19m | - | - | |
| GPT-4o | 0m 50s | 59m 39s | 0.07 | 4.96 | |
| GPT-4o-mini | 1m 15s | 1h 29m | 0.00 | 0.32 | |
| **Base Forecaster: Chronos-Large** | | | | | |
| Llama3.1-405B-Inst | 8m 21s | 9h 53m | 0.03 | 2.47 | |
| Llama3.3-70B-Inst | 4m 52s | 5h 46m | - | - | |
| Llama3-8B-Inst | 1m 57s | 2h 19m | - | - | |
| Llama3.2-1B-Inst | 1m 1s | 1h 10m | - | - | |
| Llama3.2-3B-Inst | 1m 28s | 1h 44m | - | - | |
| Qwen2.5-72B-Inst | 17m 55s | 21h 12m | - | - | |
| Qwen2.5-0.5B-Inst | 3m 42s | 3h 46m | - | - | |
| Qwen2.5-1.5B-Inst | 3m 31s | 4h 9m | - | - | |
| Qwen2.5-14B-Inst | 8m 28s | 10h 2m | - | - | |
| Qwen2.5-32B-Inst | 10m 57s | 12h 58m | - | - | |
| Qwen2.5-3B-Inst | 5m 2s | 5h 58m | - | - | |
| Qwen2.5-7B-Inst | 2m 54s | 3h 23m | - | - | |
| GPT-4o | 0m 50s | 1h 0m | 0.07 | 5.06 | |
| GPT-4o-mini | 1m 18s | 1h 32m | 0.00 | 0.31 | |
| **Base Forecaster: Lag-Llama** | | | | | |
| Llama3.1-405B-Inst | 8m 16s | 9h 47m | 0.03 | 2.43 | |
| Llama3.3-70B-Inst | 4m 44s | 5h 36m | - | - | |
| Llama3-8B-Inst | 1m 56s | 2h 17m | - | - | |
| Llama3.2-1B-Inst | 0m 57s | 1h 7m | - | - | |
| Llama3.2-3B-Inst | 1m 27s | 1h 43m | - | - | |
| Qwen2.5-72B-Inst | 17m 37s | 20h 51m | - | - | |
| Qwen2.5-0.5B-Inst | 2m 57s | 3h 29m | - | - | |
| Qwen2.5-1.5B-Inst | 3m 23s | 4h 0m | - | - | |
| Qwen2.5-14B-Inst | 9m 29s | 11h 14m | - | - | |
| Qwen2.5-32B-Inst | 13m 17s | 15h 43m | - | - | |
| Qwen2.5-3B-Inst | 6m 0s | 7h 6m | - | - | |
| Qwen2.5-7B-Inst | 2m 47s | 3h 15m | - | - | |
| GPT-4o | 0m 52s | 1h 2m | 0.07 | 4.86 | |
| GPT-4o-mini | 1m 32s | 1h 49m | 0.00 | 0.33 | |

Table 22: Cost of performing inference with the **Median-CorDP** method. "Total" values represent the time (or) cost of running the models on all tasks in the CiK benchmark (Williams et al., 2025), "Average" values represent the average time (or) cost of running the models on a single task from the benchmark.

| Metric | Total time taken | | Total Token Cost (USD) | | |
|---|---|---|---|---|---|
| Stat | Avg | Total | Avg | Total | |
| Model | | | | | |
| **Base Forecaster: Arima** | | | | | |
| Llama3.1-405B-Inst | 8m 32s | 10h 6m | 0.03 | 2.38 | |
| Llama3.3-70B-Inst | 42m 39s | 50h 29m | - | - | |
| Llama3-8B-Inst | 14m 22s | 16h 46m | - | - | |
| Llama3.2-1B-Inst | 5m 25s | 6h 20m | - | - | |
| Llama3.2-3B-Inst | 8m 37s | 10h 3m | - | - | |
| Qwen2.5-72B-Inst | 1h 6m | 79h 6m | - | - | |
| Qwen2.5-0.5B-Inst | 9m 43s | 11h 30m | - | - | |
| Qwen2.5-1.5B-Inst | 11m 28s | 13h 23m | - | - | |
| Qwen2.5-14B-Inst | 31m 37s | 36h 53m | - | - | |
| Qwen2.5-32B-Inst | 45m 5s | 52h 36m | - | - | |
| Qwen2.5-3B-Inst | 21m 3s | 24h 55m | - | - | |
| Qwen2.5-7B-Inst | 13m 34s | 15h 50m | - | - | |
| GPT-4o | 2m 34s | 3h 2m | 0.14 | 9.62 | |
| GPT-4o-mini | 4m 20s | 5h 8m | 0.01 | 0.59 | |
| **Base Forecaster: Chronos-Large** | | | | | |
| Llama3.1-405B-Inst | 8m 29s | 10h 3m | 0.03 | 2.34 | |
| Llama3.3-70B-Inst | 42m 38s | 50h 27m | - | - | |
| Llama3-8B-Inst | 14m 45s | 17h 28m | - | - | |
| Llama3.2-1B-Inst | 5m 39s | 6h 19m | - | - | |
| Llama3.2-3B-Inst | 8m 45s | 10h 22m | - | - | |
| Qwen2.5-72B-Inst | 1h 8m | 80h 29m | - | - | |
| Qwen2.5-0.5B-Inst | 1m 23s | 1h 38m | - | - | |
| Qwen2.5-1.5B-Inst | 12m 5s | 14h 18m | - | - | |
| Qwen2.5-14B-Inst | 32m 6s | 37h 27m | - | - | |
| Qwen2.5-32B-Inst | 45m 42s | 53h 19m | - | - | |
| Qwen2.5-3B-Inst | 21m 21s | 25h 16m | - | - | |
| Qwen2.5-7B-Inst | 13m 48s | 16h 20m | - | - | |
| GPT-4o | 2m 30s | 2h 58m | 0.14 | 9.61 | |
| GPT-4o-mini | 4m 20s | 5h 7m | 0.01 | 0.60 | |
| **Base Forecaster: Lag-Llama** | | | | | |
| Llama3.1-405B-Inst | 8m 28s | 10h 1m | 0.03 | 2.30 | |
| Llama3.3-70B-Inst | 41m 48s | 49h 27m | - | - | |
| Llama3-8B-Inst | 14m 14s | 16h 37m | - | - | |
| Llama3.2-1B-Inst | 5m 5s | 5h 51m | - | - | |
| Llama3.2-3B-Inst | 8m 31s | 9h 57m | - | - | |
| Qwen2.5-72B-Inst | 1h 5m | 77h 24m | - | - | |
| Qwen2.5-0.5B-Inst | 9m 8s | 10h 49m | - | - | |
| Qwen2.5-1.5B-Inst | 11m 15s | 13h 8m | - | - | |
| Qwen2.5-14B-Inst | 30m 56s | 36h 6m | - | - | |
| Qwen2.5-32B-Inst | 43m 59s | 51h 19m | - | - | |
| Qwen2.5-3B-Inst | 20m 28s | 24h 13m | - | - | |
| Qwen2.5-7B-Inst | 13m 4s | 15h 15m | - | - | |
| GPT-4o | 2m 30s | 2h 58m | 0.13 | 9.39 | |
| GPT-4o-mini | 4m 10s | 4h 55m | 0.01 | 0.59 | |

Table 23: Cost of performing inference with the **SampleWise-CorDP** method. "Total" values represent the time (or) cost of running the models on all tasks in the CiK benchmark (Williams et al., 2025), "Average" values represent the average time (or) cost of running the models on a single task from the benchmark.

## H.5    RouteDP

We report the time taken and cost incurred per model to produce the difficulty score of tasks using the RouteDP method, in Table 24.

| Model | Average (s) | Total (s) |
|---|---|---|
| Qwen2.5-0.5B-Inst | 0.18 | 12.48 |
| Qwen2.5-1.5B-Inst | 0.17 | 12.10 |
| Qwen2.5-3B-Inst | 0.19 | 13.58 |
| Qwen2.5-7B-Inst | 0.23 | 16.31 |
| Qwen2.5-14B-Inst | 0.28 | 19.90 |
| Qwen2.5-32B-Inst | 0.48 | 34.35 |

Table 24: Time taken to compute the difficulty scores of tasks with the **RouteDP** method. "Total" values represent the time to compute the score for all tasks in the CiK benchmark (Williams et al., 2025), "Average" values represent the average time taken to compute the score for a single task from the benchmark.

