# OpenReview forum: "Beyond Naïve Prompting: Strategies for Improved Zero-shot Context-aided Forecasting with LLMs"
_TMLR — Rejected by TMLR_

### Review · Reviewer_ZYnx · 2025-08-29

**Summary Of Contributions:**

This paper focuses on the problem of context-aided forecasting using LLMs in the zero-shot setting. They note that in the zero-shot setting, previous work focus on simple and naive prompting, ignoring more advanced techniques. The authors thus propose 4 new strategies for prompting LLMs for this task. They include: (a) ReDP: Prompting models to include their reasoning traces. (b) CorDP: Asking LLMs to improve upon forecasts of other models. (c) IC-DP: Uses in-context learning (i.e., historical examples) when available to provide additional context to the LLM. (d) RouterDP: Uses a router that defers to a smaller LLM for easier questions and a larger one for more complicated problems. This allows for a reduction in resources while hopefully minimizing any loss in performance. The authors test each of these 4 methods on both Llama3.3 and Qwen2.5 models of various sizes. They test them on the Context-Is-Key (CiK) benchmark, that is meant to evaluate forecasts when the context is necessary for proper forecasting. The authors show that each strategy has the potential to improve upon naive prompting.

### Strengths:

1. The paper is well written and very clear. The authors do a good job of describing their motivation, why it matters, their experimental settings, and their results. This makes the paper a really enjoyable read.

2. The authors are fairly thorough in the type of strategies they propose. As in, they cover a variety of different approaches that one would logically try (e.g., listing the reasoning trace or in-context learning). Overall, I think this gives people a good idea of what does and doesn't work beyond direct prompting.

3. The use of the CiK datasets are appreciated, since they are much more indicative of real-world performance context does actually matter.


### Weaknesses:

1. For the ReDP strategy, it's unclear to me how the authors are able to truly verify the reasoning processes given by the LLM. There are several issues. First, we don't have actually ground-truth traces. As the authors note, they are generated by GPT 4.1. However, how do we know if they are actually correct? The authors say that they manually verified them and modified "them if required" (Appendix C.2). However, no explanation is given for how they were verified or modified. To be honest, I have no idea how we can be sure at all that the ground-truth are correct at all. The authors need to be more detailed in their process here.

2) In addition to the previous weakness, LLM-as-a-judge has been shown to be quite biased in multiple ways [1] (e.g., position, length, LLM). As such, beyond just the veracity of the ground-truth traces, evaluating them via LLM-as-a-judge itself often has problems. I understand the limitations one has when evaluating results for many models at this scale, but it's important that the authors consider these downsides.

3) This is not a major weakness, but it would be interesting if the authors tried using more than one strategy at a time. For example, ReDP+IC-DP. This is helpful because it gives us an idea if their improvement is orthogonal to one another or overlap considerably. Also, in real-world situations, people will often try to use as many improvement strategies as possible to boost performance. So it makes sense to test the combination of some of these methods.


4) This is a minor weakness, but additional LLMs could be considered. In many real-world setting, a model like ChatGPT (or others from OpenAI) are often used. This is more pertinent when we consider that this paper focuses on the zero-shot setting, which applies to closed-sourced models like ChatGPT very well. I think if it is cost feasible, the authors should try to include GPT in their work (note that GPT was also included in the original CiK paper).


[1] Justice or Prejudice? Quantifying Biases in LLM-as-a-Judge, ICLR'25.

**Audience:**

Yes

**Audience Explanation:**

LLM forecasting has become a popular topic. I therefore think a lot of people would be interested in these results.

**Broader Impact Concerns:**

There are no concerns.

**Claims And Evidence:**

Yes

**Claims Explanation:**

Overall, the claims are well supported. As noted in my review, my one concern is with their classification of correct and wrong reasoning traces. I'm unsure how reliable their method is for determing reasoning veracity.

**Requested Changes:**

1) Provide more detail on how the reasoning traces are verified and modified for the first strategy (see weakness 1 in my review for more). This is critical.
2) Try testing multiple strategies in tandem with one another. I of course don't expect all possible combinations, but, for example, testing ReDP with IC-DP and CorDP could be enlightening and tell us if each strategy really is necessary. This is not critical but I do think it's important.
3) Include additional LLMs in analysis (such as ChatGPT). This is minor.

---

> ### Author Response · Authors · 2025-09-19
> **Response to review - 1**
>
> Thank you for your detailed review of our work. We are glad that you found the paper well written and clear, and appreciate your recognition of the thoroughness of our work. Below, we clarify all the points raised in the review.
>
> (Weakness abbreviated as **W**, Requested Change abbreviated as **RC**)
>
> ## W-1/RC-1: Verification of ground truth reasoning traces
>
> We thank the reviewer for this question. We take this opportunity to expand on how we verified the ground truth reasoning traces.
> Once the ground truth reasoning traces are generated, for each task, we (the authors) looked at the context of the task and the ground truth reasoning trace, and verified the following:
> 1. If the ground truth reasoning trace captured the key change in forecast that the context brings about.
> 2. If the dates and duration of the context-induced change is accurate in the ground truth reasoning trace.
> 3. The ground truth reasoning trace is crisp, and only contains the direct impact of the context
>
> E.g. For the ATMBuildingClosedTask with context that says
> “Background: This is the number of cash withdrawals from an automated teller machine (ATM) in an arbitrary location in England. Constraints: None. Scenario: Consider that the building which contains the ATM is closed from 1996-11-24 00:00:00, for 10 days.”,
> We verified that the reasoning trace indicates that
> 1. There will be a period in the forecast where the number of withdrawals would be zero
> 2. The said period would be from 1996-11-24 00:00:00 for 10 days
> 3. The reasoning trace does not mention anything else other than the two points above
>
> As per the above verification method, we found that the ground truth reasoning traces met the above criteria for all tasks except 2 namely the SensorMaintenanceInPredictionTask (where criterion 1 was not met) and the DecreaseInTrafficInPredictionTask (where criterion 2 was not met). Hence, we adapted them to meet the criteria.
> We have added a discussion of this to the paper in App. C.4.1, and are happy to clarify any outstanding concerns regarding this.
>
> ---
>
> If the reviewer is curious about the exact modification made with the two tasks, for completeness, we write here the original and modified ground truth reasoning traces:
>
> ### SensorMaintenanceInPredictionTask:
>
> **Context:** Background: This series represents the occupancy rate (%) captured by a highway sensor. Constraints: None. Scenario: Consider that the meter will be offline for maintenance between 2024-01-18 08:00:00 and 2024-01-18 14:00:00, which results in zero readings.
>
> **Original reasoning trace:** A logical forecast would interpolate or exclude the zero readings during the offline maintenance period and use historical occupancy patterns from similar time windows on previous days to estimate the expected occupancy rate.
>
> > This does not meet criterion 1 as it is the wrong interpretation of the context, as the mentioned dates are in the prediction horizon and not in the historical data.
>
> **Modified reasoning trace:** The context mentions that the meter will be offline for maintenance between 2024-01-18 08:00:00 and 2024-01-18 14:00:00. During this period, one should forecast a value of 0%, as the meter will not be capturing any data.
>
> ### DecreaseInTrafficInPredictionTask:
>
> **Context:**  Background: This is hourly traffic data. Constraints: None. Scenario: Suppose that there is an accident on the road and there is 20.0% of the usual traffic from 2024-01-18 06:00:00 for 5 hours.
>
> **Original reasoning trace:** To forecast traffic during the accident, multiply the usual hourly traffic by 0.2 for each hour from 2024-01-18 06:00:00 to 2024-01-18 10:00:00, since the accident reduces traffic to 20% of normal levels for 5 hours.
>
> > This does not meet criterion 2 as the end time is incorrectly interpreted as 10:00:00 while it is supposed to be 11:00:00.
>
> **Modified reasoning trace:** To forecast traffic during the accident, multiply the usual hourly traffic by 0.2 for each hour from 2024-01-18 06:00:00 to 2024-01-18 11:00:00, since the accident reduces traffic to 20% of normal levels for 5 hours.
>
> ---
>
> As mentioned above, this discussion is included in App C.4.1 of the revised paper, and we are happy to clarify any further concerns.

---

> ### Author Response · Authors · 2025-09-19
> **Response to review - 2**
>
> ## W-2: Biases of LLM-as-a-judge
>
> We thank the reviewer for pointing us to the potential biases of the LLM-as-a-judge approach. We agree with the reviewer on the biases of the approach used to compare the ground truth reasoning traces with the models’ reasoning traces.
>
> To verify this approach, we ran a human evaluation. The evaluators see the same information as the LLM judge, namely the ground truth reasoning trace, and a model’s reasoning trace, to answer the same question of whether the model’s reasoning is aligned with the key points mentioned in the ground truth reasoning (yes / no). Note that the evaluators do not see the results of the LLM-as-a-judge, and the human evaluation was done independently. We run this for all the tasks and all the models considered in the experiment, and measure the agreement of the evaluators and the LLM-judge on the reasoning correctness as well as the meaningful improvement of the forecast with context.
>
> On average, we find that a high agreement of 89.9% on the reasoning correctness, i.e. 89.9% of the time, the model and the humans agree on the correctness of reasoning of the model. This shows that the LLM-as-a-judge approach (specifically with the GPT-4.1 model) may suffice as a scalable alternative compared to human evaluation for future models.
>
> We add a discussion on this to the paper in App. C.6, along with the statistics of agreement for each model under study. We plan to also open-source the results of the human evaluation and the LLM-as-a-judge, for transparency.
>
> We thank the reviewer again for raising this point which led us to conduct this human evaluation. We hope this addresses the concern raised by the reviewer, and are happy to clarify this further.

---

> ### Author Response · Authors · 2025-09-19
> **Response to review - 3**
>
> ## W-3/RC-2: Multiple strategies in tandem
>
> Thank you for suggesting this experiment. Following your request, we performed two experiments.
>
> 1. We combine ICDP and CorDP: this method uses CorDP as the foundation: the goal is to output a forecast given the history, context and a base forecast; when combined with IC-DP, it uses an in-context example that contains the history, context, base forecast and ground truth of the example. We abbreviate this hybrid method as IC-CorDP (In-Context Corrector Direct Prompt).
> Due to the time constraints, we use the Median-CorDP for this experiment and run experiments with a subset of LLMs from the paper. As in CorDP, we test it with multiple base forecasters (Lag-Llama, Chronos-Large, and ARIMA).
>
> Below is **Table 12 from the paper:** Aggregate results of the hybrid method IC-Median-CorDP on CiK, accompanied by standard errors. The best performing method for each model is in bold.

---

> ### Author Response · Authors · 2025-09-19
> **Response to review - 4**
>
> | **Model** | **Direct Prompt (DP)** | | **Median Corrector (Median-CorDP)** |  |  |               | **In-Context Median Corrector (IC-Median-CorDP)** | | | |
> |----|----|------|-----|------------------------|----|--|--|--|--|--|
> |                                         | |      | Lag-Llama                                                        | Chronos Large | ARIMA                                                                          |   | Lag-Llama         | Chronos Large     | ARIMA    |
> | **Llama3.2-1B-Inst**                                             | 0.396 $\pm$ 0.027                                         |      | 0.394 $\pm$ 0.004                                                       | 0.515 $\pm$ 0.007      | 0.612 $\pm$ 0.018  |   | **0.315 $\pm$ 0.004** | 0.390 $\pm$ 0.031          | 0.480 $\pm$ 0.010 |
> | **Llama3.2-3B-Inst**                                             | 0.687 $\pm$ 0.025                                         |      | 0.344 $\pm$ 0.011                                                       | 0.455 $\pm$ 0.009      | 0.573 $\pm$ 0.022                                                                       |   | **0.334 $\pm$ 0.008** | 0.354 $\pm$ 0.011          | 0.478 $\pm$ 0.016 |
> | **Qwen2.5-0.5B-Inst**                                            | 0.592 $\pm$ 0.027                                         |      | 0.633 $\pm$ 0.002                                                         | 0.801 $\pm$ 0.003      | 0.761 $\pm$ 0.054                                                                       |   | **0.358 $\pm$ 0.005** | 1.734 $\pm$ 0.008          | 0.548 $\pm$ 0.010 |
> | **Qwen2.5-1.5B-Inst**                                            | 0.616 $\pm$ 0.018                                         |      | 0.426 $\pm$ 0.013                                                       | 0.537 $\pm$ 0.003      | 0.682 $\pm$ 0.006                                                                       |   | **0.305 $\pm$ 0.004** | 0.390 $\pm$ 0.028          | 0.334 $\pm$ 0.009 |
> | **Qwen2.5-3B-Inst**                                              | 0.424 $\pm$ 0.017                                         |      | 0.490 $\pm$ 0.005                                                         | 0.491 $\pm$ 0.004      | 0.597 $\pm$ 0.009                                                                       |   | **0.326 $\pm$ 0.008** | 0.475 $\pm$ 0.009          | 0.399 $\pm$ 0.013 |
> | **Qwen2.5-7B-Inst**                                              | 0.401 $\pm$ 0.006                                         |      | 0.419 $\pm$ 0.004                                                         | 0.641 $\pm$ 0.008      | 0.633 $\pm$ 0.008                                                                       |   | **0.322 $\pm$ 0.008** | 0.334 $\pm$ 0.009          | 0.449 $\pm$ 0.010 |
> | **Qwen2.5-14B-Inst**                                             | **0.247 $\pm$ 0.006**                                |      | 0.315 $\pm$ 0.003                                                         | 0.334 $\pm$ 0.006      | 0.423 $\pm$ 0.004                                                                       |   | 0.256 $\pm$ 0.006          | 0.293 $\pm$ 0.006          | 0.336 $\pm$ 0.010 |
> | **Qwen2.5-32B-Inst**                                             | 0.397 $\pm$ 0.008                                         |      | **0.248 $\pm$ 0.004**                                                | 0.272 $\pm$ 0.005      | 0.329 $\pm$ 0.008                                                                       |   | 0.261 $\pm$ 0.005        | 0.261 $\pm$ 0.007          | 0.383 $\pm$ 0.009 |
> | **Qwen2.5-72B-Inst**                                             | 0.202 $\pm$ 0.009                                         |      | 0.319 $\pm$ 0.008                                                         | 0.358 $\pm$ 0.010      | 0.428 $\pm$ 0.009                                                                       |   | 0.233 $\pm$ 0.005          | **0.180 $\pm$ 0.005** | 0.400 $\pm$ 0.008 |
> | **Llama3.1-405B-Inst**                                           | **0.173 $\pm$ 0.003**                                |      | 0.278 $\pm$ 0.009                                                         | 0.226 $\pm$ 0.004      | 0.257 $\pm$ 0.008    |   | 0.227 $\pm$ 0.006          | 0.308 $\pm$ 0.006          | 0.243 $\pm$ 0.012 |
> | **Base Forecaster**  | -  |      | 0.382 $\pm$ 0.011  | 0.492 $\pm$ 0.004| 0.636 $\pm$ 0.014 || 0.382 $\pm$ 0.011| 0.492 $\pm$ 0.004 | 0.636 $\pm$ 0.014 |
>
> We find that IC-Median-CorDP improves performance compared to Median-CorDP across LLMs across all sizes, and across multiple base quantitative forecasters that the LLM bootstraps over. The levels of gains achieved with IC-Median-CorDP depend on the LLM and the base quantitative forecaster. This shows that there is clear potential in combining the two strategies to improve performance.

---

> ### Author Response · Authors · 2025-09-19
> **Response to review - 5**
>
> 2. We combine RouteDP with IC-DP and CorDP: here, routing is done on forecasts produced by either the CorDP or IC-DP method, similar to how it is computed on forecasts of DP. We use the same large model (Llama-3.1-405B-Inst), and test various main models and router models. We observe improvements similar to with DP, across several router models and main models. This shows that the difficulties predicted by the router model may not depend on the downstream strategy (DP, IC-DP, CorDP etc.) employed. We present the results in App. F.4 (Fig 84, 85), and in Tables 16-17.
>
> We hope this answers your questions. Please let us know if you need further clarification.

---

> ### Author Response · Authors · 2025-09-19
> **Response to review - 6**
>
> ## W-4/RC-3: Performance of closed-source LLMs
>
> Thank you for suggesting this experiment. Following your request, we used the CorDP and IC-DP methods with GPT-4o and GPT-4o-mini. Similar conclusions hold for these models: Both GPT-4o and GPT-4o-mini improve with CorDP methods; SampleWise-CorDP methods achieve the best performance with both models. Both models improve significantly with IC-DP as well. We paste a concise view of the new results here:
>
> ### Results With CorDP:
>
> | Model           | DP                | Median-CorDP (Lag-Llama) | Median-CorDP (Chronos-Large) | Median-CorDP (ARIMA) | SampleWise-CorDP (Lag-Llama) | SampleWise-CorDP (Chronos-Large) | SampleWise-CorDP (ARIMA) |
> |-----------------|-------------------|--------------------------|------------------------------|----------------------|------------------------------|----------------------------------|--------------------------|
> | GPT-4o          | 0.317 $\pm$ 0.009 | 0.253 $\pm$ 0.004        | 0.240 $\pm$ 0.004            | 0.354 $\pm$ 0.007    | **0.184** $\pm$ **0.004**            | 0.196 $\pm$ 0.004                | 0.251 $\pm$ 0.008        |
> | GPT-4o-mini     | 0.389 $\pm$ 0.010 | 0.364 $\pm$ 0.006        | 0.340 $\pm$ 0.004            | 0.516 $\pm$ 0.005    | 0.302 $\pm$ 0.008            | **0.296** $\pm$ **0.005**                | 0.415 $\pm$ 0.011        |
> |                 |                   |                          |                              |                      |                              |                                  |                          |
> | Base Forecaster | -                 | 0.382 ± 0.011            | 0.492 ± 0.004                | 0.636 ± 0.014        | 0.382 ± 0.011                | 0.492 ± 0.004                    | 0.636 ± 0.014            |
>
> ### Results with IC-DP:
>
> | Model       | DP                | IC-DP             |
> |-------------|-------------------|-------------------|
> | GPT-4o      | 0.317 $\pm$ 0.009 | **0.164 $\pm$ 0.005** |
> | GPT-4o-mini | 0.389 $\pm$ 0.010 | **0.253 $\pm$ 0.004** |
>
> We have added these results to the paper, in the respective tables and figures concerning each method in the main text and appendix. We also report the cost of these models in the newly created App. H.
>
> We hope this answers your question, and are happy to clarify this further.

---

### Review · Reviewer_SYqf · 2025-09-02

**Summary Of Contributions:**

## Summary
The paper investigates how large language models can be used for context-conditioned probabilistic time-series forecasting, where historical values and textual context jointly determine predictive distributions of future outcomes. It introduces four strategies beyond direct prompting: ReDP, which elicits a short rationale before producing forecasts; CorDP, which adjusts outputs from an existing base forecaster using context; IC-DP, which provides a worked example with history, context, and future values; and RouteDP, which routes only difficult tasks to larger models. Experiments are conducted on the CiK benchmark with 71 zero-shot tasks across multiple domains and temporal scales, using LLMs of varying sizes and families. Evaluation relies on RCRPS, a constraint-augmented and ROI-aware variant of CRPS. Results show analyses of reasoning quality under ReDP, improvements from CorDP especially when paired with strong baselines, consistent gains from IC-DP across model sizes, and favorable compute–accuracy trade-offs achieved by RouteDP.

## Strengths
1. This work offers a clear formulation of context-conditioned probabilistic forecasting, framing it as the task of mapping historical sequences together with textual context into predictive distributions. The consistent setup makes experimental comparisons and ablations straightforward.

2. A notable strength is the evaluation framework, which captures deployment risks by emphasizing ROI-aware and constraint-sensitive scoring rather than relying only on plain CRPS. This focus highlights the kinds of errors that matter in realistic applications.

3. Another positive aspect is the modular design of the strategies, which include reasoning audits, context-based corrections, in-context examples, and routing mechanisms. Because the components can be applied independently, researchers and practitioners can integrate or test them flexibly.

4. Beyond simply displaying reasoning traces, the study explicitly measures whether contextual logic is correctly inferred and whether it flows into calibrated numeric predictions. This distinction makes sources of error more visible and allows a clearer diagnosis of model behavior.

5. The experimental coverage spans models of very different scales and examines both direct prompting and combinations with strong base forecasters. Such breadth provides useful insight into trade-offs between model size, computational cost, and forecasting accuracy.

## Weakness
### Evaluation setup & measurement
1. This paper lets GPT-4.1 author the “gold” reasoning (lightly edited by humans) and then asks GPT-4.1 again to grade model outputs. That tight coupling between the evaluator and the systems being evaluated invites style bias and makes replication fragile. I would expect either inter-annotator agreement with humans or cross-evaluator checks with a different model family.

2. In the CiK benchmark each task is instantiated only once. With one realization, RCRPS variability and stability are likely underestimated; the tables do report standard errors, but task-level sampling is thin, so we don’t really see within-task variance or how fragile rankings are across realizations.

3. The constraint penalty uses β=10 everywhere, inherited from the benchmark. Because β directly steers rankings, a sensitivity sweep (e.g., β∈{1,5,10,20}) is necessary; otherwise a method may look “good at constraints” simply because it matches that single β.

4. The paper analyzes reasoning quality on only the ~20 tasks with a clear ROI and a single “correct” reasoning template. That choice sidelines harder settings where the entire horizon is affected or multiple valid explanations exist, and it likely inflates the apparent interpretability and transfer of ReDP/DP.

### Method
1. Most of the big wins arrive when CorDP is paired with strong bases like Lag-Llama; the gains can then be partially credited to the base itself rather than the correction strategy. At the same time, the smallest LLMs occasionally make the base forecast worse, which raises questions about when CorDP is safe to deploy.

2. The paper acknowledges that adding an in-context example increases input length and cost, but there is no token-cost vs. accuracy curve and no guidance on the optimal number of examples. Without that, practitioners can’t trade performance for spend.

3. Task “difficulty” comes from a zero-shot LLM score with no objective, auditable target or human-rated yardstick. That subjectivity makes mis-routing hard to diagnose and can hide systematic failures.

4. Everything is zero-shot prompting. There is no systematic comparison against light finetuning or instruction-tuning plus the same strategies, so we don’t know whether small amounts of supervised adaptation would dominate the reported prompting tricks.

### ReDP
1. Gold rationales are required to be a concise single sentence. That format naturally favors “single-step ROI transforms” and under-probes multi-stage numeric chains (date arithmetic, branching conditions, compositional adjustments).

2. Reasoning correctness is tied to the first drawn sample. This can produce awkward artifacts: a model might reason correctly yet miss in that one sample, or reason poorly but land a lucky sample. A set-level check (consistency across samples, or sample-weighted scoring) would be fairer.

3. To satisfy output formatting, the setup avoids constrained decoding and instead retries up to 15 times on failure. Models that adhere to the format more easily (or that benefit more from retries) gain an evaluation advantage, and the retry policy itself is misaligned with low-latency production constraints.

### IC-DP
1. The in-context example explicitly contains the future values of the example task, which acts like strong supervision inside a “zero-shot” envelope. If the target task is distributionally close to the example, template transfer can inflate gains; robustness checks should swap examples or use cross-type examples.

2. Using one fixed example task (e.g., SensorTrendAccumulationTask) to teach the model may not generalize to heterogeneous families (non-trend, multi-factor compositions). Multiple or retrieved examples—and ablations on example choice—would clarify generality.

### RouteDP
The router predicts only easy vs. hard and reports “area captured” against random/ideal. There is no correlation report (e.g., Kendall-τ) between difficulty scores and actual RCRPS improvements. In extensions, the captured area swings wildly—including extremely low values—suggesting shaky stability and limited practical usefulness.

### Data coverage & external validity
1. Seven domains and multiple frequencies are helpful, but 71 tasks remain modest in scope. We still don’t know how robust the methods are to long contexts, noisy or partially irrelevant text, or scenarios with multiple plausible explanations—the paper itself flags this limitation.

2. The authors note that each task has a single history/future instantiation. That design pushes variance across tasks to dominate and leaves task difficulty estimates unstable. A more convincing setup would resample implementations per template and report confidence bands.

### Knowledge and retrieval gap

Real-world contexts often require retrieval or verification (weather archives, calendars, policies). This paper assumes the given text is sufficient and does not incorporate RAG or any knowledge-checking step, which limits realism for many deployments.

**Audience:**

Yes

**Audience Explanation:**

The findings of this paper would be of interest to a portion of TMLR’s readership. The study examines context-conditioned probabilistic time-series forecasting with large language models, a problem that lies at the intersection of sequence modeling, probabilistic prediction, and LLM adaptation. This is an active research area within the TMLR community, where many readers are interested in extending LLMs beyond text-only tasks. The systematic comparison of prompting strategies provides useful insights into both the potential and the limitations of applying LLMs to structured forecasting problems.

The benchmark used in the work (CiK) and the four structured prompting methods are directly relevant to researchers studying time-series forecasting, multimodal integration, and evaluation of probabilistic models. The results are also likely to attract practitioners looking for ways to combine established forecasting techniques with general-purpose LLMs. In particular, the emphasis on modular strategies and compute accuracy trade-offs offers practical guidance for applied groups seeking to balance performance and cost. For these reasons, the paper will be of clear interest to segments of TMLR’s audience.

**Broader Impact Concerns:**

This work develops prompting strategies for using large language models in context-conditioned probabilistic time-series forecasting. While the contribution is methodological, there are several broader implications worth raising.

One issue is evaluation bias. The paper relies on LLMs not only as forecasting engines but also in generating and assessing reasoning traces. When the same model family provides both the “gold” reference and the judge, there is a risk that similarity in phrasing is rewarded rather than actual correctness. This could affect reproducibility and fairness of evaluation, and it would be helpful for the authors to acknowledge this limitation and outline possible mitigations.

Another concern is robustness of context use. The benchmark assumes that the textual context is clean and necessary for solving the task. In real applications, context information is often incomplete, noisy, or even adversarial. Forecasting systems that follow this paradigm may therefore be vulnerable to manipulation or biased inputs, with potentially serious consequences in domains such as healthcare, finance, or critical infrastructure. A discussion of safeguards or robustness testing would strengthen the impact statement.

In summary, the paper raises no immediate high-risk ethical concerns, but it would benefit from a more explicit treatment of evaluation bias, robustness to imperfect context, and the environmental and equity impacts of heavy reliance on large models.

**Claims And Evidence:**

Yes

**Claims Explanation:**

The main claims in the paper are backed up with solid evidence, though there are places where the support could be stronger. The authors argue that large language models can handle context-conditioned probabilistic forecasting effectively when guided by structured prompting strategies. They introduce four such strategies and test them on the CiK benchmark, which spans 71 zero-shot tasks across several domains. The evaluation metric is tailored to account for ROI segments and constraint violations, and the results consistently show that the proposed methods work as advertised. For example, IC-DP improves performance across models of very different scales, CorDP works particularly well when paired with strong base forecasters like Lag-Llama, and RouteDP delivers the kind of compute–accuracy trade-offs the authors claim.

That said, there are also a few weaknesses in the evidence. Each task in CiK is only instantiated once, which makes it hard to judge variability or statistical stability. The fixed choice of β=10 in the RCRPS metric is never stress-tested, so we do not know if the method rankings would hold up under different penalty weights. The reasoning analysis is limited to a small subset of tasks with only one “correct” rationale, which probably paints an overly optimistic picture of interpretability. In IC-DP, the example prompt even includes future ground truth, which makes the gains look larger than what one would expect in a real setting.

**Requested Changes:**

1. The CiK benchmark currently evaluates each task with only a single realization of history and future. This makes it difficult to assess variability and can underestimate uncertainty in the reported results. The paper would be significantly strengthened if the authors included multiple realizations per template or, at minimum, reported variance estimates across repeated instantiations.

2. All results rely on a fixed choice of β=10 for the constraint penalty in RCRPS. Because this parameter directly influences task rankings, a sensitivity study across a range of β values (e.g., 1, 5, 10, 20) is necessary to confirm that the conclusions are robust.

3. The current reasoning analysis is restricted to a subset of ROI tasks with a single “correct” rationale. This design inflates the interpretability findings and does not cover harder settings. Extending the analysis to tasks with multiple plausible rationales or full-window context dependence would make the evaluation more representative.

4. IC-DP includes future ground truth from the example task, which effectively provides strong supervision in a supposed zero-shot setting. The paper should more clearly position this design choice, discuss its implications, and ideally test alternative setups where examples exclude future values.

5.  Task difficulty is assigned via zero-shot LLM judgments without independent verification. The authors should provide correlation analyses between predicted difficulty and observed error, and clarify how stable the routing decisions are across runs. This would make the compute–accuracy trade-off claims more convincing.

6. Several strategies (IC-DP, RouteDP, retry-based decoding) have direct cost implications. The paper would benefit from explicit reporting of token usage or runtime overhead, along with cost–accuracy curves, so that readers can judge trade-offs in practical settings.

7. The current tasks are designed so that textual context is always necessary and accurate. Adding experiments with noisy, irrelevant, or misleading context would provide a more realistic assessment of robustness.

---

> ### Author Response · Authors · 2025-09-19
> **Response to review - 1**
>
> Thank you for your very detailed review of our work. We appreciate your recognition of the modular design of the strategies, our evaluation framework, our error analysis with reasoning traces, and the empirical depth of our work. Below, we clarify all the points raised in the review.
>
> (Requested Change abbreviated as **RC**)
>
> ## RC-1: Choice of the number of instances of each task from the CiK benchmark
>
> Thank you for raising this point. We opted for a single instance of all tasks of CiK after communication with the authors of CiK that it would be sufficient to evaluate methods on, as there is significant diversity due to the various tasks. Further, due to the inference time and cost involved in running all the LLMs with the proposed methods (reported in App. H), we could not unfortunately not run the models with more instances of CiK before the rebuttal deadline, as our GPU compute was occupied running the other experiments that we present in the newer version of the paper. However, if required, we are more than happy to run another instance of CiK tasks, with a smaller scale of LLMs and methods, for the camera-ready.
>
> ## RC-2: Sensitivity study across a range of β values
>
> Thank you for this question. As per the RCRPS metric (expression in Appendix A.1), since the constraint violation penalty is an additive term on the RCRPS, and β is a factor that solely affects this penalty, the ranking of models will not change with β. The value of 10 was used as the authors of the CiK benchmark used the same value, however this was an arbitrary value and any suitable value can be used [1].
>
> ## RC-3: Choice of tasks considered for reasoning analysis
>
> Thank you for the suggestion. We’d like to note that we explain the rationale for this already in App. C.2. The evaluation on the subset of the tasks is conservative, but still provides a lot of value for future research since our results show that smaller models fail to reason and apply their forecasts even in this regime.
>
> In the case where the entire shape of the forecast can change, and further in different ways due to several plausible slightly different reasoning paths over the context, this would require
> 1. A better method to evaluate reasoning of models instead of comparing with single gold-standard reasoning traces
> 2. A more robust method for identifying meaningful changes in the forecast over the entire forecast region
>
> We consider our current evaluation a first step towards evaluating the reasoning of these models, and hope it inspires more work in this direction.
>
> ## RC-4: Design of IC-DP
>
> Thank you for the question. IC-DP relies on the assumption that there are historical context-aided forecasting tasks that have been previously observed, and hence uses the context, historical, and ground truth (future values) of those tasks like in in-context learning. A clarification here is the “future values” of the in-context task is not in the future, but in the historical data. We call them “future values of the task” to distinguish them from the “historical values of the task” and this distinction is made so the model can use the relation between the context and the future values of the in-context task, infer the causal change that the context has brought about, and use the inferred relation to solve the current task (for which only the historical values and context have been provided).
>
> We hope this clarifies your question, and are happy to explain further if you need.

---

> ### Author Response · Authors · 2025-09-19
> **Response to review - 2**
>
> ## RC-5: Correlation between task difficulty and observed error
>
> Thank you for the suggestion.
>
> We propose a metric to evaluate the error reduction achieved through the proposed task difficulty: for a given main model, we use the area between the random routing curve and the ideal curve as the maximum error reduction possible, and compute the total area captured by the the routing curve that is produced using the task difficulty predicted by the router model. The results are provided in appendix F.2., and the random, ideal and router curves with each main model are provided in App. F.3. We paste the results here for convenience (values to be interpreted as higher the better):
>
>  **Main Model** | **Qwen2.5-0.5B-Inst** | **Qwen2.5-1.5B-Inst** | **Qwen2.5-3B-Inst** | **Qwen2.5-7B-Inst** | **Qwen2.5-14B-Inst** | **Qwen2.5-32B-Inst** | **Qwen2.5-72B-Inst**
> ---|---|---|---|---|---|---|---
>  **Router** |  |  |  |  |  |  |
>  **Qwen2.5-0.5B-Inst** | 66.76 | 48.83 | 13.50 | 29.05 | 22.03 | 68.59 | 67.59
>  **Qwen2.5-1.5B-Inst** | 1.40 | 40.53 | 4.67 | 55.13 | 2.63 | 19.65 | 12.61
>  **Qwen2.5-3B-Inst** | 3.10 | 46.41 | 45.41 | 23.05 | 22.23 | 3.47 | 1.23
>  **Qwen2.5-7B-Inst** | 7.95 | 35.33 | 7.41 | 55.72 | 15.73 | 26.63 | 6.71
>  **Qwen2.5-14B-Inst** | 10.12 | 4.62 | 0.00 | 5.11 | 12.14 | 7.66 | 3.77
>  **Qwen2.5-32B-Inst** | 36.96 | 39.04 | 6.52 | 51.15 | 7.79 | 58.45 | 14.86
>  **Qwen2.5-72B-Inst** | 9.73 | 3.35 | 0.00 | 29.19 | 0.49 | 34.67 | 3.34
>
> This allows us to measure how much of the possible error reduction has been achieved with the RouteDP method, and shows that there is still progress to be made in accurately predicting the difficulty of tasks to capture more of the possible area in the curve, and hence reduce error further.
>
> ## RC-6: Reporting of cost
>
> Thank you for this question. We have now reported the inference time and cost involved (if any) of all experiments that we ran, in the newly-created Appendix H. Appendix H.1 contains the statistics on the DP models, H.2. on ReDP experiments, H.3. on the IC-DP experiments, H.4. on the CorDP experiments, and H.5. on the RouteDP experiments.
>
> Please let us know if you have any further clarifications.
>
> ## RC-7: Nature of textual context in the CiK benchmark
>
> Thank you for the suggestion. We’d like to note that this is a limitation of the CiK benchmark itself as stated in their paper [1]. We believe such a benchmark where contexts are always relevant and necessary makes for a high-quality evaluation set, and can be used to test models with the assumption that the given context will always be relevant, an argument that is also echoed by other works [2]. We consider our evaluation on CiK a first step towards evaluating models on open-ended context-aided forecasting systems. However, the zero-shot methods we propose do not carry the same assumption of context always being relevant, and can be evaluated in other conditions in the same way, when such benchmarks are released.
>
> We hope this clarifies your question, and are happy to discuss further.
>
> ---
>
> ### References:
>
> [1] Andrew Robert Williams, Arjun Ashok, Étienne Marcotte, Valentina Zantedeschi, Jithendaraa Subramanian, Roland Riachi, James Requeima, Alexandre Lacoste, Irina Rish, Nicolas Chapados, et al. Context is key: A benchmark for forecasting with essential textual information. In ICML, 2025
>
> [2] Zhang, X., Han, B., Fang, H., Ansari, A.F., Zhang, S., Maddix, D.C., Hu, C., Wilson, A.G., Mahoney, M.W., Wang, H. and Liu, Y., 2025. Does Multimodality Lead to Better Time Series Forecasting?. arXiv preprint arXiv:2506.21611.

---

### Review · Reviewer_t8SA · 2025-09-06

**Summary Of Contributions:**

Summary Of Contributions

This paper explores methods to enhance the performance of LLMs in time-series forecasting when relevant textual context is provided. The paper's main contributions are the introduction and evaluation of four novel strategies designed to improve zero-shot context-aided forecasting with LLMs: (1) ReDP (Direct Prompting with Reasoning over Context): This strategy focuses on improving interpretability. It prompts the LLM to provide a step-by-step reasoning trace before generating the forecast. This allows for a separate evaluation of the model's reasoning process and its forecasting accuracy, helping to diagnose whether failures are due to poor reasoning or an inability to apply correct reasoning. (2) CorDP (Direct Prompting for Forecast Correction): This method aims to improve applicability in real-world pipelines by using LLMs not as a primary forecaster, but as a "corrector." The LLM is given a forecast from a traditional quantitative model and is tasked with modifying it based on the provided context. This approach leverages the strengths of both traditional forecasting models and the contextual understanding of LLMs, reducing the computational load on the LLM and allowing for easy integration into existing systems (3) IC-DP (In-Context Direct Prompting): To boost accuracy, this strategy provides the LLM with a past example of a similar context-aided forecasting task. The example includes the historical data, context, and ground truth forecast, which helps the model learn to apply its capabilities more effectively. (4) RouteDP (Direct Prompting with Model Routing): This strategy addresses resource efficiency. It uses a smaller LLM as a "router" to estimate the difficulty of a task. Easier tasks are handled by a smaller, more cost-effective model, while more challenging tasks are routed to a larger, more powerful LLM. This approach aims to achieve substantial accuracy improvements at a fraction of the cost of using a large model for every task.
The authors use the Context-Is-Key (CiK) benchmark, which is specifically designed for evaluating context-aided forecasting where the context is essential for an accurate prediction.


Weaknesses:

1. One of the weakness is the absence of systematic comparison with advanced, program-aided frameworks such as TS-Reasoner, which leverage stepwise reasoning and explicit integration of domain knowledge and constraint handling—offering a fundamentally different approach to complex time-series tasks than the prompt-manipulation techniques studied here. This omission limits the contextualization of the proposed strategies within the broader landscape of LLM-based forecasting, leaving it unclear how these advances stack up against or could complement more structured, program-driven reasoning paradigms.

[1] Ye, Wen, et al. "Domain-Oriented Time Series Inference Agents for Reasoning and Automated Analysis." arXiv preprint arXiv:2410.04047 (2024).

2. The analysis shows that smaller models can sometimes generate correct reasoning traces (i.e., they successfully process and explain the context), but this does not reliably translate into improved quantitative forecasts. This suggests a disconnect between explicit reasoning and actionable prediction, where smaller models fail to operationalize their contextual understanding in the generation of better forecasts, resulting in limited overall benefit from context integration in these models.

**Audience:**

Yes

**Audience Explanation:**

The paper addresses an evolving research area: using large language models (LLMs) for zero-shot, context-aided time series forecasting. The work goes beyond simple methods to explore novel strategies that offer distinct benefits over naïve prompting, which is a topic highly relevant to the machine learning research community.

**Broader Impact Concerns:**

A more thorough analysis of generalization and failure modes is necessary, documenting systematic failures and discussing the limitations when facing unseen or out-of-domain contexts to provide realistic expectations for practitioners.

**Claims And Evidence:**

Yes

**Claims Explanation:**

The claims made in the submission are well-supported by clear and convincing evidence within the document.

- ReDP (Direct Prompting with Reasoning over Context): The claim that ReDP improves interpretability and helps diagnose model failures is supported by the detailed "Reasoning Quality Analysis" in the paper. The authors use a protocol with an LLM judge to evaluate reasoning traces and present a table showing that reasoning correctness increases with model size. They also provide multiple examples of both correct and incorrect reasoning traces to illustrate their findings

- CorDP (Direct Prompting for Forecast Correction): The claim that this method enhances applicability and improves performance is backed by a quantitative comparison against the Direct Prompting (DP) baseline. Table 2 shows that CorDP methods achieve the best performance for 8 out of 12 LLMs. The paper also explains how this approach fits into existing workflows by augmenting, rather than replacing, traditional models.

- IC-DP (In-Context Direct Prompting): The assertion that IC-DP substantially improves accuracy is clearly evidenced by the quantitative results.

- RouteDP (Direct Prompting with Model Routing): The claim of optimizing resource efficiency is supported by a compelling plot (Figure 3) and data in Table 4.

**Requested Changes:**

1. The paper needs a new section dedicated to a unified solution or framework. While it introduces four complementary strategies, it lacks a cohesive guide on how to integrate and apply them in practice. This new section should offer a decision workflow, outlining when to use each method based on factors like computational budget or the need for interpretability. It should also clarify how these modules can be combined or composed into a single meta-algorithm for real-world deployment.

2. The authors should clarify the position of their prompting-based methods relative to other advanced techniques like TS-Reasoner, which also use LLMs to 'route' tasks. This discussion should highlight how their proposed strategies offer unique trade-offs in areas like interpretability or efficiency compared to a program-aided approach.

---

> ### Author Response · Authors · 2025-09-19
> **Response to review**
>
> Thank you for your thorough feedback. We appreciate the recognition of our contributions, and we are glad that you find the paper highly relevant. Below, we clarify all the points raised in the review.
>
> (Weakness abbreviated as **W**, Requested Change abbreviated as **RC**)
>
> ## W-1/RC-2: Systematic comparison with frameworks such as TS-Reasoner
>
> We thank the reviewer for pointing us to the TS-Reasoner paper [1], which is highly related to ours. We find that the TS-Reasoner approach is well placed to take advantage of the insights we have developed in this paper, and that the Task Decomposer already plays the role of a router in the TS-Reasoner architecture. So if DP was included in its selection of models (with $C$ containing the textual context) and the other components removed, then TS-Reasoner would be equivalent to our RouteDP method.
>
> We have updated the paper to discuss the possibilities of using an agentic approach to combine LLM and TS in our future work section, citing TS-Reasoner as an example of research already going in that direction.
>
> [1] Ye, Wen, et al. "Domain-Oriented Time Series Inference Agents for Reasoning and Automated Analysis." arXiv preprint arXiv:2410.04047 (2024).
>
> ##  W-2: Inability of small models to apply reasoning to improve forecasts
>
> We thank the reviewer for noting this finding from our work. We agree that the inability of small models to translate their reasoning traces into useful context-aided forecasts indicates a weakness of these models, and limits their applicability. This shows that improving the reasoning application of these models is a low-hanging fruit to improve their context-aided forecasting capabilities. We’d like to note that we discuss this in detail in Sec 4.3, with examples in App. C.6. We are more than happy to clarify this point further if needed.
>
> ## RC-1: Cohesive guide on integrating and applying the proposed strategies
>
> Thank you for this suggestion. We have now included a new **Section 8: Unifying the Proposed Strategies** which discusses how the strategies can be integrated cohesively in a framework for practitioners seeking to deploy systems in the real-world. The section references the inference time and cost of different methods, as well as new results that demonstrate the complementarity of the methods. We discuss factors such as available compute budget and time, the need for interpretability, and availability of historical context-aided forecasting tasks, that can influence the selection of specific zero-shot context-aided forecasting method(s).
>
> We hope our responses address your concerns, and are happy to clarify any concerns further.

---

### Author Response · Authors · 2025-09-19
**General comment**

We sincerely thank the reviewers for taking the time to review and provide useful feedback on our work, all of which have significantly improved the quality of the paper. We have addressed each reviewer’s concerns separately, and have uploaded a new version of the paper that is updated based on the feedback. For convenience, we list here all new additions in this version, which are formatted in blue in the paper:

**\[Comparison with agentic frameworks in the future work\]** We have added a comparison with agentic frameworks such as TS-Reasoner in Section 9 (Discussion and Future Work).

**\[Cohesive guide on integrating and applying the proposed strategies\]** We have now included a new Section 8: Unifying the Proposed Strategies, that discusses how the strategies can be integrated cohesively in a framework for practitioners seeking to deploy systems in the real-world.

**\[Detailed discussion on the human verification of ground truth reasoning traces\]** We have now included a detailed discussion on how each ground truth reasoning trace was verified, in App C.4.1, including each task whose reasoning trace was modified.

**\[Manual reasoning quality evaluation with human judges\]** To validate the quality of the LLM-judge evaluation of reasoning correctness, we ran a human evaluation experiment where humans were used as the judges and asked to compare a ground truth reasoning trace with a model’s reasoning trace. We detail the procedure and discuss the correlation with the LLM judge evaluation in App. C.6, along with the statistics of agreement for each model under study.

**\[Multiple strategies in tandem\]** We ran two sets of experiments that demonstrate the complementary nature of the proposed strategies, and the potential value in combining them:

1. We combined ICDP and CorDP, where the model is expected to perform a forecast correction and if further provided with a historical in-context example of a forecast correction. The results of this experiment are in App. D.3.
2. We used RouteDP with IC-DP and CorDP in two separate experiments, where the downstream model performs IC-DP or CorDP instead of DP, respectively. Here, routing is similarly performed with predicted difficulty scores from the router model, and the routing curve is compared with the respective random and ideal routing curves. We present and discuss these results in App. F.4.

**\[Experiments with closed-source LLMs\]** We conducted experiments with the IC-DP and CorDP methods, with two closed-source LLMs, namely the GPT-4o and GPT-4o-mini models. We have added these results to the paper, in the respective tables (Table 2 and Table 3\) and figures (Fig 2\) concerning each method in the main text and appendix. The respective paragraphs interpreting the results were also updated to reflect the new results.

**\[Inference Time and Cost\]** We have added the inference time and cost of all experiments that were run in the paper in Appendix H.

We thank the reviewers again for the effort in reviewing our work, and hope that the rebuttal and new version of the paper addresses the concerns. We are more than happy to clarify any outstanding concerns further.

---

### Decision · Action_Editor_uVNE · 2025-10-26

**Recommendation:** Reject

**Additional Comments:**

The paper addressed an important problem on zero-shot contextual forecasting. The proposed solution is reasonable supported by interesting experiment results. The paper receives mixed reviews. Therefore a series of discussions were held between reviewers, AE, and EIC. After the discussion, it was agreed that the evaluation could be further improved.

The authors are advised to conduct more rigorous evaluation and resubmit the draft. The suggested revisions include:  (a) providing guidelines on how to make the evaluation more objective, (b)  recruiting external subjects to evaluate reasoning traces and compare the results with the ones evaluated by the authors themselves, (c) adding discussions on whether the results/analysis results change under the new evaluation.

**Audience:**

Yes

**Audience Explanation:**

Zero-shot context-aided forecasting is an important and timely topic. The research results would be appreciated by a large body of the TMLR audience who are interested in time series.

**Claims And Evidence:**

No

**Claims Explanation:**

The majority of the claims are well-supported. One reviewer has raised the concern that evaluation of the results are conducted by the authors themselves.

**Resubmission Of Major Revision:**

The authors may consider submitting a major revision at a later time.

---

> ### Author Response · Authors · 2025-11-19
>
> Dear Action Editor,
>
> Thank you for your efforts in overseeing the review process.
>
> We acknowledge the decision, and agree that these points will strengthen the work. We will revise the paper accordingly and resubmit at a later time.
>
> Authors